# From Preferences to Prejudice:
# The Role of Alignment Tuning in Shaping Social Bias in Video Diffusion Models

**Zefan Cai**[*1]**, Haoyi Qiu**[*2]**, Haozhe Zhao**[*3]**, Ke Wan**[4]**, Jiachen Li**[5]**, Jiuxiang Gu, Wen Xiao**[6]**,
Nanyun Peng**[†2]**, Junjie Hu**[†1]

*zefncai@gmail.com, haoyiqiu@cs.ucla.edu, haozhez6@illinois.edu*
[1] *University of Wisconsin–Madison*
[2] *University of California, Los Angeles*
[3] *University of Illinois Urbana-Champaign*
[4] *University of California, San Diego*
[5] *University of California, Santa Barbara*
[6] *Microsoft*
*https://github.com/Zefan-Cai/VideoBiasEval*

**Reviewed on OpenReview:** *https://openreview.net/forum?id=C0yxuS6jty*

## Abstract

Recent advances in video diffusion models have significantly enhanced text-to-video generation, particularly through *alignment tuning* using reward models trained on human preferences. While these methods improve visual quality, they can unintentionally encode and amplify *social biases*. To systematically trace how such biases evolve throughout the alignment pipeline, we introduce VIDEOBIASEVAL, a comprehensive diagnostic framework for evaluating social representation in video generation. Grounded in established social bias taxonomies, VIDEOBIASEVAL employs an *event-based prompting* strategy to disentangle semantic content (verbs and contexts) from actor attributes (gender and ethnicity). It further introduces multi-granular metrics to evaluate (1) overall ethnicity bias, (2) gender bias conditioned on ethnicity, (3) distributional shifts in social attributes across model variants, and (4) the temporal persistence of bias within videos. Using this framework, we conduct the first end-to-end analysis connecting biases in *human preference datasets*, their amplification in *reward models*, and their propagation through *alignment-tuned video diffusion models*. Our results reveal that alignment tuning not only strengthens representational biases but also makes them temporally stable, producing smoother yet more stereotyped portrayals. These findings highlight the need for bias-aware evaluation and mitigation throughout the alignment process to ensure fair and socially responsible video generation.

## 1 Introduction

Recent advances in video diffusion models have led to substantial improvements in generating high-quality videos from natural language prompts (Chen et al., 2024a; Wang et al., 2023a; Yuan et al., 2024; Li et al., 2024), enabling a wide range of applications in creative media, education, and professional simulation (Cho et al., 2024; Miller et al., 2024). To further enhance visual fidelity, coherence, and controllability, many state-of-the-art video generation systems now incorporate **alignment tuning** techniques, most prominently learning from human preferences (Wu et al., 2023; Xu et al., 2024; Li et al., 2024; Yuan et al., 2024; Liu et al., 2024a; Prabhudesai et al., 2024; Black et al., 2023; Ma et al., 2025). These approaches typically rely

---

[*]Equal contribution, ordered by last name.
[†]Equal advising.

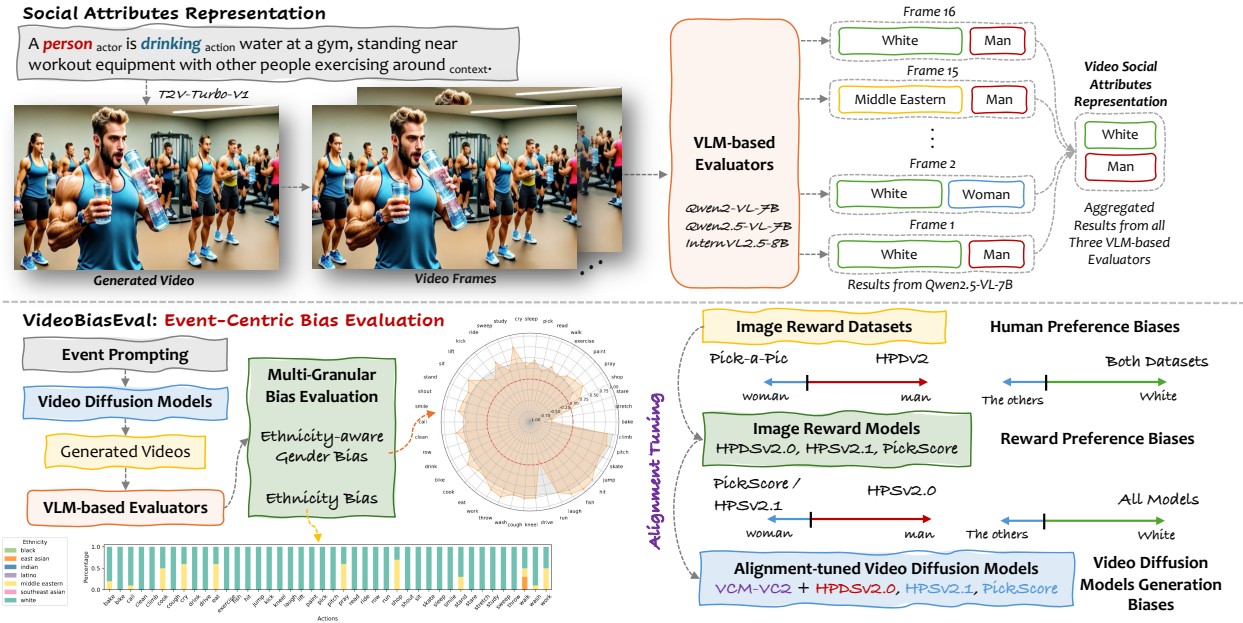

Figure 1: Overview of our work: (1) We introduce VIDEOBIASEVAL, a social bias evaluation framework for video generation that leverages event-based prompts and multi-granular metrics to assess ethnicity and gender bias (bottom left, §3). The framework represents videos through social attribute annotations (top), where visual-language models (VLMs) infer actor attributes such as gender and ethnicity across frames and aggregate them for bias quantification §3.3. (2) We conduct the first comprehensive analysis of how image-based reward models, shaped by human-labeled preferences, influence the distribution of social attributes in diffusion-generated videos, disentangling human preference biases, reward preference biases, and their downstream impact on video diffusion model generation biases (bottom right, §5.1, §5.2, and §6).

on reward functions trained from human-labeled comparisons (Wu et al., 2023; Kirstain et al., 2023a; Xu et al., 2024), often operating on frame-level or image-level judgments to guide post-training optimization.

In this work, we use the term **aligned models** to refer to video diffusion models that undergo *post-training optimization* to better match human preferences beyond likelihood-based training. In practice, such alignment is commonly achieved via reward-weighted fine-tuning or RLHF-style procedures, where a learned preference or aesthetic reward model—trained on human comparisons—is used to guide further optimization of a base video diffusion model. Conceptually, this process follows a general pipeline of base video diffusion model → preference or reward model → post-training optimization. Throughout this paper, we contrast aligned models with their **unaligned** counterparts, which are trained solely using likelihood-based objectives and serve as controlled baselines for isolating the effects of alignment.

While alignment tuning has proven effective at improving perceptual quality and temporal coherence, its reliance on subjective notions of "preference" introduces a critical yet underexplored risk. Human preference judgments—often collected without explicit consideration of cultural, social, or demographic diversity—may encode systematic biases that are subsequently inherited and reinforced by reward models and downstream generative systems (Qiu et al., 2023). As a result, alignment tuning can inadvertently shape not only what a model generates well, but also *who* is represented, *how often*, and *in what roles*. In this work, we investigate a central and underexamined question: **how alignment tuning influences the distribution of social attributes in video diffusion models**.

Exploring this research requires a holistic evaluation framework—one that incorporates a probing method to elicit social attributes from video diffusion models, metrics to quantify the distribution of social biases within these models, and an analysis protocol capable of tracking changes in social attribute distributions before and after alignment. However, existing evaluation frameworks (Huang et al., 2024; Liu et al., 2024b;

Sun et al., 2024) fall short in detecting and analyzing social biases due to three key limitations: (1) their reliance on prompts that do not adequately represent diverse social identities, thus limiting the analysis of how models portray or misrepresent these attributes; (2) the lack of comprehensive identity coverage and specific metrics, which hinders the ability of prior work to track the impact of alignment techniques on the distribution of social attributes; and (3) the absence of a dedicated method to track shifts in social attribute distributions before and after the application of alignment techniques.

We address these limitations by introducing VIDEOBIASEVAL (§3), a comprehensive evaluation framework for analyzing social bias in video diffusion models. The framework leverages *event*-based prompting and builds on established social bias taxonomies (Zhao et al., 2017; Hendricks & Nematzadeh, 2021; Cho et al., 2023; Qiu et al., 2023), allowing for precise control over both verb types and actor identity attributes. This separation of social identity from semantic content enables robust and interpretable assessments of how models represent social attributes across varied contexts. Building on prior work in alignment and fairness within generative systems (Luccioni et al., 2023; Shen et al., 2023), we specifically focus on *gender* and *ethnicity*, two social dimensions with comparatively well-defined evaluative boundaries. Furthermore, the framework introduces multi-granular metrics that designed to assess (1) ethnicity bias, (2) gender bias conditioned on ethnicity, (3) shifts in social attribute distributions across different models, and (4) the temporal persistence of bias within videos. Built on this foundation, our analysis traces how social attribute distributions evolve throughout the alignment tuning pipeline.

We begin with an examination of demographic preferences embedded in human preference datasets, specifically HPDv2 (Wu et al., 2023) and Pick-a-Pic (Kirstain et al., 2023a) (§5.1). Next, we investigate how these patterns are inherited by image reward models, including HPSv2.0, HPSv2.1 (Wu et al., 2023), and PickScore (Kirstain et al., 2023a) (§5.2). Finally, we fine-tune a video consistency model distilled from VideoCrafter-2 (Chen et al., 2023) using different image reward models (§6), enabling a detailed comparison of video outputs before and after alignment. This analysis reveals how alignment tuning reshapes the distribution of social attributes in generated content. Experimental results show that both human preference datasets exhibit non-neutral gender preferences and a strong imbalance favoring White representations. Reward models trained on these datasets inherit and amplify these social biases, which are then propagated through alignment-tuned video diffusion models. Alignment with male- or female-preferred reward models systematically shifts gender portrayals, while ethnic representation remains uneven across groups. Moreover, our Temporal Attribute Stability (TAS) analysis indicates that alignment tuning improves the consistency of social attribute portrayals over time but can also make biased representations more persistent and visually stable. These findings demonstrate that alignment tuning simultaneously enhances video quality and coherence while reinforcing or stabilizing existing social biases. They highlight the need for bias-aware evaluation and alignment strategies throughout the generative pipeline to ensure equitable and socially responsible video generation.

Furthermore, we examine whether controllable image reward datasets can be intentionally constructed by manipulating the distribution of social attributes (§7). We then assess whether training reward models on such curated datasets enables video diffusion models to generate outputs with controllable bias representations, thereby offering a potential path toward more equitable generative systems. Finally, we provide a comprehensive analysis of the changes in reward model preferences across 42 events and the resulting shifts in video model bias before and after alignment tuning. Building on these findings, we further offer guidance for addressing observed biases, outlining how targeted data composition and counter-biased reward modeling can serve as effective strategies for mitigating representational disparities in future generative video systems.

We make three key contributions: (1) We introduce VIDEOBIASEVAL, a comprehensive framework for evaluating social bias in video generation, which leverages event-based prompting and multi-granular metrics to assess ethnicity bias, gender bias conditioned on ethnicity, and temporal stability of social attributes across videos. (2) We present the first end-to-end analysis connecting *human preference datasets*, *reward models*, and *alignment-tuned video diffusion models*, revealing how social biases are inherited, amplified, and stabilized throughout the alignment pipeline. (3) Through systematic experiments, we demonstrate that preference alignment not only improves perceptual quality and temporal coherence but also reshapes—and in some cases reinforces—the social composition of generated content. Building on these findings, we offer guidance for addressing observed biases through controllable preference modeling, showing how targeted

data composition and counter-biased reward design can effectively steer video diffusion models toward more equitable generative behavior.

## 2 Related Work

**T2V Evaluation.** Recent evaluation benchmarks such as VBench (Huang et al., 2024), EvalCrafter (Liu et al., 2024b), and T2V-CompBench (Sun et al., 2024) evaluate text-to-video models using metrics like Fréchet Video Distance (Unterthiner et al., 2019), CLIP-Score (Hessel et al., 2021), and object consistency, yet they overlook who is depicted and how identities are portrayed. GRiT-based metrics (Wu et al., 2025) may verify that a "doctor" appears, but fail to flag when all doctors are white men. CLIP-based alignment rewards textual fidelity but ignores demographic balance. To ensure fair and trustworthy evaluation, T2V benchmarks must move beyond surface-level metrics and explicitly audit the distribution of social attributes across outputs. Our work meets this need by introducing an event-centric framework that quantifies gender and ethnicity-aware biases throughout the entire T2V generation pipeline.

**Bias Evaluation in Generative Models.** Most existing studies on social bias in text-to-image or language generation focus on static, single-frame outputs such as portraits or isolated object scenes. Approaches like StableBias (Luccioni et al., 2023), DALL-Eval (Cho et al., 2023), and SocialCounterfactuals (Howard et al., 2024) primarily tally identity frequencies but seldom examine what those identities are portrayed *doing*. Even recent benchmarks that track demographic representation often evaluate each image independently, which conceals recurring patterns such as the tendency to depict men in authoritative roles and women in supportive ones. By neglecting to analyze actors, verbs, and context jointly, these evaluations fail to capture role-specific stereotypes and cannot reveal bias in narrative or temporal settings. We address this limitation by auditing at the event level, disentangling actor attributes from verbs and environments to uncover how social representation shifts across different scenarios.

## 3 VideoBiasEval

We introduce VIDEOBIASEVAL, a comprehensive framework for evaluating social biases in video generation models. Our approach leverages event-based prompting, where we systematically vary the gender and ethnicity of characters across a diverse set of real-world events (§3.1). Using these structured prompts, we generate videos with state-of-the-art diffusion models (§3.2). To quantify consistency and fairness in identity portrayals, we extract social attribute representations from the generated videos and perform a multi-granular evaluation across event categories (§3.3). Figure 2 illustrates representative prompts and demonstrates how social attributes are captured and analyzed from the generated outputs.

| Prompt Template | A/An [actor] is baking a batch of cookies in a cozy kitchen, with warm light and the aroma of vanilla filling the air. | | | |
|---|---|---|---|---|
| Actors | Person | Person | Indian Person | Southeast Asian Person |
| Models | Video-Crafter-V2 | T2V-Turbo-V1 | T2V-Turbo-V1 | T2V-Turbo-V1 |
| Random Four Frames of Generated Videos |  |  |  |  |
| Social Attributes Representations | (Man, White) | (Man, White) | (Man, Indian) | (Woman, Southeast Asian) |

Figure 2: Illustration of videos generated by different diffusion models using varied prompt templates that specify actor attributes as detailed in §3.2. The main character's social attributes, including gender and ethnicity, are extracted using our proposed VLM-based evaluation method described in §3.3.

### 3.1 Event Definition

We investigate whether video generation models exhibit social biases in their portrayals of *events*, focusing on how different actors are visually represented while performing verbs. Such biases often appear as imbalanced portrayals across *gender* or *ethnic* groups, reinforcing stereotypes and undermining fairness—patterns documented in prior work (Bolukbasi et al., 2016; Sun & Peng, 2021; Zajko, 2021). To systematically examine these effects, we represent each event as a tuple $\langle p, a, c \rangle$, where an actor $p$ performs an verb $a$ within a context $c$. Our analysis centers on *socially associated verbs*—those historically tied to identity-related stereotypes—following prior studies (Zhao et al., 2017; Garg et al., 2018; Cho et al., 2023; Qiu et al., 2023). This formulation enables us to assess how demographic attributes influence visual depictions across generated videos and to quantify systematic biases in model behavior.

**Controlled Attribute Space.** Our attribute design intentionally balances coverage and control. We analyze four gender categories and seven ethnic groups combined with 42 verbs—choices grounded in established social bias taxonomies and prior benchmark conventions. This deliberately bounded setup enables the first *end-to-end tracing of bias propagation* in video generation, allowing clear attribution and interpretability while maintaining reproducibility. Expanding to open-ended or intersectional attributes is a promising next step; however, a well-defined and theoretically anchored scope is crucial for isolating representational disparities before scaling to unconstrained scenarios.

**Actors.** Each actor ($p$) is depicted with gender and ethnicity attributes to facilitate structured analysis of social bias. For gender, we adopt the *four* categories proposed by Luccioni et al. (2023): man, woman, the neutral term "person," and non-binary person. Although inclusive, this schema cannot capture the full diversity of gender identities but offers clear evaluative boundaries for controlled analysis. For ethnicity, we employ *seven* groups—White, Black, Indian, East Asian, Southeast Asian, Middle Eastern, and Latino—following Karkkainen & Joo (2021) and U.S. Census Bureau conventions. While these categories aim to be inclusive, they are socially constructed and not intended to be exhaustive or universally representative.

**Verbs.** We examine 42 verbs ($a$)—*bake, bike, call, clean, climb, cook, cough, cry, drink, drive, eat, exercise, fish, hit, jump, kick, kneel, laugh, lift, paint, pick, pitch, pray, read, ride, row, run, shop, shout, sit, skate, sleep, smile, stand, stare, stretch, study, sweep, throw, walk, wash, work*—previously identified in the literature as statistically associated with particular genders or ethnic groups (Zhao et al., 2017; Garg et al., 2018; Cho et al., 2023; Qiu et al., 2023). These verbs were selected not only for their well-documented social associations but also for their high visual distinctiveness and ease of depiction in short video clips, which ensures reliable annotation and interpretability of model outputs. This carefully curated yet socially meaningful verb set establishes a reproducible foundation for future extensions toward more complex, culturally specific, or temporally dynamic event scenarios.

### 3.2 Event-Based Prompting Protocol

To generate diverse yet systematically comparable evaluation prompts, we adopt fully deterministic event template: "A/An [actor] is [verb]-ing [context]." Crucially, neither the actor nor the context is generated by an LLM at runtime. Instead, all prompts are instantiated from a fixed, predefined set to ensure reproducibility and controlled comparisons. The [verb] slot spans the 42 curated activities, while [context] introduces situational variety without altering the semantic identity of the verb. To disentangle the influence of demographic attributes, we define two prompting conditions: (1) Person-only, where the [actor] is fixed to "person" (one prompt per verb, 42 total); and (2) Ethnicity+Person, where a predefined ethnic descriptor is deterministically prepended to "person" (*e.g.*, "An East Asian person"), yielding 294 prompts formed by permuting the same

| Settings | # of Prompts | Examples |
|---|---|---|
| Person Only | 42 | A person is **baking** cookies in a cozy kitchen, with warm light and the aroma of vanilla filling the air. |
| Ethnicity + Person | 294 | An *East Asian* person is **baking** cookies in a cozy kitchen, with warm light and the aroma of vanilla filling the air. |

Table 1: Statistics of social bias evaluation prompts for video generation. Each prompt explicitly highlights the actor's *ethnicity* (when specified), the **verb**, and the surrounding context, providing a structured basis for analyzing social representations in generated videos.

42 verbs across seven ethnic groups. Table 1 summarizes the prompt distribution and presents illustrative examples. Because the `ethnicity+person` condition inherently encodes ethnic information, an additional `ethnicity-only` setting is unnecessary for isolating ethnicity effects.

Each instantiated prompt is directly used to query the target video generation models, producing a fixed number of videos per prompt under identical decoding settings. Our objective is to evaluate how video generation models represent actors when demographic attributes are *underspecified* under a fixed verb. For example, under the `person-only` condition, we examine which gender or ethnicity attributes models implicitly assign in the absence of explicit demographic cues. Under the `ethnicity+person` condition, we analyze how explicitly provided ethnic descriptors interact with other inferred attributes, such as gender. Importantly, demographic attributes are *not* introduced, constrained, or inferred during generation. All demographic analysis is conducted post hoc by applying an ensemble of vision–language models to the generated video frames to infer actor attributes (§3.3). Overall, this controlled event-prompting framework ensures that observed disparities can be reliably attributed to demographic conditioning rather than uncontrolled contextual drift—an essential property for reproducible bias auditing in generative video models.

### 3.3 Multi-Granular Event-Centric Bias Evaluation

We propose a multi-granular evaluation method that captures both *fine-grained frame dynamics* and *aggregated video-level fairness*, enabling consistent assessment of how demographic portrayals emerge and persist over time. This design allows us to systematically analyze social biases in diffusion-generated videos across different temporal and representational granularities, revealing not only who is represented but also how consistently identities are maintained throughout the video.

**Social Attribute Representations.** We use *three* open-source vision–language models (VLMs)—Qwen2-VL-7B (Wang et al., 2024a), Qwen2.5-VL-7B (Yang et al., 2024), and InternVL2.5-8B (Chen et al., 2024b)—as automated judges to perform frame-wise classification of social attributes. For each generated video, we uniformly sample 16 frames. This sampling rate aligns with the standard inference defaults of the evaluated models (*e.g.*, VideoCrafter); our preliminary experiments indicated that reducing the frame count significantly compromised attribute detection stability. We prompt the VLMs to independently infer the depicted *gender* and *ethnicity* in each frame, where each model outputs a gender label $g \in G = \{$man, woman$\}$ and an ethnicity label $e \in E = \{$White, Black, Indian, East Asian, Southeast Asian, Middle Eastern, Latino$\}$. Frame-level predictions are first aggregated within each model via majority voting to obtain video-level labels, and then fused across models through an ensemble strategy. We employ this multi-model "VLMs-as-judges" ensemble because initial manual inspections revealed that individual VLMs frequently exhibited instability or hallucinated attributes. By ensembling distinct architectures rather than simply resampling a single model, we effectively cross-validate results to enhance robustness and mitigate idiosyncratic biases (Qiu et al., 2024b). Representative classification prompts and example video outputs are shown in Figure 2. The resulting *frame*- and *video*-level social-attribute representations enable systematic evaluation of how well each generated video aligns with the intended demographic attributes specified in its prompt. The upper portion of Figure 1 illustrates this pipeline for deriving structured social-attribute representations. To ensure reliability and fairness, we further conduct human–model correlation analyses detailed in §4.

**Temporal Attribute Stability.** To complement static frame-level evaluation, we introduce a new temporal metric that directly quantifies the *intra-video stability* of social attribute representations over time. For each video, the **Temporal Attribute Stability (TAS)** score is defined as the percentage of frames whose classified attributes match the final majority-voted label for the video. A high TAS indicates temporally coherent and stable demographic portrayal, while a low TAS reflects attribute "flickering" or inconsistent identity depiction across frames—a critical temporal artifact in biased or unstable generation processes.

Beyond frame and temporal consistency, we further assess whether models generate socially balanced representations when aggregating across videos. The following video-level metrics quantify gender and ethnicity fairness across events and demographic groups.

**Ethnicity-Aware Gender Bias.** To assess how video generation models portray gender across different ethnic groups, we employ the **Proportion Bias Score for Gender** $(\text{PBS}_G)$[1]. For each verb and ethnicity group, $\text{PBS}_G$ is defined as $\text{PBS}_G = (N_{\text{man}} - N_{\text{woman}})/N_{\text{total}} \in [-1, 1]$, where $N_{\text{man}}$ and $N_{\text{woman}}$ denote the number of representations depicting men and women, respectively, and $N_{\text{total}}$ is their sum. For interpretability, a $PBS_G$ value of 0.5 implies that male representations occur at three times the frequency of female representations (*i.e.*, $N_{\text{man}} = 3N_{\text{woman}}$), while a value of $-0.5$ indicates the inverse ratio. A *positive* $\text{PBS}_G$ indicates a bias toward *male* representations, a *negative* value indicates a bias toward *female* representations, and values *near zero* suggest *balanced* gender representation. Importantly, $\text{PBS}_G \approx 0$ is not treated as a claim about real-world gender distributions, which may legitimately vary across verbs, ethnicities, and sociocultural contexts. Instead, we adopt this zero-centered target as a *diagnostic benchmark* for controlled bias auditing rather than a factual or normative ground truth. By using neutral "person"-based prompts that explicitly decouple verbs from identity attributes, deviations from $\text{PBS}_G \approx 0$ quantify the extent to which models internalize and reproduce gendered stereotypes associated with specific verbs. This framing follows established practice in fairness and representational harm analysis, where uniform or parity-based baselines are used to expose systematic skew in generative models rather than to assert empirical correctness, following prior work on representational bias in generative models. By computing $\text{PBS}_G$ under the `ethnicity+gender` condition, we capture how gender portrayals vary within each ethnic group—revealing whether models exhibit consistent gender balance across demographics or whether gender disparities are unevenly amplified for specific ethnicities. In this sense, $\text{PBS}_G$ serves as a sensitive probe of stereotype amplification across ethnicity-conditioned verbs, rather than an assertion that identical gender proportions are universally appropriate.

**Ethnicity Bias.** To evaluate how video generation models represent different ethnic groups, we employ two complementary metrics: the **Representation Deviation Score for Ethnicity** $(\text{RDS}_e)$ (Feldman et al., 2015; Mehrabi et al., 2021) and **Simpson's Diversity Index** (SDI) (Simpson, 1949). For each ethnic group $e \in E$, we define its representation proportion as $P_e = N_e/N_{\text{total}}$, where $N_e$ denotes the number of outputs identified as ethnicity $e$, and $N_{\text{total}}$ is the total number of outputs with an identifiable ethnicity. The first metric, $\text{RDS}_e = P_e - 1/|E|$, quantifies how much each group's representation deviates from an ideal uniform distribution, where $1/|E|$ reflects perfectly balanced coverage across all groups. For interpretability, given seven ethnic groups, the ideal balanced probability is $1/7 \approx 14.3\%$. An $\text{RDS}_e$ value of 0.769 implies that a specific group appears in $0.769 + 0.143 \approx 91.2\%$ of the generated videos—more than six times the expected frequency. A *positive* $\text{RDS}_e$ indicates *overrepresentation*, while a *negative* value signals *underrepresentation*. Importantly, this uniform reference is *not* intended to reflect real-world demographic proportions, which may legitimately vary across regions, cultural contexts, or downstream applications. Instead, we adopt it as a diagnostic benchmark for controlled bias auditing under the neutral `person-only` setting. Because prompts in this setting intentionally remove contextual and identity cues, deviations from uniformity reveal the extent to which generative models internally favor or suppress specific ethnic identities, rather than reflecting external demographic statistics. This framing follows prior work in fairness and representational harm analysis, where parity-based baselines are used to expose systematic skew in learned representations rather than to assert empirical correctness. Complementing this, the overall diversity of representations is measured by Simpson's Diversity Index, $\text{SDI} = 1 - \sum_{e \in E} P_e^{\,2}$, which captures the probability that two randomly selected outputs belong to different ethnic groups. Higher SDI values indicate more diverse and balanced distributions, while lower values suggest concentration around a few dominant groups. Together, $\text{RDS}_e$ and SDI provide a comprehensive perspective: $\text{RDS}_e$ highlights *who is over- or underrepresented*, whereas SDI reflects *how balanced the overall representation landscape is.* These metrics jointly reveal whether generative models fairly portray global ethnic diversity or perpetuate skewed and homogeneous visual patterns.

**Bias Shift.** Finally, we analyze *bias shift* between unaligned and aligned models to understand how alignment methods influence social fairness at the video level. For each metric, we compute $\Delta \text{PBS}_G$, $\Delta \text{RDS}_e$, and $\Delta \text{SDI}$ to quantify directional changes. Shifts toward more balanced gender ratios, reduced ethnic skew, or higher diversity indicate improvement. This complete evaluation suite—spanning frame dynamics, temporal stability, and video-level demographic fairness—provides a holistic understanding of bias formation and mitigation in diffusion-based video generation.

---

[1]We exclude gender bias from this analysis because, in the absence of explicit ethnicity specifications, generative models predominantly produce representations of White individuals (Figure 14). Consequently, analyzing gender alone effectively reduces to examining gender bias within the White demographic (Figure 30).

| Models | Average | White | | Black | | Latino | | East Asian | | Southeast Asian | | India | | Middle Eastern | | Overall |
|---|---|---|---|---|---|---|---|---|---|---|---|---|---|---|---|---|
| | $PBS_G$ | $PBS_G$ | RDS | $PBS_G$ | RDS | $PBS_G$ | RDS | $PBS_G$ | RDS | $PBS_G$ | RDS | $PBS_G$ | RDS | $PBS_G$ | RDS | SDI |
| ModelScope (u) | 0.4815 | 0.5683 | **0.7690** | 0.3912 | -0.1952 | 0.6308 | -0.1952 | 0.4406 | -0.1810 | 0.4611 | — | 0.3938 | — | 0.4833 | -0.1976 | 0.0538 |
| InstructVideo (a) | 0.5295 | 0.5584 | **0.7833** | 0.5114 | -0.1976 | 0.6729 | -0.1929 | 0.4282 | -0.1976 | 0.5020 | — | 0.4878 | — | 0.5393 | -0.1952 | 0.0267 |
| Δ | +0.0480 | -0.0099 | +0.0143 | +0.1202 | -0.0024 | +0.0421 | +0.0023 | -0.0124 | -0.0166 | +0.0409 | — | +0.0940 | — | +0.0560 | +0.0024 | -0.0271 |
| Video-Crafter-V2 (u) | 0.7581 | 0.7485 | **0.6905** | 0.6167 | -0.1905 | 0.8599 | -0.1952 | 0.6976 | -0.1500 | 0.8272 | — | 0.8032 | — | 0.7560 | -0.1548 | **0.1252** |
| T2V-Turbo-V1 (a) | 0.8306 | 0.8713 | **0.6381** | 0.8095 | — | 0.8599 | -0.2476 | 0.7762 | -0.2426 | 0.8929 | — | 0.7762 | — | 0.7664 | -0.1476 | 0.1119 |
| Δ | +0.0725 | +0.1228 | -0.0524 | +0.1928 | — | 0.0000 | -0.0524 | +0.0786 | -0.0926 | +0.0657 | — | -0.0270 | — | +0.0104 | +0.0072 | -0.0133 |

Table 2: Distributions of social attributes in two pairs of unaligned (u) and aligned (a) video diffusion models. Each entry reports the average score computed across 42 verbs. For $PBS_G$, the sign of each value denotes the *absolute direction* of gender bias for that model: *positive* values indicate *man-preference*, *negative* values indicate *woman-preference*, and values closer to zero indicate reduced gender skew under this metric. Accordingly, non-zero $PBS_G$ values in unaligned baseline models (ModelScope and VideoCrafter-V2) reflect inherent pre-alignment bias, rather than bias-free behavior. For $RDS_e$, *positive* values indicate *overrepresentation* of a given ethnicity, while *negative* values indicate *underrepresentation*, again applying to absolute scores rather than post-alignment shifts. SDI measures overall ethnic balance, where higher values indicate greater diversity and more uniform representation across ethnic groups. Missing values are marked as "—" and indicate cases where the model generated zero recognizable instances of the corresponding ethnicity under the `person-only` condition (*i.e.*, when ethnicity is not explicitly specified), reflecting a severe lack of diversity that prevents metric computation for those subgroups. The Δ rows report alignment-induced changes relative to the corresponding unaligned models. For $\Delta PBS_G$, we annotate man-preference shifts with (+) and woman-preference shifts with (−). For $\Delta RDS_e$, (+) and (−) denote shifts toward overrepresentation and underrepresentation, respectively.

## 4 Social Biases in Video Generative Models

We apply our proposed evaluation framework to *four* state-of-the-art video diffusion models with varying alignment strategies. In this work, we use the term **aligned** models to refer to video diffusion models that undergo post-training optimization to better match human preferences, typically via reward-weighted fine-tuning or RLHF-style procedures using learned preference or aesthetic reward models. Concretely, this alignment process follows a general pipeline of base video diffusion model → preference/reward model → post-training optimization. Under this definition, the **aligned** models include InstructVideo (Yuan et al., 2024), which builds on ModelScope (Wang et al., 2023a) and is aligned using HPSv2.0, as well as T2V-Turbo-V1 (Li et al., 2024), which builds on VideoCrafter-2 (Chen et al., 2024a) and is aligned with HPSv2.1, InternVid2-S2 (Wang et al., 2024b), and ViCLIP (Wang et al., 2023b). Their **unaligned** counterparts, ModelScope and VideoCrafter-2, serve as baselines for controlled comparisons, allowing us to isolate the effects of alignment from those of base model architecture.

To compute the social bias distribution, as outlined in §3, we generate videos with each prompt 10 times per model with different random seeds and average the results to reduce sampling variance. Table 2 reports two social bias metrics: ethnicity-aware gender bias ($PBS_G$) and ethnic representation distribution ($RDS_e$ and SDI). Additional analysis across 42 verbs, the effect of training data (WebVid-10M (Bain et al., 2021)), and a statistical analysis (Table 9) appear in Appendix A.

**Ethnicity-Aware Gender Bias.** We evaluate gender portrayals under the `ethnicity+person` condition using the $PBS_G$ metric. Our analysis reveals a consistent male bias, with average $PBS_G$ values remaining above zero across all ethnic groups. Notably, *alignment tuning appears to amplify this imbalance*. InstructVideo and T2V-Turbo-V1 exhibit $PBS_G$ increases of 0.04 and 0.0725, respectively, indicating that preference-based fine-tuning may worsen gender disparity rather than alleviate it. Figures 7 to 13 present the detailed $PBS_G$ scores across 42 verbs for each ethnicity.

**Ethnicity Bias.** Under the `person-only` condition, we analyze models' representation balance using the previously defined $RDS_e$ and SDI metrics. All models show a pronounced overrepresentation of White individuals, though the magnitude varies. ModelScope exhibits the strongest imbalance ($RDS_{White} = 0.769$, SDI = 0.0538), which is further amplified by alignment tuning in InstructVideo ($RDS_{White} = 0.783$, SDI =

0.0267). VideoCrafter-2 achieves moderately improved balance ($\text{RDS}_{\text{White}}$ = 0.6905, SDI = 0.1252), while T2V-Turbo-V1 further reduces White dominance ($\text{RDS}_{\text{White}}$ = 0.6381) but at the cost of lower diversity (SDI = 0.1119). Overall, while alignment tuning may alleviate certain ethnic skews, it can also suppress demographic diversity, suggesting a trade-off between bias reduction and representational variety. Figure 14 show the ethnicity bias across 42 verbs.

**Human Evaluation.** To ensure the reliability of our VLM-based evaluators, we conduct human verification across 400 generated videos annotated by *three* independent annotators for gender and ethnicity. All annotations were performed by three independent researchers with prior experience in social bias analysis. Annotators followed standardized written guidelines and received task-specific training grounded in established gender and ethnicity categorization conventions used in prior bias evaluation work. The annotation task was limited to verifying whether the generated content correctly reflected the prompted demographic attribute, rather than eliciting subjective or normative judgments. Our ensemble of three open-source VLMs—aggregated via majority voting—shows strong alignment with human judgments, achieving Pearson correlations of 0.89 (gender) and 0.73 (ethnicity). While the ethnicity correlation reflects residual disagreement, further analysis reveals that many discrepancies arise from differences between *human holistic judgment* and the *evaluator's frame-level precision*: fleeting but visually valid attributes (*e.g.*, faces visible for only a few frames in fast-motion clips) are sometimes missed or inferred by humans but explicitly captured by the VLM pipeline, indicating higher recall rather than systematic false positives. Agreement metrics further confirm high consistency: for video-level verification across the full pool of 400 videos, Cohen's Kappa reaches 0.91 (gender) and 0.78 (ethnicity), with Fleiss' Kappa of 0.92 and 0.82, respectively, indicating strong to near-perfect inter-annotator agreement. To examine whether evaluator disagreement affects downstream conclusions, we further conduct a *sensitivity analysis* on a high-agreement subset where human majority votes perfectly align with VLM predictions. We observe high consensus rates (99.0% for gender and 83.3% for ethnicity), and critically, all core bias metrics remain statistically stable when restricting to this subset (*e.g.*, $PBS_G$: 0.846 $\rightarrow$ 0.863, $p = 0.68$; SDI: 0.745 $\rightarrow$ 0.761). This *non-significant difference* confirms that residual evaluator uncertainty does not distort the reported bias measurements. Finally, we emphasize that human verification is used to ground and sanity-check the automated evaluation pipeline, rather than to replace large-scale evaluation. While larger-scale human annotation would be ideal, it is constrained by substantial labor costs and follows established practice in prior work that validates model-based judges using small-scale but high-consensus human review. Taken together, these results demonstrate that the VLM ensemble provides a reliable, scalable, and human-aligned evaluation mechanism for large-scale social attribute analysis.

These findings lead to our central research question: **how does alignment tuning shape the distribution of social attributes in video generative models?** To answer this, we (1) analyze demographic distributions embedded in the *image reward datasets* (§5.1), (2) examine the social biases in the trained *reward models* (§5.2), (3) assess how these biased reward models influence the representation of gender and ethnicity in video outputs when used for *alignment tuning* (§6).

## 5 Social Biases in Image Reward Datasets and Reward Models

Using image-based reward models has become the *de facto* and state-of-the-art approach for alignment tuning in video diffusion models (Wu et al., 2023; Xu et al., 2024; Li et al., 2024; Yuan et al., 2024; Liu et al., 2024a; Prabhudesai et al., 2024; Black et al., 2023; Ma et al., 2025). Because these reward models are trained on static images yet guide learning in temporally coherent video generation, their inherent social biases can propagate and even amplify across frames. Understanding such bias transfer from reward datasets to trained reward models is therefore crucial—not only to uncover demographic disparities embedded in human-labeled image preferences, but also to reveal how these biases shape the temporal evolution of social attributes in generated videos.

### 5.1 Preference in Image Reward Datasets

We analyze *two* widely used image reward datasets to investigate human preference biases: HPDv2 (Wu et al., 2023) and Pick-a-Pic (Kirstain et al., 2023b). For each dataset, we extract gender, ethnicity, and verb attributes from image captions using GPT-4o-mini, and classify attributes from images using three VLMs

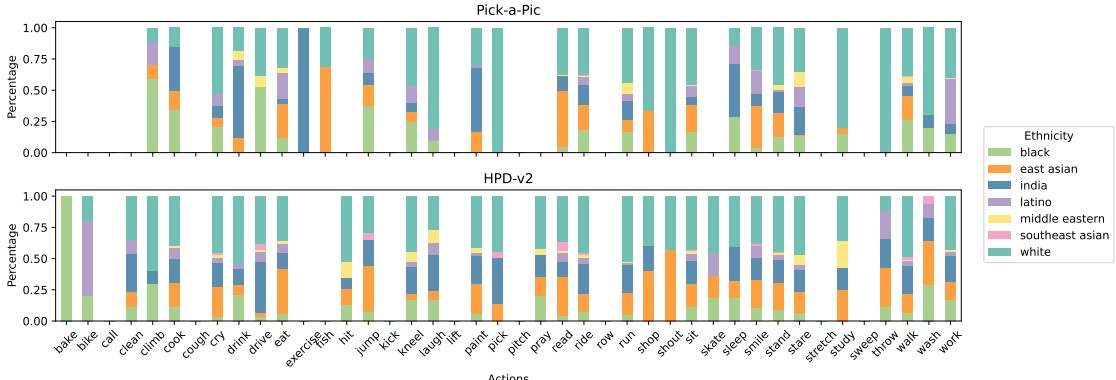

Figure 4: Image reward datasets ethnicity preference across 42 verbs.

(Qwen2-VL-7B, Qwen2.5-VL-7B, InternVL2.5-8B). We then aggregate the social attributes from both caption and image modalities, retaining only instances featuring one of our predefined verbs. After processing, HPDv2 contains 28,783 validated (images, caption, preference) tuples covering 29 verbs, and Pick-a-Pic contains 14,958 across 19 verbs. Each tuple presents two images, with a human annotator selecting the one that best matches the caption. To assess potential human preference biases, we measure how often annotators *prefer* specific gender or ethnicity representations for given verbs.

We directly analyze preference pairs where the preferred image depicts one gender (*e.g.*, a man) and the dispreferred image depicts the other (*e.g.*, a woman), thereby capturing explicit gender preference patterns within individual comparison pairs. Figure 3 shows the gender preference bias across 42 verbs in the two datasets. After filtering to ensure valid man–woman pairs, 26 out of 42 verbs in **HPDv2** met our criteria. Among these, 69.23% (18/26) showed a preference for men, revealing a consistent **man-preferred** tendency. In contrast, in **Pick-a-Pic**, 18 out of 42 verbs qualified, and 61.11% (11/18) showed a preference for women, indicating a relatively **woman-preferred** trend. Furthermore, Table 10 and Figure 4 present the ethnicity preference distributions across the two image reward datasets. Notably, both datasets exhibit a strong preference for the **White** group, 43.34% in HPDv2 and 40.08% in Pick-a-Pic, followed by East Asian and Indian representations. Despite certain verbs showing distinct preferences (*e.g.*, "bake" favoring Black individuals and "fish" favoring East Asians),

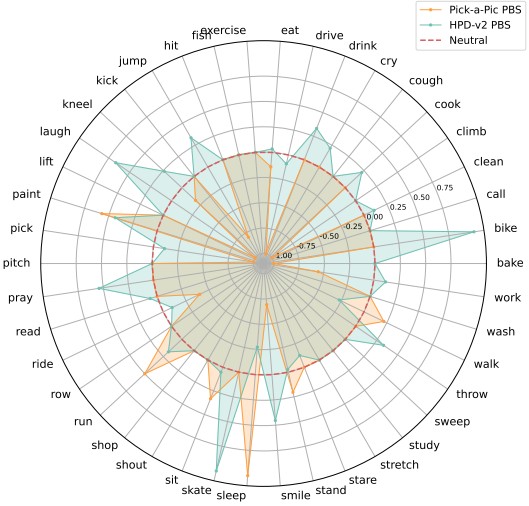

Figure 3: Image reward datasets gender preference distribution across 42 verbs.

the overall distributions reveal collected human preferences may implicitly favor Western-centric aesthetics and representation. These imbalances in human preferences might risk propagating representational bias during reward model training, thereby reinforcing existing social inequities in downstream video generation. Collectively, our findings underscore the urgent need for more inclusive and demographically representative preference datasets that capture global diversity.

## 5.2 Preference in Image Reward Models

Building on our analysis of gender and ethnicity biases in human preference datasets, we next examine how such biases propagate through *reward models* trained on these datasets.

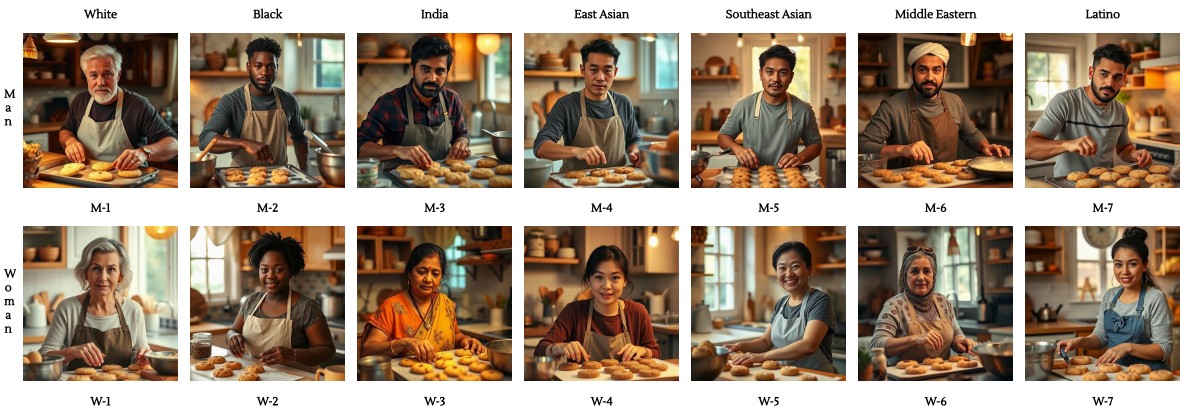

Figure 5: Image examples of our constructed benchmark for evaluating preference in image reward models with generation prompts: "A/An `[ethnicity][gender]` is baking `[context]`." We only show the images with `[gender]` ∈ {man, woman}.

**Evaluation Benchmark Construction.** To systematically evaluate social biases in reward models, we construct a controlled benchmark based on text-to-image (T2I) generation, inspired by HPDv2 (Wu et al., 2023) and ImageRewardDB (Xu et al., 2024). Using the event prompting templates introduced in Section 3.2, we employ FLUX (Labs, 2023), a state-of-the-art T2I model, to generate diverse image sets varying systematically across gender, ethnicity, and verb dimensions. The benchmark includes *two* generation settings: (1) `Ethnicity+Person`, where prompts specify only the actor's ethnicity, and (2) `Ethnicity+Gender`, where both gender and ethnicity are explicitly indicated. Table 3 summarizes prompt coverage and provides representative examples. To ensure statistical robustness, we generate 100 images per prompt, resulting in a large and diverse evaluation set. Sample outputs are illustrated in Figure 5.

The use of FLUX is not intended to assume a bias-free image generator, but rather to provide a **controlled and reproducible setting** for probing preference patterns in reward models. By conditioning on systematically varied demographic attributes, our benchmark isolates reward model preferences independently of generator-specific artifacts. To assess benchmark integrity, we conduct a human verification study in which three annotators independently reviewed 100 randomly sampled images. While 77% of samples achieved unanimous agreement, this figure is intentionally conservative: a deeper inspection shows that 97% of samples reached majority agreement (at least 2/3 annotators confirming correct attribute representation), with only three images lacking consensus. Importantly, these rare disagreements were confined to visually ambiguous prompts rather than systematic failures or demographic-specific distortions. This high signal-to-noise ratio indicates that the visual gender and ethnicity cues are sufficiently clear for reward models to process reliably, and that residual label noise is unlikely to materially affect the observed preference patterns.

| Settings | # of Prompts | Examples |
|---|---|---|
| Ethnicity + Person | 294 | An *East Asian* person is **baking** cookies in a cozy kitchen, with warm light and the aroma of vanilla filling the air. |
| Ethnicity + Gender | 1176 | An *East Asian* woman is **baking** cookies in a cozy kitchen, with warm light and the aroma of vanilla filling the air. |

Table 3: Statistics of Benchmark Construction Prompts for Image Reward Models. Each prompt explicitly highlights the actor's *ethnicity* (when specified), the **verbs**, and the surrounding context, providing a structured basis for analyzing social representations on image reward models.

**Preference Bias Evaluation.** We evaluate *four* image reward models: (1) HPSv2.0 (Wu et al., 2023), trained on the HPDv2 dataset; (2) HPSv2.1 (Wu et al., 2023), trained on the unreleased HPDv2.1 dataset; (3) PickScore (Kirstain et al., 2023b), trained on the Pick-a-Pic dataset; and (4) CLIP (Radford et al., 2021), which serves as the shared base model for HPSv2.0, HPSv2.1, and PickScore prior to fine-tuning on their respective preference datasets. Each generated sample is assigned a *continuous scalar preference score* predicted by the corresponding reward model (*e.g.*, the HPSv2 score), rather than a discrete class

label. These standardized scores are used to quantify the model's *relative preference* for gendered and ethnic attributes across controlled comparisons. Table 4 reports two complementary bias metrics—ethnicity-aware gender bias ($PBS_G$) and ethnic representation distribution ($RDS_e$ and SDI). Appendix C provides further implementation details, a comprehensive analysis across 42 verbs, and a statistical analysis (Table 11).

| Models | Average | White | | Black | | Latino | | East Asian | | Southeast Asian | | India | | Middle Eastern | | Overall |
|---|---|---|---|---|---|---|---|---|---|---|---|---|---|---|---|---|
| | $PBS_G$ | $PBS_G$ | RDS | $PBS_G$ | RDS | $PBS_G$ | RDS | $PBS_G$ | RDS | $PBS_G$ | RDS | $PBS_G$ | RDS | $PBS_G$ | RDS | SDI |
| CLIP | -0.0726 | 0.0343 | **0.0182** | -0.1198 | 0.0002 | -0.0934 | -0.0013 | -0.1315 | 0.0141 | -0.0865 | 0.0094 | -0.0508 | -0.0299 | -0.0607 | -0.0108 | **0.8495** |
| HPSv2.0 | 0.6039 | 0.6090 | -0.0423 | 0.7341 | -0.0069 | 0.6512 | 0.0237 | 0.4752 | -0.0031 | 0.5192 | -0.0100 | 0.5922 | 0.0070 | 0.6464 | **0.0315** | 0.8492 |
| Δ | +0.6765 | +0.5747 | -0.0605 | +0.8539 | -0.0071 | +0.7446 | +0.0250 | +0.6067 | -0.0172 | +0.6057 | -0.0194 | +0.6430 | +0.0369 | +0.7071 | +0.0423 | -0.0003 |
| HPSv2.1 | -0.0984 | -0.0833 | -0.0189 | 0.0257 | -0.0321 | -0.0031 | **0.0382** | -0.3044 | 0.0091 | -0.2181 | -0.0099 | -0.0006 | -0.0077 | -0.1053 | 0.0214 | 0.8470 |
| Δ | -0.0258 | -0.1176 | -0.0371 | +0.1455 | -0.0323 | +0.0903 | +0.0395 | -0.1729 | -0.0050 | -0.1316 | -0.0193 | +0.0502 | +0.0222 | -0.0446 | +0.0322 | -0.0025 |
| PickScore | -0.1157 | 0.0321 | 0.0069 | -0.0777 | 0.0279 | -0.3479 | -0.0118 | -0.2257 | **0.0316** | -0.2163 | 0.0115 | 0.1531 | -0.0391 | -0.1277 | -0.0271 | 0.8483 |
| Δ | -0.0431 | -0.0022 | -0.0113 | +0.0421 | +0.0277 | -0.2545 | -0.0105 | -0.0942 | +0.0175 | -0.1298 | +0.0021 | +0.2039 | -0.0092 | -0.0670 | -0.0163 | -0.0012 |

Table 4: Preference bias of different reward models. All values represent *average* scores across 42 verbs.

**Ethnicity-Aware Gender Bias.** We construct preference evaluation prompts in the format "A/An [`ethnicity`] person is [`verb`]-ing [`context`]", covering all combinations of ethnicity and verb (evaluation: `ethnicity`+`person`). For each preference prompt, we generate images using generation prompts in the format "A/An [`ethnicity`] [`gender`] is [`verb`]-ing [`context`]", where gender, ethnicity, and verb are explicitly specified (generation: `ethnicity`+`gender`). The evaluation prompt omits gender to measure reward model's inherent preference, while the generation prompt explicitly specifies gender. The reward scores assigned to these images by a reward model are standardized using their mean and standard deviation. We then compute the average standardized score across the 100 images for each generation prompt. To compute the final $PBS_G$, we fix the ethnicity and verb, and subtract the average standardized score for women from that for men. Because of the adaptation, $PBS_G$ score here can be greater than one. A positive $PBS_G$ score indicates a preference for men, while a negative score reflects a preference for women. CLIP shows a slight woman-preference bias (–0.0726). Fine-tuning on HPDv2 shifts HPSv2.0 toward a *strong* man-preference (+0.6039), consistent across ethnic groups. In contrast, PickScore (–0.1157) and HPSv2.1 (–0.0984) show woman-preference biases, with the latter's training data undisclosed. These shifts align with each model's training data, revealing consistent gender preferences across ethnicities. Figures 19 to 25 presents the $PBS_G$ scores across 42 verbs for each ethnicity group.

**Ethnicity Bias.** We use preference evaluation prompts in the form "A person is [`verb`]-ing [`context`]" (evaluation: `person-only`). For each preference prompt, we have generated images using more specific generation prompts of the form "A/An [`ethnicity`] person is [`verb`]-ing [`context`]", where the ethnicity and verb are explicitly specified (generation: `ethnicity`+`person`). The evaluation prompt omits ethnicity to measure reward model's inherent preference, while the generation prompt explicitly specifies ethnicity. The reward scores for these images provided by a reward model are standardized with mean and standard deviation. We then compute the average standardized score across the 100 images for each generation prompt. To calculate $RDS_e$ and SDI, we fix the verb and apply softmax function (Bridle, 1990; Bishop, 2006) to normalize the scores for each ethnicity, indicating ethnicity preference within each verb context. A positive $RDS_e$ indicates overrepresentation of an ethnicity, while a negative score indicates underrepresentation. A higher SDI score corresponds to more balanced and diverse outputs across all groups. The base model, CLIP, slightly favors White individuals (RDS = 0.0182) and achieves the highest SDI score (0.8495), indicating relatively balanced ethnic representation. After fine-tuning, HPSv2.0 shifts toward Middle Eastern (RDS = 0.0315), HPSv2.1 toward Latino (RDS = 0.0382), and PickScore toward East Asian individuals (RDS = 0.0352). All show reduced SDI, indicating decreased ethnic diversity post-alignment. Figures 26 to 29 show the ethnicity bias across 42 verbs.

# 6 Social Biases in Preference Alignment

Building on our analysis of gender and ethnicity biases in image reward models, we examine how preference alignment tuning affects bias in video generation. We fine-tune a Video Consistency Model distilled from VideoCrafter-V2 (VCM-VC2) (Li et al., 2024) using *three* image reward models, HPSv2.0, HPSv2.1, and

PickScore, and compare social bias distributions before and after tuning to assess how each reward model shapes identity representation. Following the T2V-Turbo-V1 training protocol (Li et al., 2024), we incorporate reward feedback into the state-of-the-art paradigm–Latent Consistency Distillation process (Luo et al., 2023) by using single step video generation. During student model distillation from a pretrained teacher text to video model, we directly optimize the decoded video frames to maximize reward scores from the image-text alignment models, guiding each frame toward representations more aligned with human preferences. Video model post-training and inference details can be found in Appendix G.

We evaluate aligned video diffusion models using our evaluation framework (§4). Table 5 reports two metrics: $PBS_G$ for gender imbalance across ethnic groups, and $RDS_e$ and SDI for ethnicity representation disparity and overall output diversity. Appendix D includes more analysis across 42 verbs and a statistical analysis (Table 12).

| Models | Average | White | | Black | | Latino | | East Asian | | Southeast Asian | | India | | Middle Eastern | | Overall |
|---|---|---|---|---|---|---|---|---|---|---|---|---|---|---|---|---|
| | $PBS_G$ | $PBS_G$ | RDS | $PBS_G$ | RDS | $PBS_G$ | RDS | $PBS_G$ | RDS | $PBS_G$ | RDS | $PBS_G$ | RDS | $PBS_G$ | RDS | SDI |
| VCM-VC2 | 0.8034 | 0.7925 | **0.6405** | 0.7758 | -0.2381 | 0.8090 | — | 0.7115 | -0.2333 | 0.7945 | — | 0.8634 | — | 0.8071 | -0.1690 | 0.1433 |
| + HPSv2.0 | 0.9116 | 0.9667 | 0.3667 | 0.9214 | — | 0.9667 | — | 0.8500 | — | 0.9214 | — | 0.9000 | — | 0.8548 | -0.3667 | 0.1257 |
| Δ | +0.1082 | +0.1742 | -0.2738 | +0.1456 | — | +0.1577 | — | +0.1385 | — | +0.1269 | — | +0.0366 | — | +0.0477 | -0.1977 | -0.0176 |
| + HPSv2.1 | 0.2267 | 0.1321 | **0.4286** | 0.2381 | — | 0.3738 | — | 0.1571 | — | 0.3619 | — | 0.1452 | — | 0.1786 | -0.4286 | 0.0976 |
| Δ | -0.5767 | -0.6604 | -0.2119 | -0.5377 | — | -0.4352 | — | -0.5544 | — | -0.4326 | — | -0.7182 | — | -0.6285 | -0.2596 | -0.0457 |
| + PickScore | 0.3714 | 0.3429 | **0.6833** | 0.3357 | -0.1810 | 0.7190 | -0.1929 | 0.1450 | -0.1952 | 0.4548 | — | 0.2500 | — | 0.3515 | -0.1143 | 0.1467 |
| Δ | -0.4320 | -0.4496 | +0.0428 | -0.4401 | +0.0571 | -0.0900 | -0.1929 | -0.5665 | +0.0381 | -0.3397 | — | -0.6134 | — | -0.4556 | +0.0547 | +0.0034 |

Table 5: Social biases of aligned models. All values represent *average* scores across 42 verbs. Missing values are marked as "—" and indicate cases where the model generated zero recognizable instances of the corresponding ethnicity under the `person-only` condition (*i.e.*, when ethnicity is not explicitly specified), reflecting a severe lack of diversity that prevents metric computation for those subgroups.

**Ethnicity-Aware Gender Bias.** We evaluate gender portrayals under the `ethnicity+person` condition using the previously defined $PBS_G$ metric. A positive $PBS_G$ score indicates a tendency to depict men more frequently, while a negative score suggests a preference for women. The base model, VCM-VC2, demonstrates a strong man bias across all ethnicities, which becomes more pronounced with alignment using HPSv2.0. In contrast, alignment with HPSv2.1 and PickScore significantly reduces $PBS_G$, indicating a shift toward more balanced or woman-preferred outputs. This change reflects the underlying woman bias present in the HPSv2.1 and PickScore reward models, which steer the model away from the man-dominant bias of the base model. Figures 30 to 37 presents the $PBS_G$ scores across 42 verbs for each ethnicity group.

**Ethnicity Bias.** Under the `ethnicity-only` condition, we analyze models' representation balance using the previously defined $RDS_e$ and SDI metrics. Positive values indicate overrepresentation, and negative values indicate underrepresentation. Overall demographic balance is measured using SDI, where higher values reflect more equitable representation. The base model, VCM-VC2, strongly favors White individuals (RDS = 0.6405), while Black, East Asian, and Middle Eastern groups are underrepresented. Alignment with HPSv2.1 reduces some disparities by improving balance for White and Black groups, but significantly decreases Latino representation (RDS = –0.4352) and lowers SDI, indicating reduced diversity. In contrast, PickScore achieves the highest SDI and produces more balanced representation across most ethnic groups, resulting in the most demographically equitable outputs. Figure 38 shows the ethnicity bias across 42 verbs.

**Temporal Attribute Stability.** Table 6 summarizes the temporal consistency of identity portrayals across alignment-tuned models. Overall, alignment substantially improves the *technical coherence* of video generation. For ethnicity, the mean TAS increases from 0.8904 in the VCM-VC2 baseline to 0.9632 after HPSv2.1 alignment, and the proportion of videos achieving perfect stability rises by 0.2660 (from 0.5670 to 0.8330). Similar gains appear for gender, with stability reaching near saturation at 0.9895 and 0.9440 perfect stability. The standard deviation of TAS also consistently decreases (*e.g.*, –0.0654 for ethnicity), indicating *more uniform frame-level consistency across videos*. However, this improvement in stability comes with an important caveat. When alignment models inherit biased preferences, the resulting stability can entrench rather than mitigate bias. For instance, HPSv2.0 alignment not only amplifies the overall man-preference bias ($PBS_G$ rising from 0.8034 to 0.9116) but also locks that bias in temporally—with gender stability reaching

| Model | Attribute | Mean TAS | Median TAS | Std TAS | Perfect Stability |
|---|---|---|---|---|---|
| VCM-VC2 | Ethnicity | 0.8904 | 1.0000 | 0.1673 | 0.5670 |
| VCM-VC2 | Gender | 0.9763 | 1.0000 | 0.0829 | 0.8850 |
| + HPSv2 | Ethnicity | $0.9305_{+0.0401}$ | 1.0000 | $0.1358_{-0.0315}$ | $0.7050_{+0.1380}$ |
| + HPSv2 | Gender | $0.9967_{+0.0204}$ | 1.0000 | $0.0310_{-0.0519}$ | $0.9830_{+0.0980}$ |
| + HPSv2.1 | Ethnicity | $0.9632_{+0.0728}$ | 1.0000 | $0.1019_{-0.0654}$ | $0.8330_{+0.2660}$ |
| + HPSv2.1 | Gender | $0.9895_{+0.0132}$ | 1.0000 | $0.0519_{-0.0310}$ | $0.9440_{+0.0590}$ |
| + PickScore | Ethnicity | $0.9406_{+0.0502}$ | 1.0000 | $0.1302_{-0.0371}$ | $0.7560_{+0.1890}$ |
| + PickScore | Gender | $0.9884_{+0.0121}$ | 1.0000 | $0.0553_{-0.0276}$ | $0.9410_{+0.0560}$ |

Table 6: Temporal Attribute Stability (TAS) across models and attributes. A high TAS score indicates the actor's identity representation is stable and consistent throughout the video. A low TAS score indicates the representation "flickers" or changes, a key temporal artifact. Subscripts in red and blue indicate relative improvements and degradations compared to the base model VCM-VC2.

0.9967 and nearly all videos (0.9830) achieving perfect stability. In other words, the model becomes *better at being biased*: it produces smoother, more coherent, yet more stereotyped portrayals. This finding highlights a critical insight revealed uniquely by our temporal evaluation framework: alignment tuning, while improving the perceptual and temporal quality of generation, can inadvertently make social bias more persistent and deeply embedded in the generative process. VIDEOBIASEVAL thus not only detects whether a model is biased but also exposes how alignment can make such bias temporally resilient—transforming representational artifacts into stable, systematic distortions.

**Summary and Implications.** Taken together, these findings demonstrate that preference alignment can both mitigate and amplify existing social biases, depending on the characteristics of the guiding reward model. While HPSv2.0 reinforces the base model's male- and White-preferred tendencies, HPSv2.1 and PickScore—both exhibiting woman-preferred reward patterns—successfully counteract the original man-dominant bias, steering the aligned model toward more balanced portrayals. However, neither alignment achieves complete demographic parity, as residual disparities across ethnic groups persist. These observations suggest that the direction and magnitude of social bias in aligned video diffusion models are largely inherited from the bias profile of their reward models.

# 7 Controllable Preference Modeling for Video Diffusion Models

Building on prior findings, we next examine whether such biases can be made *controllable*. Specifically, we investigate if adjusting the distribution of social attributes in image preference datasets can systematically steer reward models—and consequently video diffusion models—toward more equitable or intentionally calibrated portrayals (Sheng et al., 2020). To make this concrete, we take **gender** as a case study, examining how varying gender composition in preference data influences the directional bias and alignment dynamics of the resulting reward and video models.

## 7.1 Image Reward Dataset Construction

We construct two reward datasets: a man-preferred version and a woman-preferred version, using images from §5.1 to guide diffusion models toward gender-specific representations. Each dataset includes 2.94 million preference pairs from the `Ethnicity+Gender` set, where each pair depicts the same action and ethnicity but differs by gender (*e.g.*, M-1 vs. W-1 in Figure 5). Prompts follow the format "A/An `[ethnicity]` person is `[action]`-ing `[context]`." In the man-preferred dataset, male images are labeled 1 and female images 0; the opposite applies in the woman-preferred dataset. To enhance face-free diversity, we also include 537,660 additional image pairs from HPDv2. When applied to a base model with man-preference bias, the woman-preferred dataset helps correct this imbalance and promotes more equitable gender representation.

## 7.2 Image Reward Model Development & Alignment Tuning

Leveraging the man-preferred and woman-preferred image datasets, we fine-tune two reward models on top of a pre-trained CLIP vision encoder: the Man-Preferred Reward Model ($RM_M$) and the Woman-Preferred Reward Model ($RM_W$). Each is optimized to reflect gender-specific aesthetic and representational preferences encoded in its respective dataset. As shown in Table 7, $RM_M$ consistently assigns higher $PBS_G$ scores across all demographic groups, aligning with man-preferred portrayals, whereas $RM_W$ exhibits the opposite tendency, systematically favoring woman-preferred content. This clear divergence demonstrates that reward tuning effectively captures and amplifies gendered preferences. Reward model training and inference details can be found in Appendix F.

| Models | Average | White | Black | Latino | East Asian | Southeast Asian | India | Middle Eastern |
|---|---|---|---|---|---|---|---|---|
| CLIP | -0.0726 | 0.0343 | -0.1198 | -0.0934 | -0.1315 | -0.0865 | -0.0508 | -0.0607 |
| $RM_M$ | $1.5280_{+1.60}$ | $1.6300_{+1.60}$ | $1.5752_{+1.70}$ | $1.5524_{+1.65}$ | $1.4323_{+1.56}$ | $1.4525_{+1.54}$ | $1.5619_{+1.61}$ | $1.4914_{+1.55}$ |
| $RM_W$ | $-0.7448_{-2.27}$ | $-0.6318_{-2.26}$ | $-0.7943_{-2.37}$ | $-0.8279_{-2.38}$ | $-0.6282_{-2.06}$ | $-0.6429_{-2.10}$ | $-0.8846_{-2.45}$ | $-0.8042_{-2.30}$ |

Table 7: Preference bias of reward models. All values represent *average* scores across 42 verbs.

We intentionally do not train a "fair" or demographically neutral reward model. As discussed in §6, the base video generator (*e.g.*, VCM-VC2) already exhibits a pronounced man bias. Under such asymmetric initialization, a neutral reward signal would merely reinforce existing imbalances rather than correct them. To counteract this skew, we employ a deliberately counter-biased reward model—specifically, the woman-preferred $RM_W$, analogous to HPSv2.1—which actively steers generation toward gender equilibrium. This targeted alignment strategy yields markedly more balanced portrayals, demonstrating that directional reward tuning can serve as an effective corrective mechanism for bias mitigation.

Building upon these reward models, we further apply $RM_M$ and $RM_W$ to guide preference alignment of the base video diffusion model (VCM-VC2). Using the same preference-optimization framework, we obtain two aligned variants: one tuned toward man-preferred and the other toward woman-preferred content. As shown in Table 8, alignment with $RM_M$ increases $PBS_G$ scores across all demographic groups, reinforcing man-preference bias, whereas alignment with $RM_W$ substantially reduces these scores, indicating a strong shift toward woman-preference bias. These results confirm that our controllable preference modeling framework enables fine-grained modulation of gender bias in video generation, offering a principled and flexible means to amplify or mitigate social tendencies in diffusion-based models. Moreover, achieving the most balanced aligned video generator can be framed as a data composition problem: by systematically adjusting the proportion of man- and woman-preferred samples in the reward dataset, one can identify an optimal mixture that minimizes overall bias while preserving aesthetic fidelity. This insight highlights a promising direction for dataset-driven bias control in alignment tuning. Figures 39 to 43 presents the $PBS_G$ scores across 42 actions for each ethnicity group.

| Models | Average | White | Black | Latino | East Asian | Southeast Asian | India | Middle Eastern |
|---|---|---|---|---|---|---|---|---|
| VCM-VC2 | 0.8034 | 0.7925 | 0.7758 | 0.8690 | 0.7115 | 0.7945 | 0.8634 | 0.8071 |
| + $RM_M$ | $0.9584_{+0.16}$ | $0.9595_{+0.17}$ | $0.9524_{+0.18}$ | $0.9756_{+0.11}$ | $0.9437_{+0.23}$ | $0.9447_{+0.15}$ | $0.9640_{+0.10}$ | $0.9709_{+0.16}$ |
| + $RM_W$ | $0.3082_{-0.50}$ | $0.3341_{-0.46}$ | $0.3913_{-0.38}$ | $0.3314_{-0.54}$ | $0.1008_{-0.61}$ | $0.2639_{-0.53}$ | $0.3446_{-0.52}$ | $0.3894_{-0.42}$ |

Table 8: Social biases of aligned models. All values represent *average* scores across 42 verbs.

**Summary and Implications.** Our controllable preference modeling framework demonstrates that bias in video diffusion models can be systematically directed through the construction and tuning of social-attribute-aware reward datasets. By varying gender composition in preference data, we show that reward models learn and transmit directional biases—reinforcing or counteracting existing tendencies in base generators. Importantly, introducing counter-biased reward models (*e.g.*, woman-preferred $RM_W$) can actively correct skewed portrayals, producing more balanced and socially representative generations. Beyond binary control, these results suggest a continuous axis of alignment, where nuanced mixtures of man- and woman-preferred data yield tunable trade-offs between bias mitigation and aesthetic coherence. The sensitivity analysis across

42 actions further reveals that even semantically neutral activities can exhibit large gender shifts under alignment, underscoring the need for fine-grained, event-level auditing. Overall, our findings highlight that dataset composition, rather than architectural intervention alone, offers a powerful and interpretable lever for steering the social behavior of generative models.

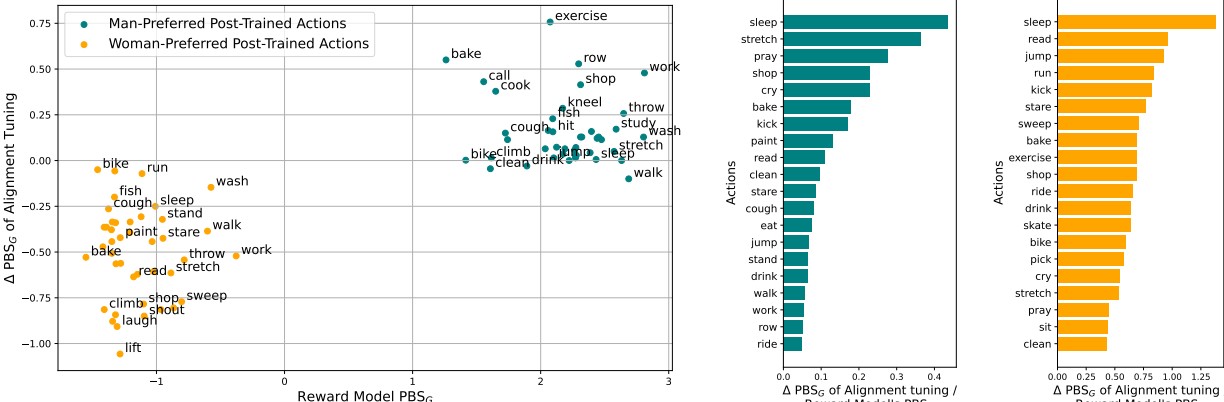

(a) $\Delta$ PBS$_G$ of video generation model before and after alignment tuning by RM$_M$ and RM$_W$. Results are broken down into actions. Figure 6b and Figure 6c are based on this figure.

(b) Sensitive actions in man-preferred alignment tuning.

(c) Sensitive actions in woman-preferred alignment tuning.

Figure 6: Action-level impact of alignment tuning guided by RM$_M$ and RM$_W$.

**Which Actions Are Most Sensitive During Alignment Tuning?** We investigate how specific actions respond to gender-oriented reward model tuning by measuring changes in PBS$_G$ scores before and after alignment. As shown in Figure 6a, actions cluster distinctly by the reward model that guides tuning: alignment with RM$_M$ (teal) amplifies male-preferred portrayals in activities such as *exercise*, *row*, and *cook*, while alignment with RM$_W$ (orange) shifts representations toward women-preferred actions like *bake*, *sleep*, and *sweep*. Figure 6b and Figure 6c further quantify these shifts by ranking actions according to their normalized sensitivity (*i.e.*, $\Delta$PBS$_G$ divided by each reward model's baseline PBS$_G$). The top-ranked actions—*sleep*, *stretch*, and *read*—emerge as the most sensitive under both man- and woman-preferred tuning, revealing that even socially neutral or domestic activities can exhibit pronounced gender bias once alignment is applied. Together, these results highlight that alignment tuning induces systematic, action-specific bias amplification, and demonstrate the effectiveness of our event-centric evaluation framework in exposing fine-grained behavioral sensitivities across gendered dimensions.

**Responsible Use of Controllable Preference Modeling.** While our results demonstrate that controllable preference modeling can effectively steer social attribute distributions, we emphasize that technical "debiasing" is not a silver bullet and carries significant risks if applied unilaterally. This technique should be applied thoughtfully as a tool to support, rather than replace, diverse stakeholder input. Defining "equitable representation" is fundamentally a context-dependent normative decision that requires active community consultation. We encourage practitioners to engage with the communities impacted by these systems to determine appropriate representation goals, rather than relying solely on automated optimization to resolve complex social biases.

## 8 Conclusion

In summary, our work exposes and investigates key blind spots in evaluating social bias within text-to-video generation. Through the introduction of VIDEOBIASEVAL, we establish a comprehensive framework that decouples identity attributes from content semantics and systematically tracks how alignment tuning reshapes social representations. Our analyses demonstrate that reward-model-based alignment not only inherits but frequently amplifies existing biases encoded in human preference data. These findings highlight the importance

of integrating bias auditing and mitigation throughout every stage of the video generation pipeline, advancing the development of more equitable and socially aware generative systems.

## Limitations

While our work presents a comprehensive evaluation of social biases introduced through alignment tuning in video diffusion models, several limitations warrant further consideration. *First*, our analysis focuses on two social dimensions, gender and ethnicity, using predefined categories based on U.S. Census conventions and prior literature. These categories, while practical for controlled evaluation, are inherently socially constructed and cannot fully capture the fluidity, intersectionality, or cultural nuances of identity Yin et al. (2024); Qiu et al. (2024a; 2025). Future work should explore richer identity representations, including intersectional groups. *Second*, our VLM-based evaluators, though validated against human judgments, rely on image-level classification and may exhibit their own biases or inaccuracies, particularly when interpreting identity in stylized or ambiguous frames. While we ensemble multiple models to mitigate this, ground truth annotations for a larger and more diverse set of videos would further strengthen the reliability of our measurements. *Third*, we primarily assess alignment impacts under a specific training strategy (single-step latent consistency distillation) and a limited set of reward models. Other training protocols, such as RL-based tuning or multi-turn video instruction alignment, may exhibit different bias dynamics not captured in our study. *Fourth*, our controllable preference modeling experiments, while demonstrating the feasibility of targeted bias modulation, are constrained to synthetic manipulations of gender preference. These interventions do not address broader questions of value alignment, normative appropriateness, or long-term societal impact, which are crucial for the responsible deployment of generative video systems. *Lastly*, our evaluation framework, VIDEOBIASEVAL, is currently benchmarked on a fixed set of 42 socially associated actions. While this enables fine-grained control, it may limit generalizability to open-ended generation settings or novel actions not covered in our taxonomy. We hope that these limitations encourage further research into holistic, culturally grounded, and ethically aligned evaluation pipelines for video generative models.

## Broader Impact Statement

As video diffusion models transition from research prototypes to widely deployed tools for content creation, their influence on visual culture and societal representation becomes increasingly significant. Our work highlights a critical but often overlooked consequence of this adoption: the role of alignment tuning in stabilizing and entrenching social biases.

**Stabilization of Bias.** While alignment tuning (*e.g.*, via RLHF) is essential for improving the visual fidelity and temporal consistency of generated videos, our findings reveal that it can inadvertently act as a "bias stabilizer." By removing visual artifacts and smoothing temporal flickers, alignment tuning makes stereotyped representations appear more natural, coherent, and authoritative. Unlike the obvious errors in unaligned models, these "polished" biases are harder to detect as failures, risking the subtle normalization of skewed demographic portrayals in synthetic media. If left unchecked, this stabilization could reinforce harmful societal stereotypes under the guise of high-quality, preferred content.

**Dual-Use of Controllable Preference Modeling.** Furthermore, our exploration of controllable preference modeling in §7 underscores the dual-use nature of these techniques. We demonstrate that by curating preference datasets, reward models can be engineered to systematically amplify or suppress specific social attributes. While this capability offers a powerful mechanism for correcting historical underrepresentation and mitigating bias, it fundamentally renders the alignment process a double-edged sword. The same tools used to promote fairness could be exploited to intentionally exclude specific groups, enforce homogenization, or generate propaganda that aligns with exclusionary worldviews.

**Mitigation and Responsibility.** These risks necessitate a shift in how we evaluate generative video systems. Relying solely on aesthetic metrics is insufficient; developers must adopt multi-granular, event-centric auditing frameworks—like VIDEOBIASEVAL—to continually monitor the social composition of generated content. We urge the research community and practitioners to treat preference dataset curation with the same rigor as

model architecture design, ensuring that the "human preferences" guiding these systems reflect a diverse and equitable society rather than reinforcing existing prejudices.

## Acknowledgment

We thank the anonymous reviewers for their thoughtful feedback and suggestions. This research was supported in part by the AFOSR MURI grant #FA9550-22-1-0380, the Office of Naval Research (ONR) under Award #N00014-23-1-2780, and the National Science Foundation under Award #IIS-2449768. Additional support was provided through cloud credits from the NVIDIA Academic Grant Program and funding from the Technical AI Safety Research Program at Coefficient Giving. The views and conclusions expressed in this work are those of the authors and should not be interpreted as representing the official policies or endorsements of AFOSR, ONR, the U.S. Government, the National Science Foundation, NVIDIA, or Coefficient Giving.

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

# A  Social Biases in Video Generative Models

We apply our proposed evaluation framework to *four* state-of-the-art video diffusion models with varying alignment strategies. In this work, we use the term **aligned** models to refer to video diffusion models that undergo post-training optimization to better match human preferences, typically via reward-weighted fine-tuning or RLHF-style procedures using learned preference or aesthetic reward models. Concretely, this alignment process follows a general pipeline of base video diffusion model → preference/reward model → post-training optimization. Under this definition, the **aligned** models include InstructVideo (Yuan et al., 2024), which builds on ModelScope (Wang et al., 2023a) and is aligned using HPSv2.0, as well as T2V-Turbo-V1 (Li et al., 2024), which builds on VideoCrafter-2 (Chen et al., 2024a) and is aligned with HPSv2.1, InternVid2-S2 (Wang et al., 2024b), and ViCLIP (Wang et al., 2023b). Their **unaligned** counterparts, ModelScope and VideoCrafter-2, serve as baselines for controlled comparisons, allowing us to isolate the effects of alignment from those of base model architecture.

For implementation, we use the official code repositories provided by the respective papers and run inference on 1 to 8 NVIDIA A100 80GB GPUs. To compute the social bias distribution, as outlined in §3, we generate videos with each prompt 10 times per model with different random seeds and average the results to reduce sampling variance. Table 2 reports two social bias metrics: ethnicity-aware gender bias ($PBS_G$) and ethnic representation distribution ($RDS_e$ and SDI).

**Ethnicity-Aware Gender Bias.** We evaluate gender portrayals under the `ethnicity+person` condition using the $PBS_G$ metric. Our analysis reveals a consistent male bias, with average $PBS_G$ values remaining above zero across all ethnic groups. Notably, *alignment tuning appears to amplify this imbalance.* InstructVideo and T2V-Turbo-V1 exhibit $PBS_G$ increases of 0.04 and 0.0725, respectively, indicating that preference-based fine-tuning may worsen gender disparity rather than alleviate it. Figures 7 to 13 present the detailed $PBS_G$ scores across 42 verbs for each ethnicity.

**Ethnicity Bias.** Under the `person-only` condition, we analyze models' representation balance using the previously defined $RDS_e$ and SDI metrics. All models show a pronounced overrepresentation of White individuals, though the magnitude varies. ModelScope exhibits the strongest imbalance ($RDS_{White} = 0.769$, $SDI = 0.0538$), which is further amplified by alignment tuning in InstructVideo ($RDS_{White} = 0.783$, $SDI = 0.0267$). VideoCrafter-2 achieves moderately improved balance ($RDS_{White} = 0.6905$, $SDI = 0.1252$), while T2V-Turbo-V1 further reduces White dominance ($RDS_{White} = 0.6381$) but at the cost of lower diversity ($SDI = 0.1119$). Overall, while alignment tuning may alleviate certain ethnic skews, it can also suppress demographic diversity, suggesting a trade-off between bias reduction and representational variety. Figure 14 show the ethnicity bias across 42 verbs.

**How Does the WebVid-10M Distribution Shape Baseline Model Behavior?** To interpret the source of baseline model behaviors, we analyzed the demographic composition of the WebVid-10M (Bain et al., 2021) training dataset. By filtering the dataset for our 42 target verbs, we extracted a subset of 469,153 videos containing gender attributes and 67,081 videos containing ethnicity attributes. Our analysis reveals a distinct misalignment regarding gender: while the training data exhibits a slight female skew (mean $PBS_G$ of -0.110), the unaligned base models demonstrate a significant male bias ($PBS_G$ of +0.4815 for ModelScope and +0.7581 for Video-Crafter-V2). This indicates that the strong "Male Default" in baseline models does not stem from the WebVid-10M verb distribution, but likely originates from biases inherent in the underlying Text-to-Image backbone during pre-training. Conversely, ethnicity biases in the base models mirror the training data's distribution—which skews towards White ($RDS_{White}$ of +0.2489) and Southeast Asian identities—but exaggerate the magnitude. These findings clarify that while pre-training establishes a biased baseline (sometimes contradictory to the video training data), our study specifically isolates how subsequent alignment tuning propagates or amplifies these pre-existing states. Figure 15 and Figure 16 visualize these distributions across all 42 verbs.

**Statistical Analysis of T2V Models.** We perform paired statistical significance tests on alignment-induced differences ($\Delta$) between each base model and its aligned counterpart, with results summarized in Table 9. For each metric, we compute $\Delta = \mu_{aligned} - \mu_{base}$, where positive values indicate an increase after alignment. Statistical reliability is assessed using the $p$-value, which measures the probability of observing a difference

| Metric | ModelScope → InstructVideo | | | | | | | VideoCrafter-V2 → T2V-Turbo-V1 | | | | | | |
|---|---|---|---|---|---|---|---|---|---|---|---|---|---|---|
| | Base | Aligned | Δ | p-value | d | Eff. Size | Sig. | Base | Aligned | Δ | p-value | d | Eff. Size | Sig. |
| *Ethnicity-Aware Gender Bias: Proportion Bias Score for Gender ($PBS_G$)* | | | | | | | | | | | | | | |
| Average | 0.481 | 0.529 | +0.048 | 0.1079 | -0.254 | Small-Med | No | 0.758 | 0.831 | +0.072 | 0.0089 | -0.424 | Small-Med | Yes** |
| White | 0.568 | 0.558 | -0.010 | 0.8261 | 0.034 | Small | No | 0.749 | 0.871 | +0.123 | 0.0027 | -0.493 | Small-Med | Yes** |
| Black | 0.391 | 0.511 | +0.120 | 0.0030 | -0.487 | Small-Med | Yes** | 0.617 | 0.809 | +0.193 | 0.0001 | -0.679 | Med-Large | Yes*** |
| Indian | 0.394 | 0.488 | +0.094 | 0.0306 | -0.346 | Small-Med | Yes* | 0.803 | 0.776 | -0.027 | 0.5997 | 0.082 | Small | No |
| East Asian | 0.441 | 0.428 | -0.012 | 0.8373 | 0.032 | Small | No | 0.698 | 0.776 | +0.079 | 0.0920 | -0.266 | Small-Med | No |
| Southeast Asian | 0.461 | 0.502 | +0.041 | 0.3987 | -0.132 | Small | No | 0.827 | 0.893 | +0.066 | 0.0675 | -0.290 | Small-Med | No |
| Middle Eastern | 0.483 | 0.539 | +0.056 | 0.2304 | -0.188 | Small | No | 0.756 | 0.766 | +0.011 | 0.7669 | -0.046 | Small | No |
| Latino | 0.631 | 0.673 | +0.042 | 0.3537 | -0.145 | Small | No | 0.860 | 0.921 | +0.062 | 0.0073 | -0.436 | Small-Med | Yes** |
| *Ethnicity Bias: Representation Deviation Score for Ethnicity ($RDS_e$)* | | | | | | | | | | | | | | |
| Black | -0.195 | -0.198 | -0.002 | 0.5700 | 0.088 | Small | No | — | — | — | — | — | — | — |
| East Asian | -0.181 | -0.198 | -0.017 | 0.0331 | 0.340 | Small-Med | Yes* | -0.150 | -0.326 | -0.176 | 0.0000 | 1.105 | Large | Yes*** |
| Latino | -0.195 | -0.193 | +0.002 | 0.5700 | -0.088 | Small | No | — | — | — | — | — | — | — |
| Middle Eastern | -0.198 | -0.195 | +0.002 | 0.6602 | -0.068 | Small | No | -0.150 | -0.229 | -0.079 | 0.0049 | 0.459 | Small-Med | Yes** |
| White | 0.769 | 0.783 | +0.014 | 0.2439 | -0.182 | Small | No | 0.686 | 0.555 | -0.131 | 0.0001 | 0.692 | Med-Large | Yes*** |
| *Ethnicity Bias: Simpson's Diversity Index (SDI)* | | | | | | | | | | | | | | |
| Overall | 0.054 | 0.027 | -0.027 | 0.1710 | 0.215 | Small-Med | No | 0.131 | 0.109 | -0.021 | 0.4920 | 0.107 | Small | No |

Table 9: Statistical comparison of base and aligned text-to-video diffusion models under paired evaluation. We report mean metric values for each base model and its aligned counterpart, along with the alignment-induced difference $\Delta = \mu_{\text{aligned}} - \mu_{\text{base}}$. Statistical significance is assessed using paired tests across the same set of verbs, with $p$-values indicating the reliability of observed differences and Cohen's $d$ quantifying standardized effect size. Significance levels are denoted as $^{***}p < 0.001$, $^{**}p < 0.01$, and $^{*}p < 0.05$. Effect sizes are categorized as Small ($|d| < 0.2$), Small–Medium ($0.2 \leq |d| < 0.5$), Medium–Large ($0.5 \leq |d| < 0.8$), and Large ($|d| \geq 0.8$). $PBS_G$ measures ethnicity-aware gender bias, $RDS_e$ captures deviation from uniform ethnic representation, and SDI reflects overall ethnic diversity (higher is more balanced). Missing entries (—) indicate cases where metrics are undefined due to zero observable instances for the corresponding subgroup.

at least as large as $\Delta$ under the null hypothesis of no alignment effect (with $p < 0.05$ indicating statistical significance), while practical magnitude is quantified using Cohen's $d$, which captures the standardized effect size independent of sample size (with larger $|d|$ indicating more substantial changes)[2]. Interpreting these measures jointly allows us to distinguish statistically reliable effects from practically meaningful ones. For **ModelScope → InstructVideo** (HPSv2.0 alignment), only 3/14 metrics exhibit statistically significant changes ($p < 0.05$), all with small-to-medium effect sizes, indicating localized rather than global bias shifts: $PBS_G$ increases for Black ($\Delta = +0.120$, $p = 0.0030$, $d = 0.49$) and Indian ($\Delta = +0.094$, $p = 0.0306$, $d = 0.35$) groups, while East Asian representation shows a modest decrease in $RDS_e$ ($\Delta = -0.017$, $p = 0.0331$, $d = 0.34$), with no significant change in overall ethnic diversity as measured by SDI. In contrast, **VideoCrafter-V2 → T2V-Turbo-V1** (HPSv2.1 alignment) yields broader and stronger effects, with 7/12 metrics reaching statistical significance and several exhibiting medium-to-large or large effect sizes, including a substantial decrease in East Asian $RDS_e$ ($\Delta = -0.176$, $p < 0.001$, $d = 1.11$) and a marked increase in Black $PBS_G$ ($\Delta = +0.193$, $p = 0.0001$, $d = 0.68$), alongside significant shifts in White representation and gender preference ($RDS_e$: $\Delta = -0.131$, $p = 0.0001$, $d = 0.69$; $PBS_G$: $\Delta = +0.123$, $p = 0.0027$, $d = 0.49$), again without corresponding improvements in SDI. Overall, these results show that alignment tuning induces statistically reliable and sometimes large redistributions in gender and ethnic representation, with the scope and magnitude of these shifts strongly dependent on the reward model, and that statistically significant changes do not necessarily imply uniform debiasing but often reflect bias reallocation across demographic groups.

---

[2]**Interpreting statistical tests.** The $p$-value quantifies the probability of observing a difference at least as large as the measured $\Delta$ under the null hypothesis of no difference between models. We adopt standard significance thresholds: $p < 0.001$ (***, extremely strong evidence), $p < 0.01$ (**, strong evidence), $p < 0.05$ (*, moderate evidence), and $p \geq 0.05$ (insufficient evidence to reject the null). Complementarily, Cohen's $d$ measures the standardized effect size, capturing the practical magnitude of the difference independent of sample size, with conventional interpretations of $|d| < 0.2$ (small), $0.2 \leq |d| < 0.5$ (small–medium), $0.5 \leq |d| < 0.8$ (medium–large), and $|d| \geq 0.8$ (large).

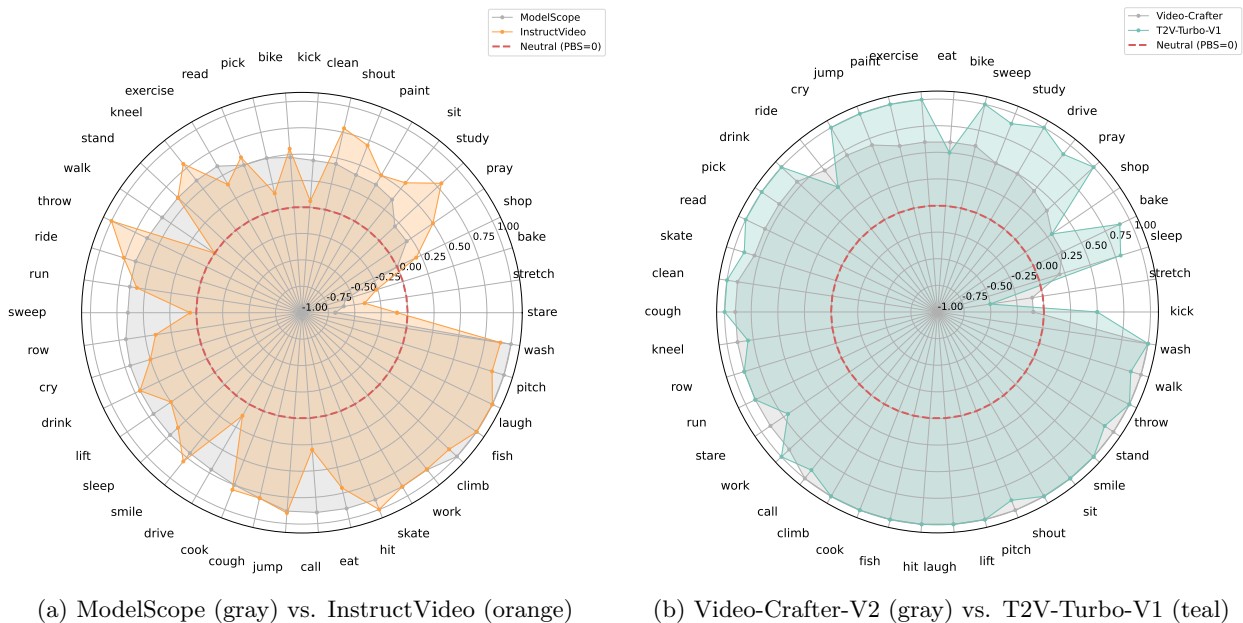

(a) ModelScope (gray) vs. InstructVideo (orange)  (b) Video-Crafter-V2 (gray) vs. T2V-Turbo-V1 (teal)

Figure 7: Ethnicity-aware gender bias for the **White** subgroup across 42 verbs. We compare unaligned and aligned model pairs: ModelScope (unaligned) vs. InstructVideo (aligned), and VideoCrafter-V2 (unaligned) vs. T2V-Turbo-V1 (aligned). Values are plotted relative to a neutral zone centered at zero; regions *outside* this zone indicate systematic bias, with *positive* values denoting man-preference (+) and *negative* values denoting woman-preference (−).

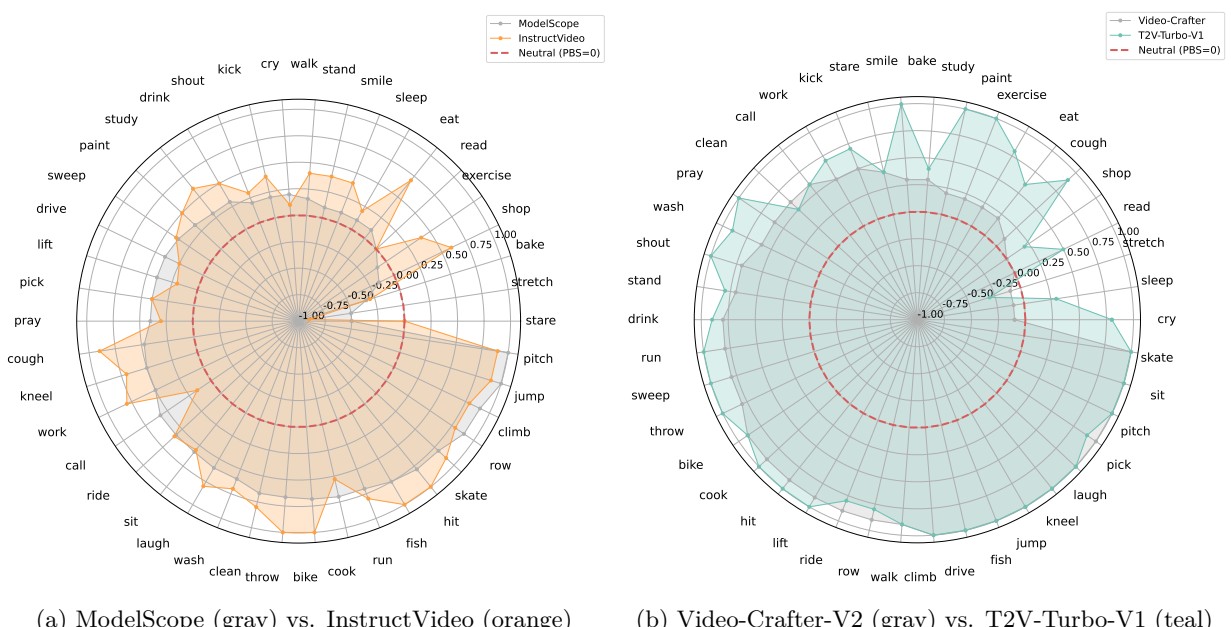

(a) ModelScope (gray) vs. InstructVideo (orange)  (b) Video-Crafter-V2 (gray) vs. T2V-Turbo-V1 (teal)

Figure 8: Ethnicity-aware gender bias for the **Black** subgroup across 42 verbs. We compare unaligned and aligned model pairs: ModelScope (unaligned) vs. InstructVideo (aligned), and VideoCrafter-V2 (unaligned) vs. T2V-Turbo-V1 (aligned). Values are plotted relative to a neutral zone centered at zero; regions *outside* this zone indicate systematic bias, with *positive* values denoting man-preference (+) and *negative* values denoting woman-preference (−).

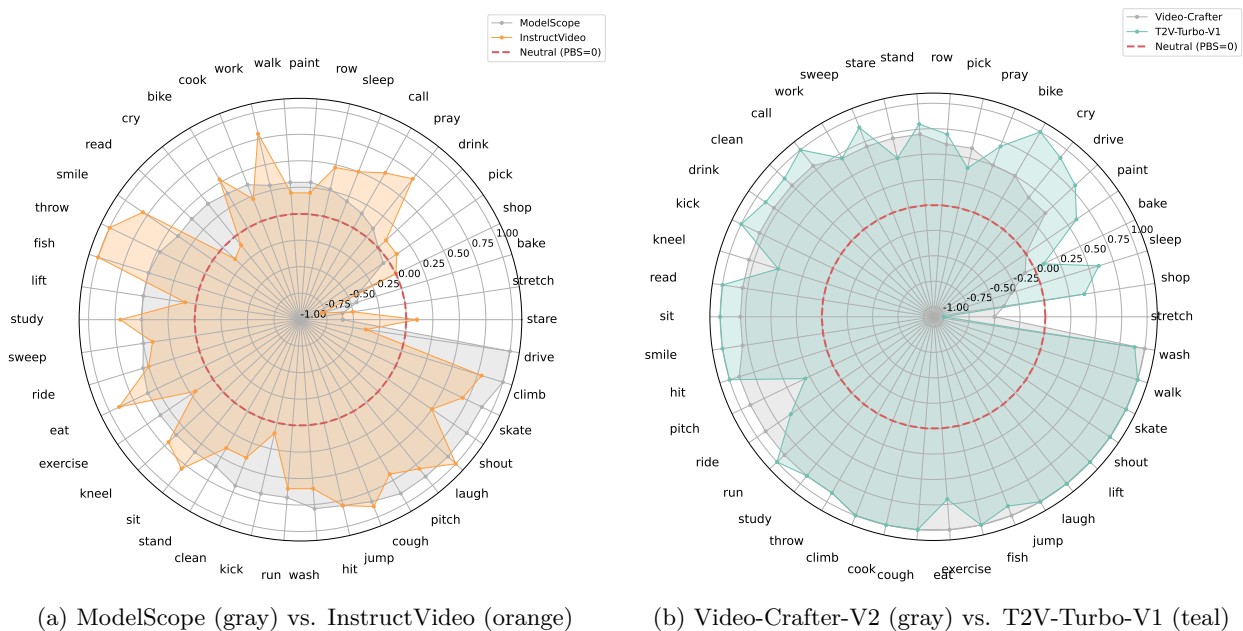

(a) ModelScope (gray) vs. InstructVideo (orange)  (b) Video-Crafter-V2 (gray) vs. T2V-Turbo-V1 (teal)

Figure 9: Ethnicity-aware gender bias for the **East Asian** subgroup across 42 verbs. We compare unaligned and aligned model pairs: ModelScope (unaligned) vs. InstructVideo (aligned), and VideoCrafter-V2 (unaligned) vs. T2V-Turbo-V1 (aligned). Values are plotted relative to a neutral zone centered at zero; regions *outside* this zone indicate systematic bias, with *positive* values denoting man-preference (+) and *negative* values denoting woman-preference (−).

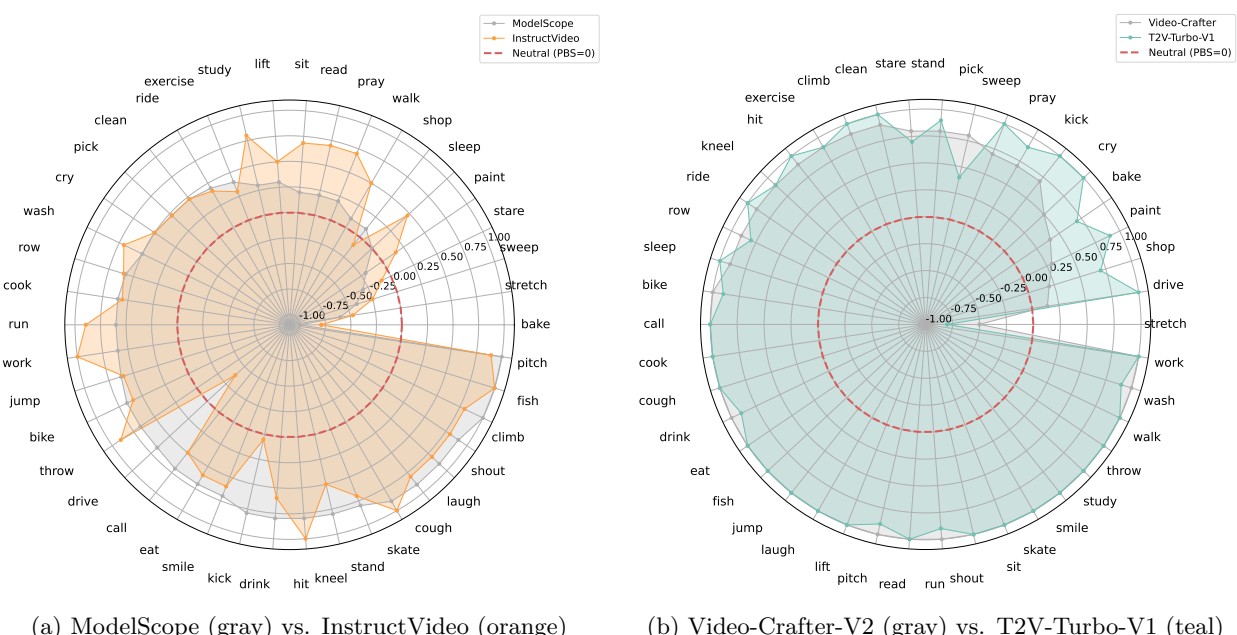

(a) ModelScope (gray) vs. InstructVideo (orange)  (b) Video-Crafter-V2 (gray) vs. T2V-Turbo-V1 (teal)

Figure 10: Ethnicity-aware gender bias for the **Southeast Asian** subgroup across 42 verbs. We compare unaligned and aligned model pairs: ModelScope (unaligned) vs. InstructVideo (aligned), and VideoCrafter-V2 (unaligned) vs. T2V-Turbo-V1 (aligned). Values are plotted relative to a neutral zone centered at zero; regions *outside* this zone indicate systematic bias, with *positive* values denoting man-preference (+) and *negative* values denoting woman-preference (−).

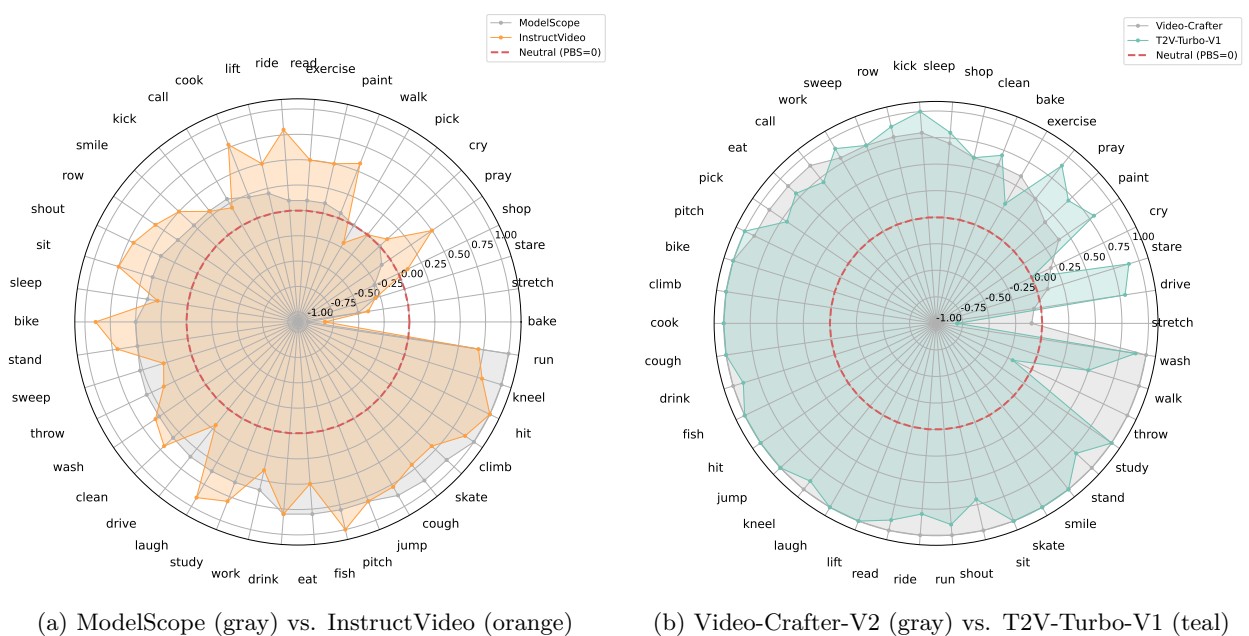

(a) ModelScope (gray) vs. InstructVideo (orange)

(b) Video-Crafter-V2 (gray) vs. T2V-Turbo-V1 (teal)

Figure 11: Ethnicity-aware gender bias for the **Indian** subgroup across 42 verbs. We compare unaligned and aligned model pairs: ModelScope (unaligned) vs. InstructVideo (aligned), and VideoCrafter-V2 (unaligned) vs. T2V-Turbo-V1 (aligned). Values are plotted relative to a neutral zone centered at zero; regions *outside* this zone indicate systematic bias, with *positive* values denoting man-preference (+) and *negative* values denoting woman-preference (−).

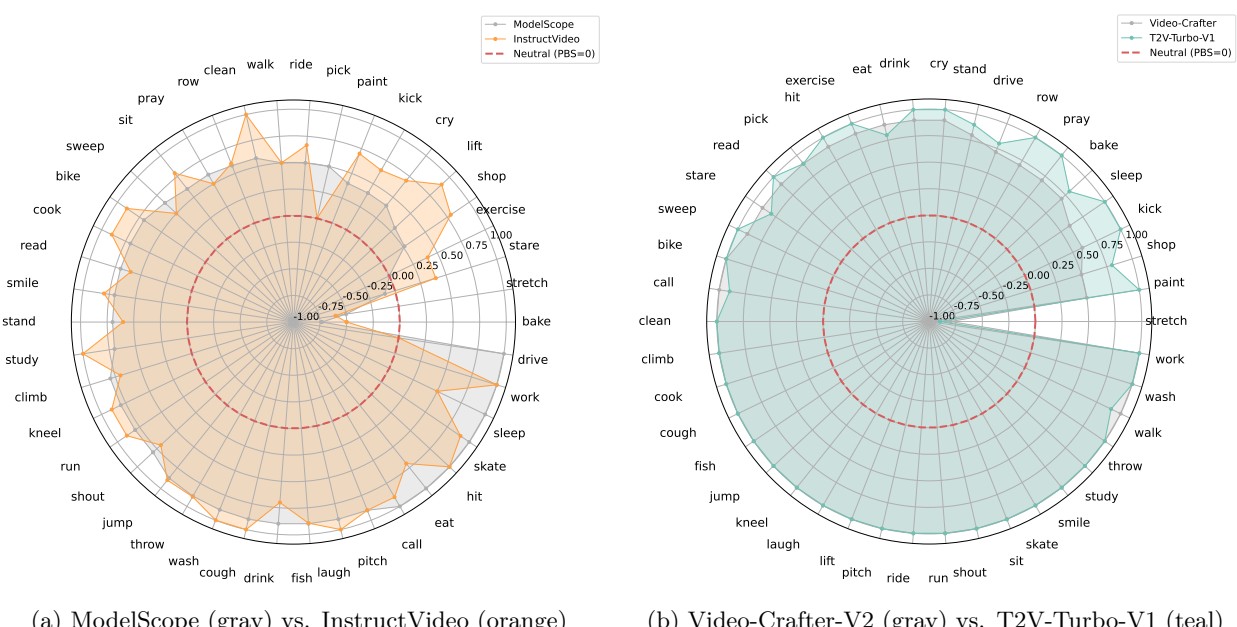

(a) ModelScope (gray) vs. InstructVideo (orange)

(b) Video-Crafter-V2 (gray) vs. T2V-Turbo-V1 (teal)

Figure 12: Ethnicity-aware gender bias for the **Latino** subgroup across 42 verbs. We compare unaligned and aligned model pairs: ModelScope (unaligned) vs. InstructVideo (aligned), and VideoCrafter-V2 (unaligned) vs. T2V-Turbo-V1 (aligned). Values are plotted relative to a neutral zone centered at zero; regions *outside* this zone indicate systematic bias, with *positive* values denoting man-preference (+) and *negative* values denoting woman-preference (−).

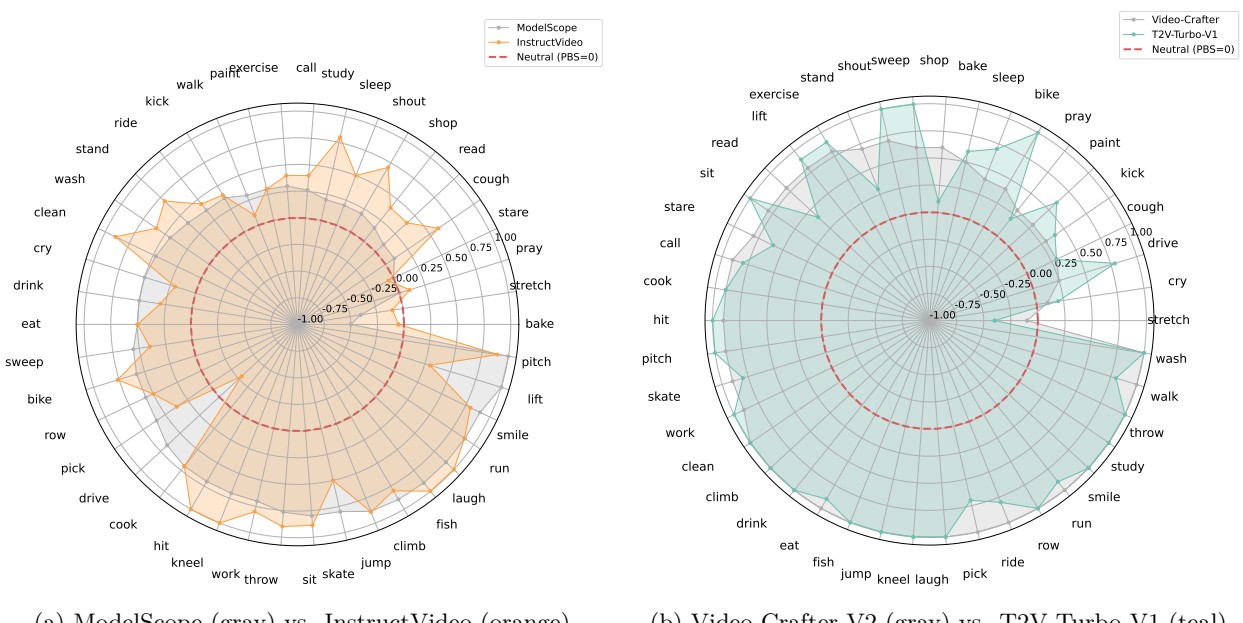

(a) ModelScope (gray) vs. InstructVideo (orange)    (b) Video-Crafter-V2 (gray) vs. T2V-Turbo-V1 (teal)

Figure 13: Ethnicity-aware gender bias for the **Middle Eastern** subgroup across 42 verbs. We compare unaligned and aligned model pairs: ModelScope (unaligned) vs. InstructVideo (aligned), and VideoCrafter-V2 (unaligned) vs. T2V-Turbo-V1 (aligned). Values are plotted relative to a neutral zone centered at zero; regions *outside* this zone indicate systematic bias, with *positive* values denoting man-preference (+) and *negative* values denoting woman-preference (−).

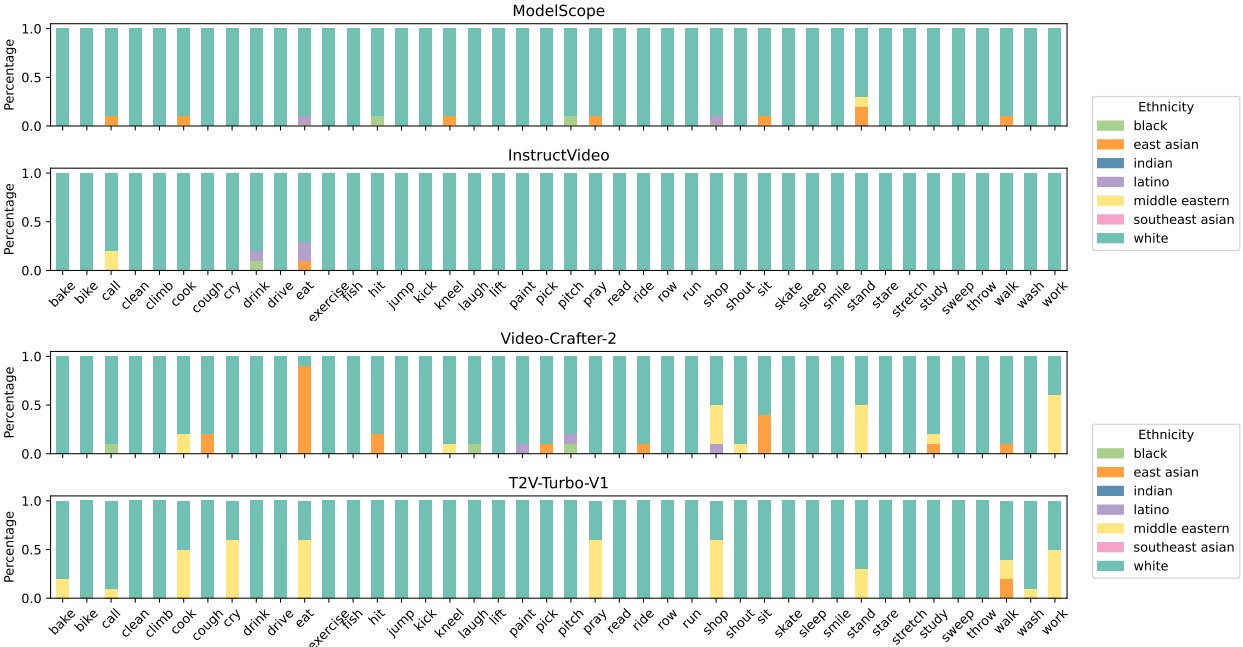

Figure 14: Ethnicity bias distribution across different unaligned and aligned model pairs: ModelScope (unaligned) vs. InstructVideo (aligned), and VideoCrafter-V2 (unaligned) vs. T2V-Turbo-V1 (aligned).

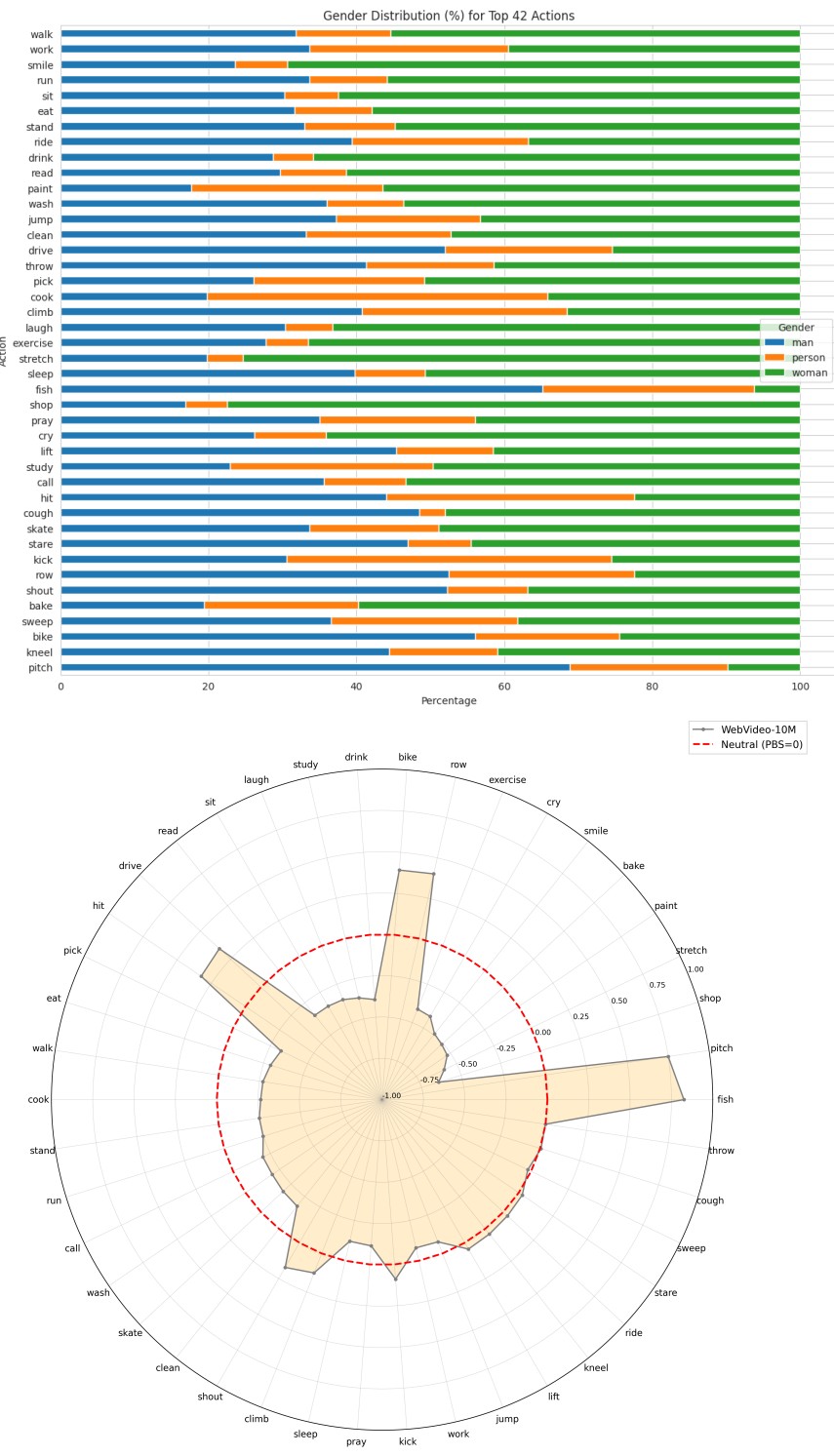

Figure 15: Gender distribution across 42 verbs in the WebVid-10M training data. Top: raw gender frequency distribution. Bottom: relative gender distribution measured by proportional bias score (PBS), highlighting deviations from a balanced baseline.

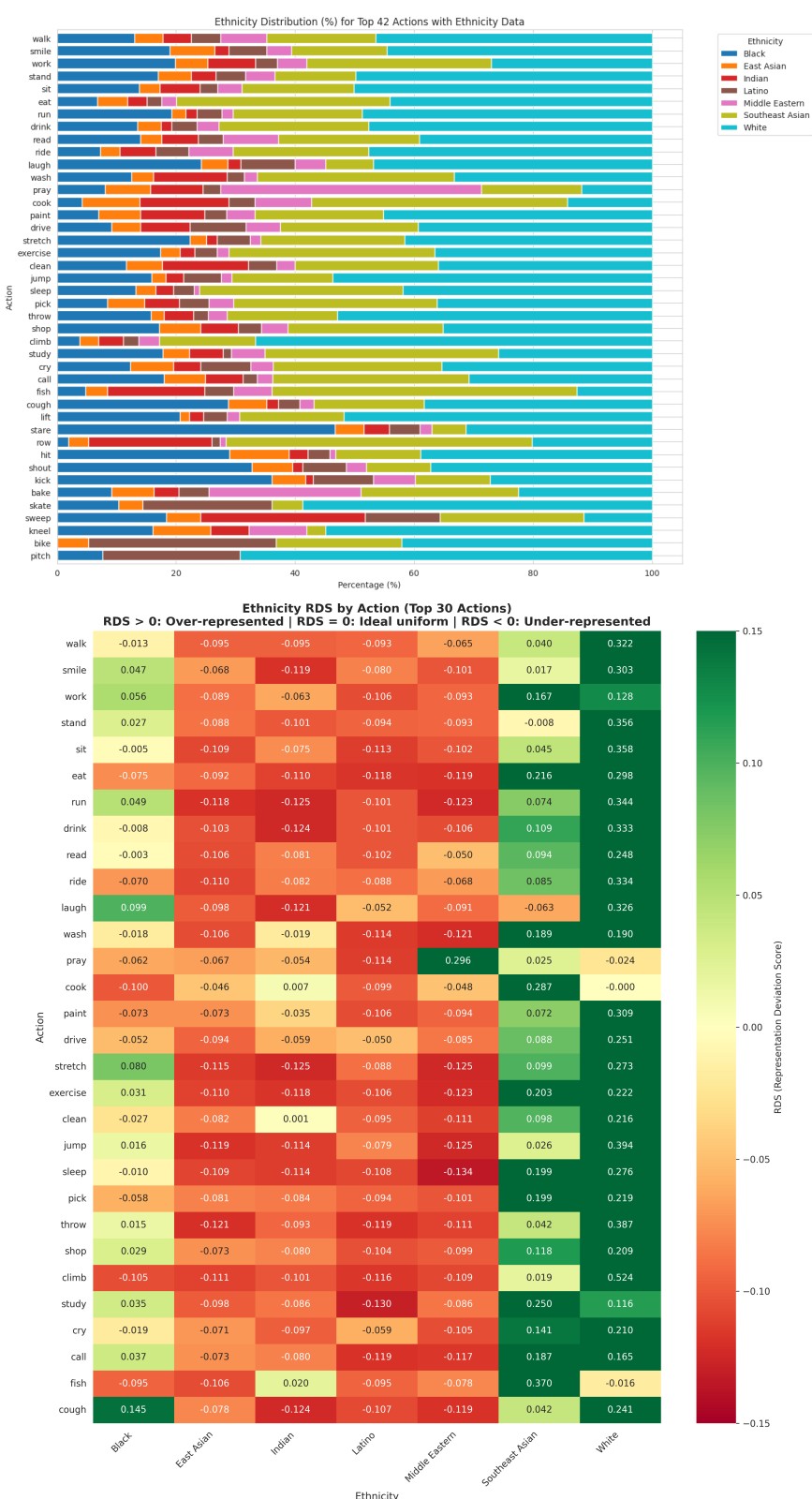

Figure 16: Ethnicity distribution across 42 verbs in the WebVid-10M training data. Top: raw gender frequency distribution. Bottom: relative gender distribution measured by Representation Deviation Score (RDS), highlighting deviations from a balanced baseline.

# B   Social Biases in Image Reward Datasets

We analyze *two* widely used image reward datasets to investigate human preference biases: HPDv2 (Wu et al., 2023) and Pick-a-Pic (Kirstain et al., 2023b). For each dataset, we extract gender, ethnicity, and verb attributes from image captions using GPT-4o-mini, and classify attributes from images using three VLMs (Qwen2-VL-7B, Qwen2.5-VL-7B, InternVL2.5-8B). We then aggregate the social attributes from both caption and image modalities, retaining only instances featuring one of our predefined verbs. After processing, HPDv2 contains 28,783 validated (images, caption, preference) tuples covering 29 verbs, and Pick-a-Pic contains 14,958 across 19 verbs. Each tuple presents two images, with a human annotator selecting the one that best matches the caption. To assess potential human preference biases, we measure how often annotators *prefer* specific gender or ethnicity representations for given verbs.

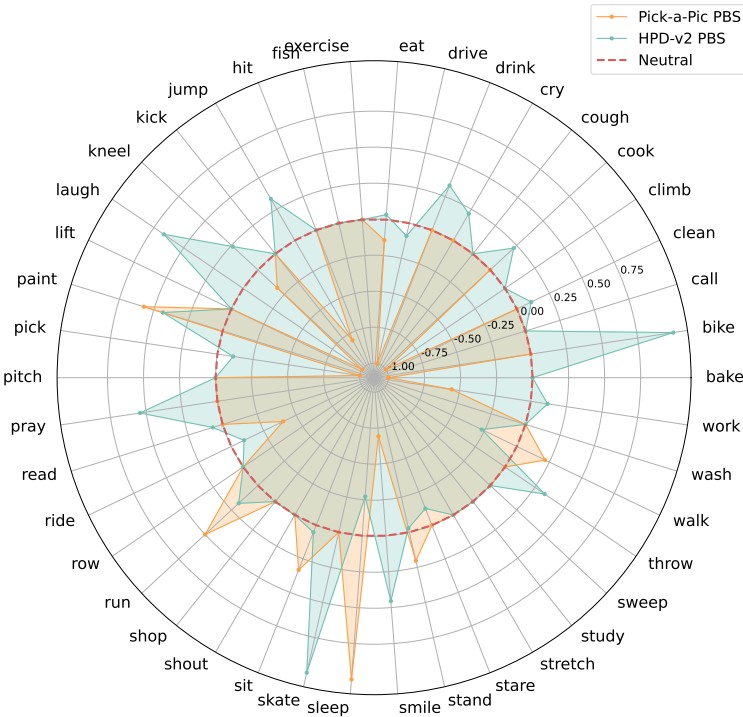

Figure 17: Image reward datasets gender preference distribution.

We directly analyze preference pairs where the preferred image depicts one gender (*e.g.*, a man) and the dispreferred image depicts the other (*e.g.*, a woman), thereby capturing explicit gender preference patterns within individual comparison pairs. Figure 17 shows the gender preference bias across 42 verbs in the two datasets. After filtering to ensure valid man–woman pairs, 26 out of 42 verbs in **HPDv2** met our criteria. Among these, 69.23% (18/26) showed a preference for men, revealing a consistent **man-preferred** tendency. In contrast, in **Pick-a-Pic**, 18 out of 42 verbs qualified, and 61.11% (11/18) showed a preference for women, indicating a relatively **woman-preferred** trend. Furthermore, Table 10 and **??** present the ethnicity preference distributions across the two image reward datasets. Notably, both datasets exhibit a strong preference for the **White** group, 43.34% in HPDv2 and 40.08% in Pick-a-Pic, followed by East Asian and Indian representations. Despite certain verbs showing distinct preferences (*e.g.*, "bake" favoring Black individuals and "fish" favoring East Asians), the overall distributions reveal collected human preferences may implicitly favor Western-centric aesthetics and representation. These imbalances in human preferences might risk propagating representational bias during reward model training, thereby reinforcing existing social inequities in downstream video generation. Collectively, our findings underscore the urgent need for more inclusive and demographically representative preference datasets that capture global diversity.

Table 10 presents the ethnicity preference distribution across the two image reward datasets, while Figure 4 provides a fine-grained breakdown across 42 verbs. Notably, both datasets exhibit a strong preference for the

**White** group, 43.34% in HPDv2 and 40.08% in Pick-a-Pic, followed by East Asian and Indian representations. Despite certain verbs showing distinct preferences (*e.g.*, "bake" favoring Black individuals and "fish" favoring East Asians), the overall distributions reveal a pronounced imbalance skewed toward White representations. This suggests that the reward signals used to guide image generation may reflect and reinforce ethnic biases embedded in the datasets. This imbalance in collected preferences risks might propagate representational bias during reward model training, ultimately reinforcing societal inequities in downstream video generation. These findings underscore the urgent need for more inclusive and representative datasets that reflect global demographic diversity in both identity and activity contexts.

| Datasets | White | Black | Latino | East Asian | Southeast Asian | India | Middle Eastern |
|---|---|---|---|---|---|---|---|
| HPDv2 | 43.34 | 9.16 | 4.44 | 19.38 | 1.39 | 20.20 | 2.09 |
| Pick-a-Pic | 40.08 | 15.36 | 8.51 | 19.94 | 0.20 | 13.34 | 2.56 |

Table 10: Ethnicity distribution across reward datasets (in %).

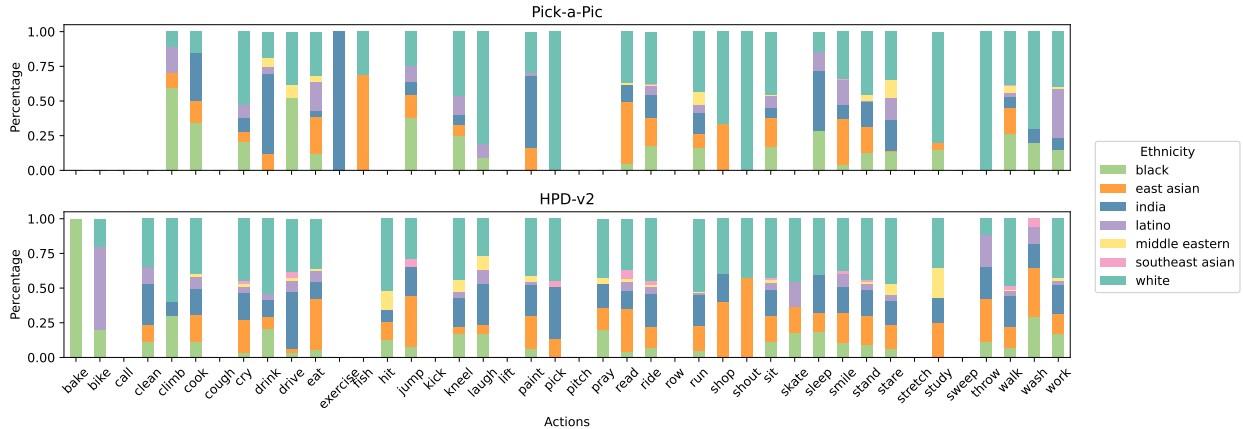

Figure 18: Ethnicity preference distribution across 42 verbs.

# C Social Biases in Image Reward Models

**Preference Bias Evaluation.** We evaluate *four* image reward models: (1) HPSv2.0 (Wu et al., 2023), trained on the HPDv2 dataset; (2) HPSv2.1 (Wu et al., 2023), trained on the unreleased HPDv2.1 dataset; (3) PickScore (Kirstain et al., 2023b), trained on the Pick-a-Pic dataset; and (4) CLIP (Radford et al., 2021), which serves as the shared base model for HPSv2.0, HPSv2.1, and PickScore prior to fine-tuning on their respective preference datasets. Each generated sample is assigned a *continuous scalar preference score* predicted by the corresponding reward model (*e.g.*, the HPSv2 score), rather than a discrete class label. These standardized scores are used to quantify the model's *relative preference* for gendered and ethnic attributes across controlled comparisons. Table 4 reports two complementary bias metrics—ethnicity-aware gender bias ($PBS_G$) and ethnic representation distribution ($RDS_e$ and SDI).

**Ethnicity-Aware Gender Bias.** We construct preference evaluation prompts in the format "A/An [ethnicity] person is [verb]-ing [context]", covering all combinations of ethnicity and verb (evaluation: ethnicity+person). For each preference prompt, we generate images using generation prompts in the format "A/An [ethnicity] [gender] is [verb]-ing [context]", where gender, ethnicity, and verb are explicitly specified (generation: ethnicity+gender). The evaluation prompt omits gender to measure reward model's inherent preference, while the generation prompt explicitly specifies gender. The reward scores assigned to these images by a reward model are standardized using their mean and standard deviation. We then compute the average standardized score across the 100 images for each generation prompt. To compute

the final $PBS_G$, we fix the ethnicity and verb, and subtract the average standardized score for women from that for men. Because of the adaptation, $PBS_G$ score here can be greater than one. A positive $PBS_G$ score indicates a preference for men, while a negative score reflects a preference for women. CLIP shows a slight woman-preference bias (–0.0726). Fine-tuning on HPDv2 shifts HPSv2.0 toward a *strong* man-preference (+0.6039), consistent across ethnic groups. In contrast, PickScore (–0.1157) and HPSv2.1 (–0.0984) show woman-preference biases, with the latter's training data undisclosed. These shifts align with each model's training data, revealing consistent gender preferences across ethnicities. Figures 19 to 25 presents the $PBS_G$ scores across 42 verbs for each ethnicity group.

**Ethnicity Bias.** We use preference evaluation prompts in the form "A person is [verb]-ing [context]" (evaluation: `person-only`). For each preference prompt, we have generated images using more specific generation prompts of the form "A/An [ethnicity] person is [verb]-ing [context]", where the ethnicity and verb are explicitly specified (generation: `ethnicity+person`). The evaluation prompt omits ethnicity to measure reward model's inherent preference, while the generation prompt explicitly specifies ethnicity. The reward scores for these images provided by a reward model are standardized with mean and standard deviation. We then compute the average standardized score across the 100 images for each generation prompt. To calculate $RDS_e$ and SDI, we fix the verb and apply softmax function (Bridle, 1990; Bishop, 2006) to normalize the scores for each ethnicity, indicating ethnicity preference within each verb context. A positive $RDS_e$ indicates overrepresentation of an ethnicity, while a negative score indicates underrepresentation. A higher SDI score corresponds to more balanced and diverse outputs across all groups. The base model, CLIP, slightly favors White individuals (RDS = 0.0182) and achieves the highest SDI score (0.8495), indicating relatively balanced ethnic representation. After fine-tuning, HPSv2.0 shifts toward Middle Eastern (RDS = 0.0315), HPSv2.1 toward Latino (RDS = 0.0382), and PickScore toward East Asian individuals (RDS = 0.0352). All show reduced SDI, indicating decreased ethnic diversity post-alignment. Figures 26 to 29 show the ethnicity bias across 42 verbs.

| Metric | CLIP Base | +HPSv2.0 Mean | Δ | p-val | d | Eff. | Sig. | +HPSv2.1 Mean | Δ | p-val | d | Eff. | Sig. | +PickScore Mean | Δ | p-val | d | Eff. | Sig. |
|---|---|---|---|---|---|---|---|---|---|---|---|---|---|---|---|---|---|---|---|
| | | *Ethnicity-Aware Gender Bias: Proportion Bias Score for Gender ($PBS_G$)* | | | | | | | | | | | | | | | | | |
| Average | -0.073 | 0.604 | +0.676 | 0.000 | -1.59 | L | Yes*** | -0.098 | -0.026 | 0.621 | 0.08 | S | No | -0.116 | -0.043 | 0.412 | 0.13 | S | No |
| White | 0.034 | 0.609 | +0.575 | 0.000 | -1.08 | L | Yes*** | -0.083 | -0.118 | 0.047 | 0.32 | S-M | Yes* | 0.032 | -0.002 | 0.976 | 0.01 | S | No |
| Black | -0.120 | 0.734 | +0.854 | 0.000 | -1.84 | L | Yes*** | 0.026 | +0.145 | 0.025 | -0.36 | S-M | Yes* | -0.078 | +0.042 | 0.422 | -0.13 | S | No |
| Indian | -0.051 | 0.592 | +0.643 | 0.000 | -1.33 | L | Yes*** | -0.001 | +0.050 | 0.445 | -0.12 | S | No | 0.153 | +0.204 | 0.006 | -0.44 | S-M | Yes** |
| East Asian | -0.132 | 0.475 | +0.607 | 0.000 | -1.31 | L | Yes*** | -0.304 | -0.173 | 0.010 | 0.42 | S-M | Yes* | -0.226 | -0.094 | 0.113 | 0.25 | S-M | No |
| Southeast Asian | -0.086 | 0.519 | +0.606 | 0.000 | -1.48 | L | Yes*** | -0.218 | -0.132 | 0.034 | 0.34 | S-M | Yes* | -0.216 | -0.130 | 0.026 | 0.36 | S-M | Yes* |
| Middle Eastern | -0.061 | 0.646 | +0.707 | 0.000 | -1.54 | L | Yes*** | -0.105 | -0.045 | 0.403 | 0.13 | S | No | -0.128 | -0.067 | 0.277 | 0.17 | S | No |
| Latino | -0.093 | 0.651 | +0.745 | 0.000 | -1.71 | L | Yes*** | -0.003 | +0.090 | 0.123 | -0.24 | S-M | No | -0.348 | -0.255 | 0.000 | 0.79 | M-L | Yes*** |
| | | *Ethnicity Bias: Representation Deviation Score for Ethnicity ($RDS_e$)* | | | | | | | | | | | | | | | | | |
| Black | 0.000 | -0.008 | -0.008 | 0.173 | 0.22 | S-M | No | -0.034 | -0.034 | 0.000 | 0.96 | L | Yes*** | 0.028 | +0.028 | 0.000 | -0.72 | M-L | Yes*** |
| East Asian | 0.013 | -0.003 | -0.017 | 0.002 | 0.53 | M-L | Yes** | 0.009 | -0.004 | 0.499 | 0.11 | S | No | 0.031 | +0.018 | 0.002 | -0.53 | M-L | Yes** |
| Indian | -0.028 | 0.007 | +0.035 | 0.000 | -0.98 | L | Yes*** | -0.007 | +0.021 | 0.001 | -0.56 | M-L | Yes** | -0.039 | -0.011 | 0.022 | 0.38 | S-M | Yes* |
| Latino | -0.001 | 0.025 | +0.025 | 0.000 | -0.74 | M-L | Yes*** | 0.038 | +0.039 | 0.000 | -1.00 | L | Yes*** | -0.011 | -0.010 | 0.053 | 0.32 | S-M | No |
| Middle Eastern | -0.010 | 0.032 | +0.042 | 0.000 | -1.62 | L | Yes*** | 0.023 | +0.033 | 0.000 | -1.18 | L | Yes*** | -0.025 | -0.015 | 0.002 | 0.54 | M-L | Yes** |
| Southeast Asian | 0.008 | -0.010 | -0.018 | 0.001 | 0.60 | M-L | Yes*** | -0.009 | -0.018 | 0.001 | 0.56 | M-L | Yes** | 0.011 | +0.003 | 0.611 | -0.08 | S | No |
| White | 0.017 | -0.043 | -0.060 | 0.000 | 1.27 | L | Yes*** | -0.020 | -0.037 | 0.000 | 0.61 | M-L | Yes*** | 0.005 | -0.012 | 0.138 | 0.24 | S-M | No |
| | | *Ethnicity Bias: Simpson's Diversity Index (SDI)* | | | | | | | | | | | | | | | | | |
| Overall | 0.850 | 0.849 | -0.001 | 0.667 | 0.07 | S | No | 0.847 | -0.003 | 0.096 | 0.27 | S-M | No | 0.849 | -0.001 | 0.377 | 0.14 | S | No |

*Eff. Size: S=Small, S-M=Small-Medium, M-L=Medium-Large, L=Large.*

Table 11: Statistical comparison of preference bias across image reward models relative to a CLIP baseline. For each metric, we report the mean value under CLIP and under CLIP augmented with a given reward model (HPSv2.0, HPSv2.1, or PickScore), along with the difference $\Delta = \mu_{\text{CLIP+RM}} - \mu_{\text{CLIP}}$. Statistical significance is assessed using paired tests across the same set of verbs, with $p$-values indicating the reliability of observed differences and Cohen's $d$ quantifying standardized effect size. Significance levels are denoted as ***$p < 0.001$, **$p < 0.01$, and *$p < 0.05$. Effect sizes are categorized as Small ($|d| < 0.2$), Small–Medium ($0.2 \leq |d| < 0.5$), Medium–Large ($0.5 \leq |d| < 0.8$), and Large ($|d| \geq 0.8$). $PBS_G$ measures ethnicity-aware gender bias, $RDS_e$ captures deviation from uniform ethnic representation, and SDI reflects overall ethnic diversity (higher is more balanced). Missing entries (—) indicate cases where metrics are undefined due to zero observable instances for the corresponding subgroup.

**Statistical Analysis of Reward Models.** We further conduct a paired statistical analysis to quantify how different image reward models deviate from the CLIP baseline, with results reported in Table 11. For each

metric, we compute the difference $\Delta = \mu_{\mathrm{reward}} - \mu_{\mathrm{CLIP}}$ and assess statistical reliability using $p$-values, alongside Cohen's $d$ to measure practical effect size. Overall, the three reward models exhibit markedly different bias profiles relative to CLIP. **CLIP $\rightarrow$ CLIP+HPSv2.0** shows the most dramatic and systematic shifts: 14/16 metrics differ significantly from CLIP ($p < 0.05$), with consistently large effect sizes across ethnicity-aware gender bias metrics. In particular, the averaged $\mathrm{PBS}_G$ increases sharply ($\Delta = +0.676$, $p < 0.001$, $d = 1.59$), and all ethnicity-specific $\mathrm{PBS}_G$ scores exhibit large, statistically robust changes, indicating a strong amplification of gender preference patterns encoded by HPSv2.0. In contrast, **CLIP $\rightarrow$ CLIP+HPSv2.1** yields fewer significant deviations (10/16 metrics), with substantially smaller magnitudes; notably, the averaged $\mathrm{PBS}_G$ change is not statistically significant ($\Delta = -0.026$, $p = 0.6208$, $d = 0.08$), suggesting that HPSv2.1 induces more moderate and heterogeneous bias shifts. **CLIP $\rightarrow$ CLIP+PickScore** produces the weakest overall departure from CLIP, with only 7/16 metrics reaching significance and no significant change in averaged $\mathrm{PBS}_G$ ($\Delta = -0.043$, $p = 0.4117$, $d = 0.13$), although several ethnicity-specific $\mathrm{RDS}_e$ metrics still show medium-to-large effects. Across all three reward models, changes in SDI are consistently small and statistically insignificant, indicating that reward modeling primarily redistributes bias across demographic groups rather than altering overall ethnic diversity. Taken together, these results demonstrate that reward model choice plays a decisive role in shaping downstream bias characteristics, with HPSv2.0 exerting the strongest and most systematic influence, while HPSv2.1 and PickScore induce comparatively weaker and more selective bias shifts.

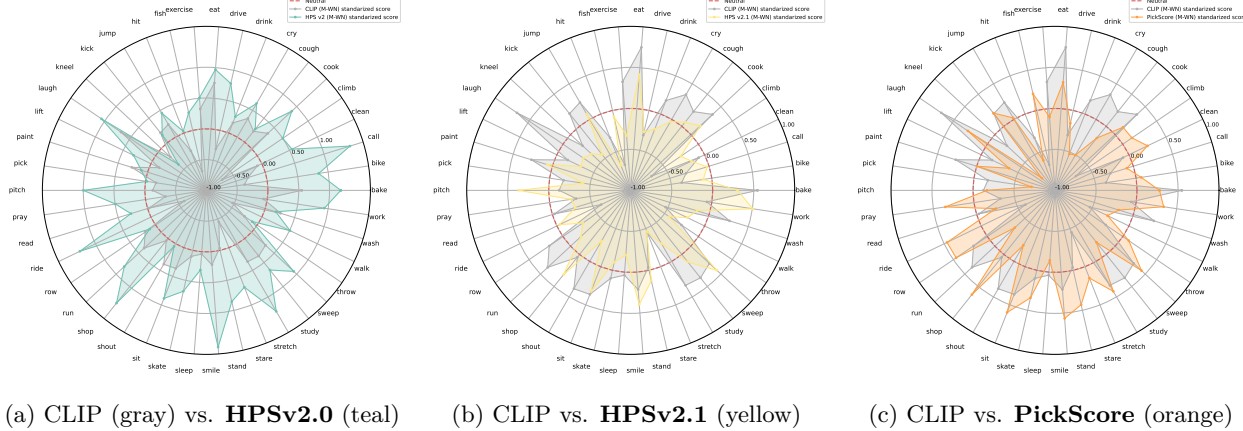

(a) CLIP (gray) vs. **HPSv2.0** (teal)  (b) CLIP vs. **HPSv2.1** (yellow)  (c) CLIP vs. **PickScore** (orange)

Figure 19: Ethnicity-aware gender bias for the **White** subgroup across 42 verbs. We compare the baseline image reward model CLIP with its preference-aligned variants (HPSv2.0, HPSv2.1, PickScore). Values are shown relative to a zero-centered neutral zone; regions *outside* this zone indicate systematic gender bias, with *positive* values corresponding to man-preference ($+$) and *negative* values corresponding to woman-preference ($-$).

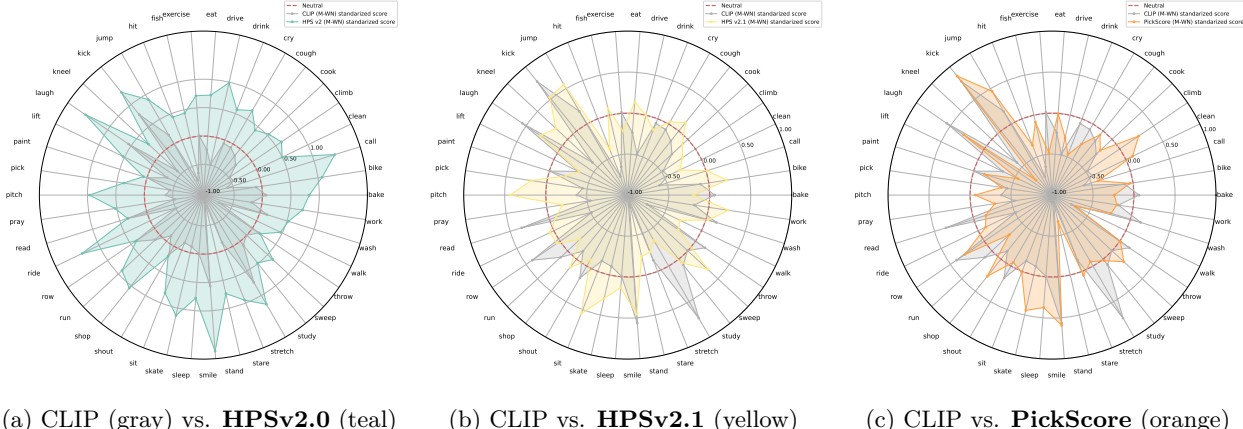

(a) CLIP (gray) vs. **HPSv2.0** (teal)    (b) CLIP vs. **HPSv2.1** (yellow)    (c) CLIP vs. **PickScore** (orange)

Figure 20: Ethnicity-aware gender bias for the **Black** subgroup across 42 verbs. We compare the baseline image reward model CLIP with its preference-aligned variants (HPSv2.0, HPSv2.1, PickScore). Values are shown relative to a zero-centered neutral zone; regions *outside* this zone indicate systematic gender bias, with *positive* values corresponding to man-preference (+) and *negative* values corresponding to woman-preference (−).

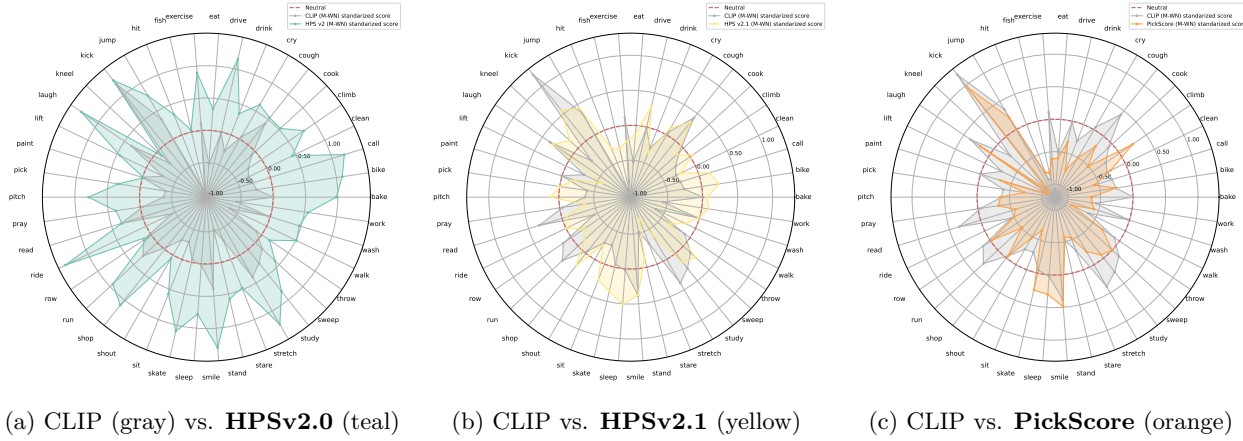

(a) CLIP (gray) vs. **HPSv2.0** (teal)    (b) CLIP vs. **HPSv2.1** (yellow)    (c) CLIP vs. **PickScore** (orange)

Figure 21: Ethnicity-aware gender bias for the **Latino** subgroup across 42 verbs. We compare the baseline image reward model CLIP with its preference-aligned variants (HPSv2.0, HPSv2.1, PickScore). Values are shown relative to a zero-centered neutral zone; regions *outside* this zone indicate systematic gender bias, with *positive* values corresponding to man-preference (+) and *negative* values corresponding to woman-preference (−).

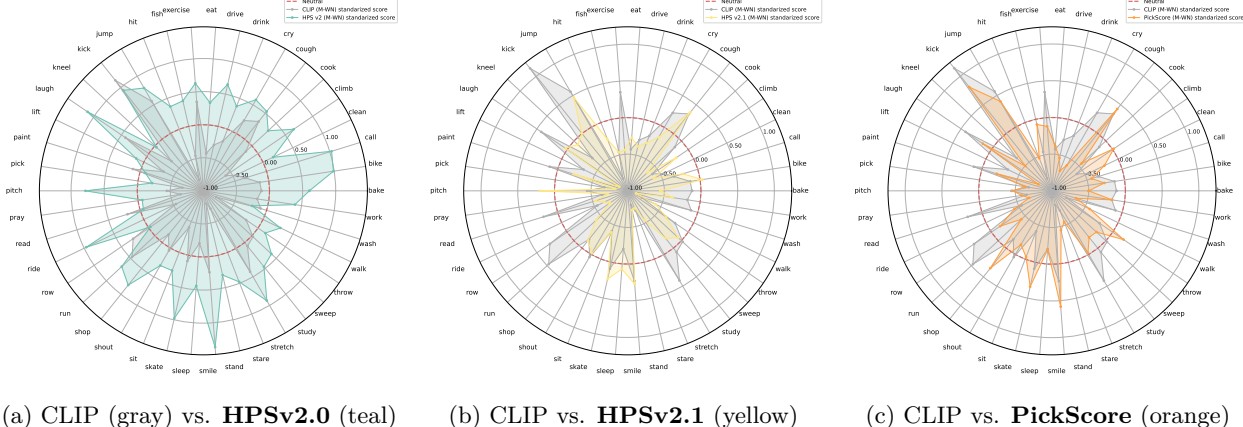

(a) CLIP (gray) vs. **HPSv2.0** (teal)    (b) CLIP vs. **HPSv2.1** (yellow)    (c) CLIP vs. **PickScore** (orange)

Figure 22: Ethnicity-aware gender bias for the **East Asian** subgroup across 42 verbs. We compare the baseline image reward model CLIP with its preference-aligned variants (HPSv2.0, HPSv2.1, PickScore). Values are shown relative to a zero-centered neutral zone; regions *outside* this zone indicate systematic gender bias, with *positive* values corresponding to man-preference (+) and *negative* values corresponding to woman-preference (−).

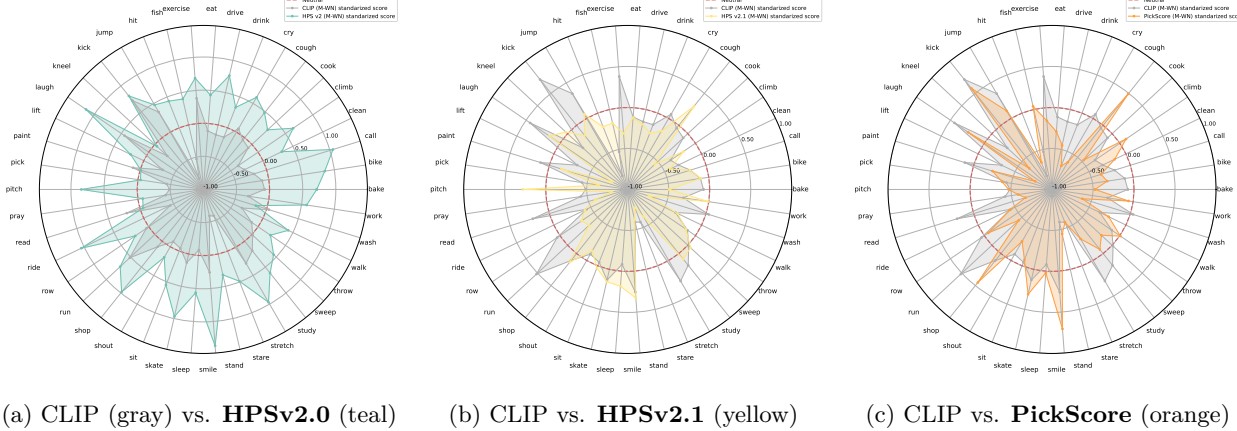

(a) CLIP (gray) vs. **HPSv2.0** (teal)    (b) CLIP vs. **HPSv2.1** (yellow)    (c) CLIP vs. **PickScore** (orange)

Figure 23: Ethnicity-aware gender bias for the **Southeast Asian** subgroup across 42 verbs. We compare the baseline image reward model CLIP with its preference-aligned variants (HPSv2.0, HPSv2.1, PickScore). Values are shown relative to a zero-centered neutral zone; regions *outside* this zone indicate systematic gender bias, with *positive* values corresponding to man-preference (+) and *negative* values corresponding to woman-preference (−).

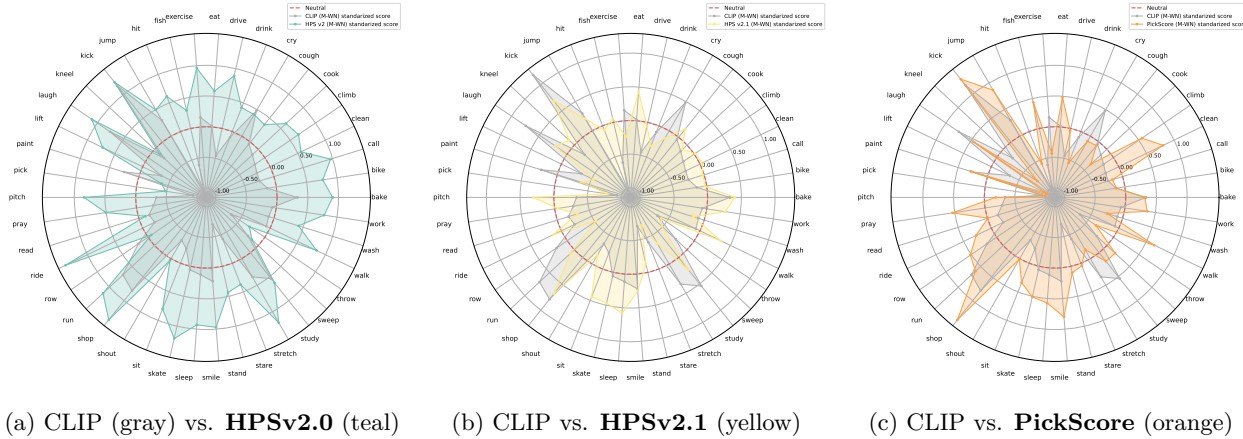

(a) CLIP (gray) vs. **HPSv2.0** (teal)  (b) CLIP vs. **HPSv2.1** (yellow)  (c) CLIP vs. **PickScore** (orange)

Figure 24: Ethnicity-aware gender bias for the **Indian** subgroup across 42 verbs. We compare the baseline image reward model CLIP with its preference-aligned variants (HPSv2.0, HPSv2.1, PickScore). Values are shown relative to a zero-centered neutral zone; regions *outside* this zone indicate systematic gender bias, with *positive* values corresponding to man-preference (+) and *negative* values corresponding to woman-preference (−).

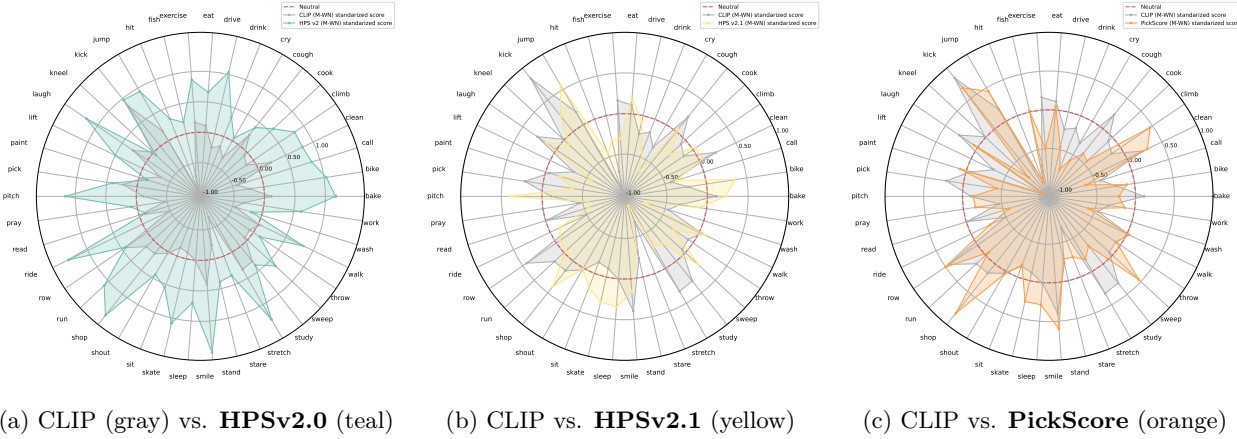

(a) CLIP (gray) vs. **HPSv2.0** (teal)  (b) CLIP vs. **HPSv2.1** (yellow)  (c) CLIP vs. **PickScore** (orange)

Figure 25: Ethnicity-aware gender bias for the **Middle Eastern** subgroup across 42 verbs. We compare the baseline image reward model CLIP with its preference-aligned variants (HPSv2.0, HPSv2.1, PickScore). Values are shown relative to a zero-centered neutral zone; regions *outside* this zone indicate systematic gender bias, with *positive* values corresponding to man-preference (+) and *negative* values corresponding to woman-preference (−).

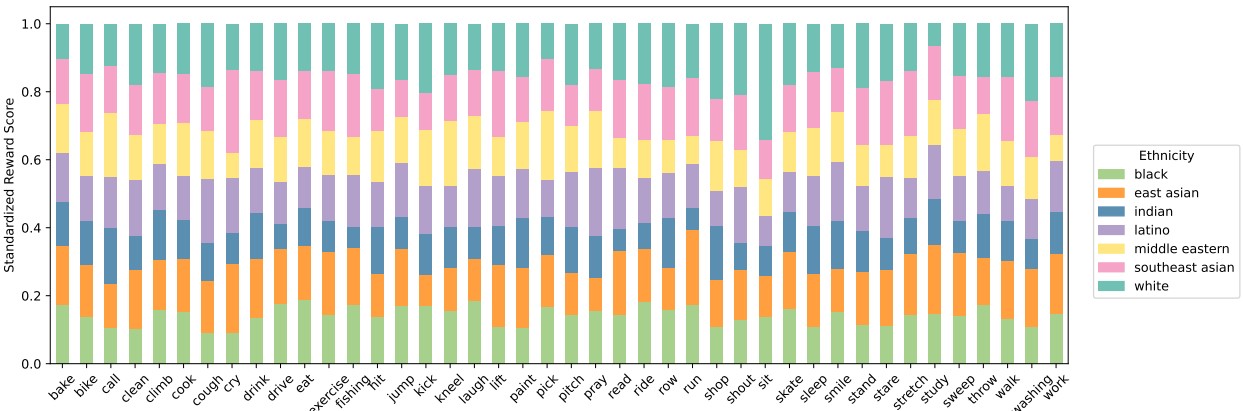

Figure 26: Ethnicity representation bias in **CLIP** across 42 verbs.

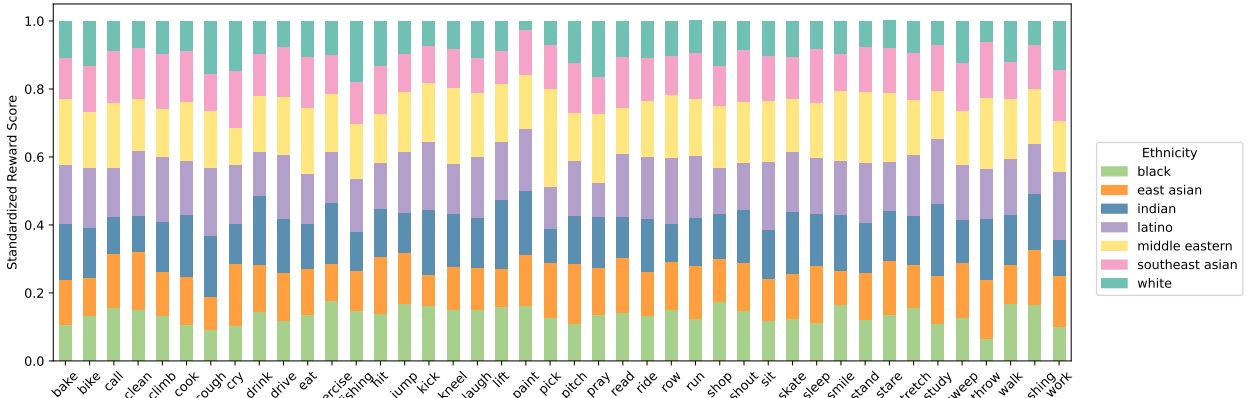

Figure 27: Ethnicity representation bias in **HPSv2.0** across 42 verbs.

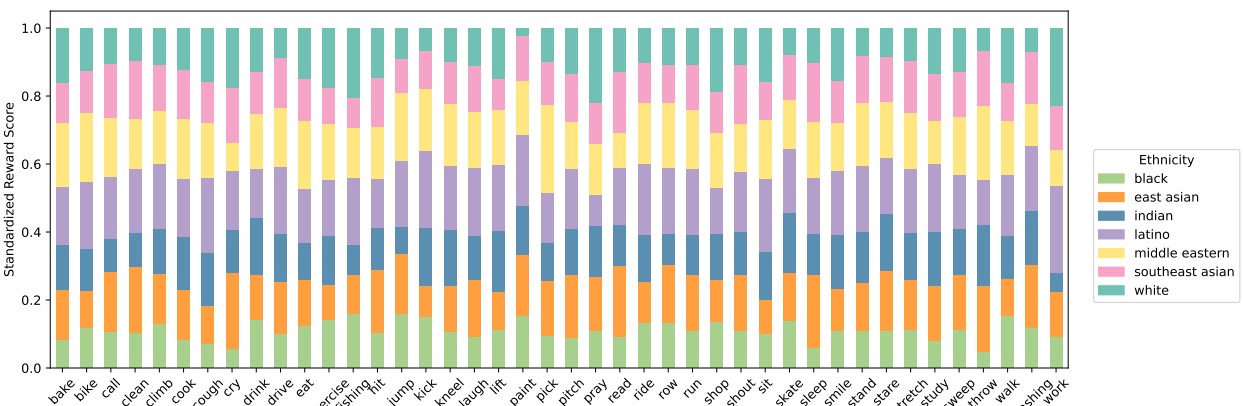

Figure 28: Ethnicity representation bias in **HPSv2.1** across 42 verbs.

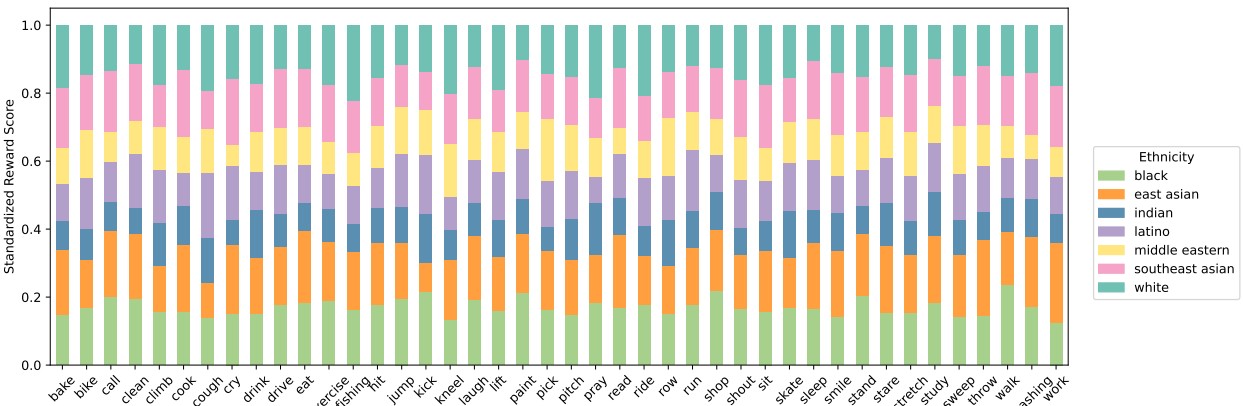

Figure 29: Ethnicity representation bias in **PickScore** across 42 verbs.

# D   Social Biases in Preference Alignment

Building on our analysis of gender and ethnicity biases in image reward models, we examine how preference alignment tuning affects bias in video generation. We fine-tune a Video Consistency Model distilled from VideoCrafter-V2 (VCM-VC2) (Li et al., 2024) using three image-text reward models, HPSv2.0, HPSv2.1, and PickScore, and compare social bias distributions before and after tuning to assess how each reward model shapes identity representation. Following the T2V-Turbo-V1 training protocol (Li et al., 2024), we incorporate reward feedback into the Latent Consistency Distillation process (Luo et al., 2023) by using single step video generation. During student model distillation from a pretrained teacher text to video model, we directly optimize the decoded video frames to maximize reward scores from the image-text alignment models, guiding each frame toward representations more aligned with human preferences.

We evaluate aligned video diffusion models using our bias framework (§4). Table 5 reports two metrics: $\text{PBS}_G$ for gender imbalance across ethnic groups, and $\text{RDS}_e$ and SDI for ethnicity representation disparity and overall output diversity.

**Ethnicity-Aware Gender Bias.** We evaluate gender portrayals under the `ethnicity+person` condition using the previously defined $\text{PBS}_G$ metric. A positive $\text{PBS}_G$ score indicates a tendency to depict men more frequently, while a negative score suggests a preference for women. The base model, VCM-VC2, demonstrates a strong man bias across all ethnicities, which becomes more pronounced with alignment using HPSv2.0. In contrast, alignment with HPSv2.1 and PickScore significantly reduces $\text{PBS}_G$, indicating a shift toward more balanced or woman-preferred outputs. This change reflects the underlying woman bias present in the HPSv2.1 and PickScore reward models, which steer the model away from the man-dominant bias of the base model. Figures 30 to 37 presents the $\text{PBS}_G$ scores across 42 verbs for each ethnicity group.

**Ethnicity Bias.** Under the `ethnicity-only` condition, we analyze models' representation balance using the previously defined $\text{RDS}_e$ and SDI metrics. Positive values indicate overrepresentation, and negative values indicate underrepresentation. Overall demographic balance is measured using SDI, where higher values reflect more equitable representation. The base model, VCM-VC2, strongly favors White individuals (RDS = 0.6405), while Black, East Asian, and Middle Eastern groups are underrepresented. Alignment with HPSv2.1 reduces some disparities by improving balance for White and Black groups, but significantly decreases Latino representation (RDS = –0.4352) and lowers SDI, indicating reduced diversity. In contrast, PickScore achieves the highest SDI and produces more balanced representation across most ethnic groups, resulting in the most demographically equitable outputs. Figure 38 shows the ethnicity bias across 42 verbs.

**Statistical Analysis of Aligned Video Models.** We analyze the impact of alignment tuning on VCM-VC2 by comparing the base model against versions aligned with different reward models (HPSv2.0, HPSv2.1, and PickScore), with paired statistical results reported in Table 12. For each metric, we compute $\Delta = \mu_{\text{aligned}} - \mu_{\text{base}}$ and evaluate statistical reliability using $p$-values alongside Cohen's $d$ to quantify effect magnitude. Overall, alignment tuning induces large and highly systematic shifts in ethnicity-aware gender bias, with both the direction and magnitude of change strongly dependent on the reward model. Alignment with **HPSv2.0** results in a consistent *increase* in male bias, with 8/11 $\text{PBS}_G$ metrics showing statistically significant changes; in particular, the averaged $\text{PBS}_G$ increases by +0.108 ($p = 0.0003$, $d = 0.61$), with medium-to-large effect sizes observed across most ethnic groups. In contrast, alignment with **HPSv2.1** produces a pronounced *reduction* in gender bias, with 11/12 metrics reaching statistical significance and uniformly large effect sizes; the averaged $\text{PBS}_G$ decreases sharply by $-0.577$ ($p < 0.001$, $d = 1.72$), and all ethnicity-specific $\text{PBS}_G$ scores exhibit large, highly significant declines. Similarly, **PickScore**-based alignment substantially reduces gender bias, with 8/14 metrics significant and a large decrease in averaged $\text{PBS}_G$ ($\Delta = -0.432$, $p < 0.001$, $d = 1.39$), though the magnitude of reduction is more heterogeneous across ethnic groups. Changes in ethnic representation ($\text{RDS}_e$) further reveal reward-specific redistribution effects—most notably, large reductions in White overrepresentation and Middle Eastern deviation under HPSv2.0 and HPSv2.1—while overall diversity as measured by SDI remains statistically unchanged across all alignment variants. Taken together, these results demonstrate that alignment tuning is a powerful intervention that can either amplify or substantially mitigate gender bias in video generation models, with most effects achieving strong statistical significance and large practical impact, underscoring the central role of reward model choice in shaping downstream social bias outcomes.

| Metric | VCM-VC2 | + HPSv2.0 | | | | | | + HPSv2.1 | | | | | | + PickScore | | | | | |
|---|---|---|---|---|---|---|---|---|---|---|---|---|---|---|---|---|---|---|---|
| | Base | Mean | Δ | p-val | d | Eff. | Sig. | Mean | Δ | p-val | d | Eff. | Sig. | Mean | Δ | p-val | d | Eff. | Sig. |
| *Ethnicity-Aware Gender Bias: Proportion Bias Score for Gender (PBS$_G$)* | | | | | | | | | | | | | | | | | | | |
| Average | 0.803 | 0.912 | +0.108 | 0.000 | -0.61 | M-L | Yes*** | 0.227 | -0.577 | 0.000 | 1.72 | L | Yes*** | 0.371 | -0.432 | 0.000 | 1.39 | L | Yes*** |
| White | 0.792 | 0.967 | +0.174 | 0.000 | -0.67 | M-L | Yes*** | 0.132 | -0.660 | 0.000 | 1.61 | L | Yes*** | 0.343 | -0.450 | 0.000 | 1.16 | L | Yes*** |
| Black | 0.776 | 0.921 | +0.146 | 0.000 | -0.67 | M-L | Yes*** | 0.238 | -0.538 | 0.000 | 1.50 | L | Yes*** | 0.336 | -0.440 | 0.000 | 1.25 | L | Yes*** |
| Indian | 0.863 | 0.900 | +0.037 | 0.281 | -0.17 | S | No | 0.145 | -0.718 | 0.000 | 2.04 | L | Yes*** | 0.250 | -0.613 | 0.000 | 1.48 | L | Yes*** |
| East Asian | 0.712 | 0.850 | +0.139 | 0.010 | -0.42 | S-M | Yes* | 0.157 | -0.554 | 0.000 | 1.26 | L | Yes*** | 0.145 | -0.567 | 0.000 | 1.24 | L | Yes*** |
| Southeast Asian | 0.794 | 0.921 | +0.127 | 0.010 | -0.42 | S-M | Yes** | 0.362 | -0.433 | 0.000 | 0.97 | L | Yes*** | 0.455 | -0.340 | 0.000 | 0.76 | M-L | Yes*** |
| Middle Eastern | 0.807 | 0.855 | +0.048 | 0.230 | -0.19 | S | No | 0.179 | -0.629 | 0.000 | 1.89 | L | Yes*** | 0.351 | -0.456 | 0.000 | 1.20 | L | Yes*** |
| Latino | 0.869 | 0.967 | +0.098 | 0.000 | -0.83 | L | Yes*** | 0.374 | -0.495 | 0.000 | 0.94 | L | Yes*** | 0.719 | -0.150 | 0.009 | 0.42 | S-M | Yes** |
| *Ethnicity Bias: Representation Deviation Score for Ethnicity (RDS$_e$)* | | | | | | | | | | | | | | | | | | | |
| Black | -0.188 | — | — | — | — | — | — | — | — | — | — | — | — | -0.191 | -0.002 | 0.660 | 0.07 | S | No |
| East Asian | -0.188 | — | — | — | — | — | — | — | — | — | — | — | — | -0.198 | -0.009 | 0.160 | 0.22 | S-M | No |
| Latino | -0.195 | — | — | — | — | — | — | -0.331 | -0.136 | 0.000 | 5.04 | L | Yes*** | -0.188 | +0.007 | 0.183 | -0.21 | S-M | No |
| Middle Eastern | -0.117 | -0.367 | -0.250 | 0.000 | 1.26 | L | Yes*** | -0.264 | -0.148 | 0.000 | 1.05 | L | Yes*** | -0.110 | +0.007 | 0.740 | -0.05 | S | No |
| White | 0.688 | 0.367 | -0.321 | 0.000 | 1.55 | L | Yes*** | 0.595 | -0.093 | 0.001 | 0.57 | M-L | Yes*** | 0.686 | -0.002 | 0.923 | 0.02 | S | No |
| *Ethnicity Bias: Simpson's Diversity Index (SDI)* | | | | | | | | | | | | | | | | | | | |
| Overall | 0.148 | 0.126 | -0.022 | 0.520 | 0.10 | S | No | 0.089 | -0.059 | 0.064 | 0.29 | S-M | No | 0.145 | -0.003 | 0.923 | 0.02 | S | No |

*Eff. Size: S=Small, S-M=Small-Medium, M-L=Medium-Large, L=Large.*

Table 12: Statistical impact of alignment tuning on bias metrics in the VCM-VC2 video generation model. We compare the base VCM-VC2 model against versions aligned with different reward models (HPSv2.0, HPSv2.1, and PickScore), reporting mean values, alignment-induced differences $\Delta = \mu_{\text{aligned}} - \mu_{\text{base}}$, paired $p$-values, and Cohen's $d$ effect sizes. Statistical significance is assessed using paired tests across the same set of verbs, with $p$-values indicating the reliability of observed differences and Cohen's $d$ quantifying standardized effect size. Significance levels are denoted as ***$p < 0.001$, **$p < 0.01$, and *$p < 0.05$. Effect sizes are categorized as Small ($|d| < 0.2$), Small–Medium ($0.2 \leq |d| < 0.5$), Medium–Large ($0.5 \leq |d| < 0.8$), and Large ($|d| \geq 0.8$). PBS$_G$ measures ethnicity-aware gender bias, RDS$_e$ captures deviation from uniform ethnic representation, and SDI reflects overall ethnic diversity (higher is more balanced). Missing entries (—) indicate cases where metrics are undefined due to zero observable instances for the corresponding subgroup.

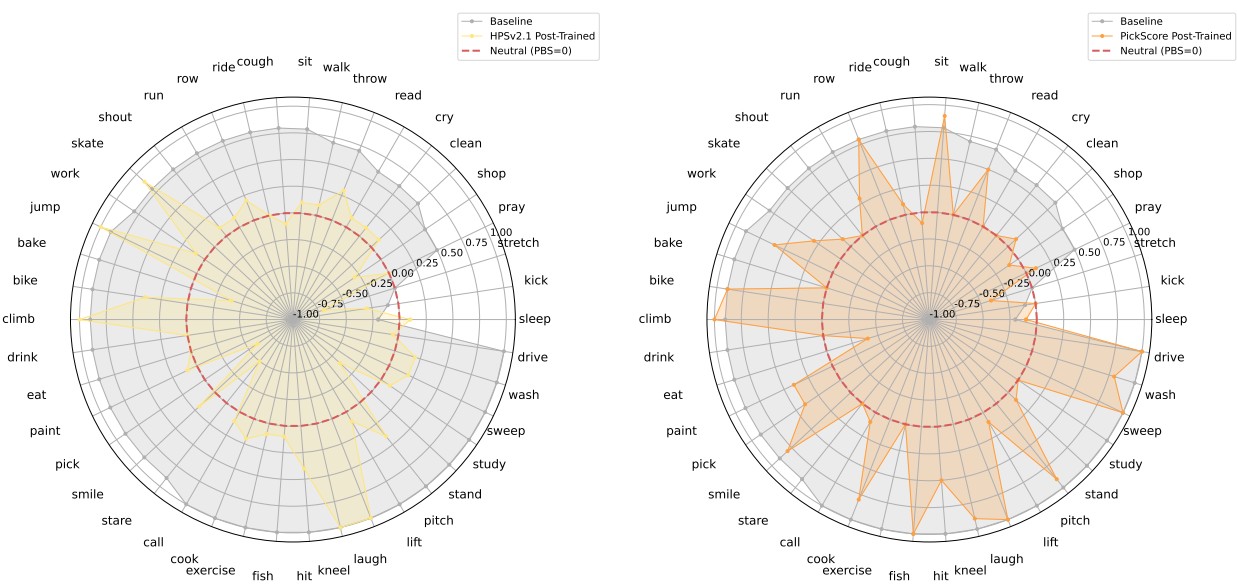

(a) VCM-VC2 (gray) vs. VCM-VC2+**HPSv2.1** (yellow)   (b) VCM-VC2 (gray) vs. VCM-VC2+**PickScore** (orange)

Figure 30: Ethnicity-aware gender bias for the **White** subgroup across 42 verbs. We compare the unaligned baseline video diffusion model VCM-VC2 with its preference-aligned variants trained using HPSv2.1 and PickScore rewards. Scores are plotted relative to a zero-centered neutral zone; values *outside* this region indicate systematic gender bias, with *positive* values denoting man-preference (+) and *negative* values denoting woman-preference (−).

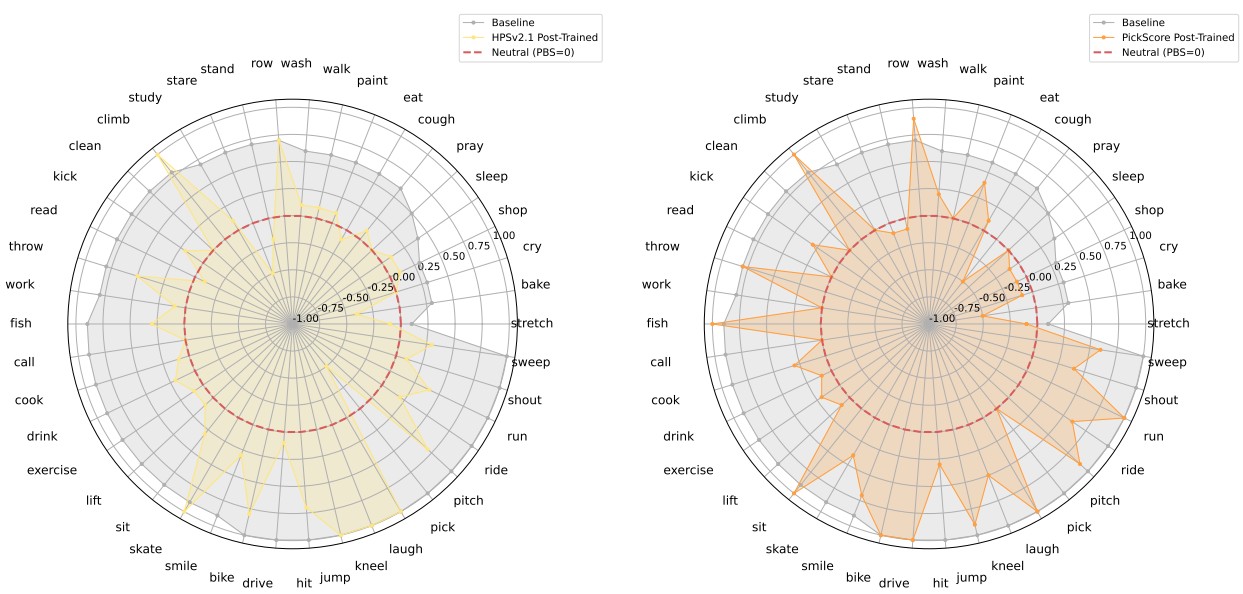

(a) VCM-VC2 (gray) vs. VCM-VC2+**HPSv2.1** (yellow)  (b) VCM-VC2 (gray) vs. VCM-VC2+**PickScore** (orange)

Figure 31: Ethnicity-aware gender bias for the **Black** subgroup across 42 verbs. We compare the unaligned baseline video diffusion model VCM-VC2 with its preference-aligned variants trained using HPSv2.1 and PickScore rewards. Scores are plotted relative to a zero-centered neutral zone; values *outside* this region indicate systematic gender bias, with *positive* values denoting man-preference (+) and *negative* values denoting woman-preference (−).

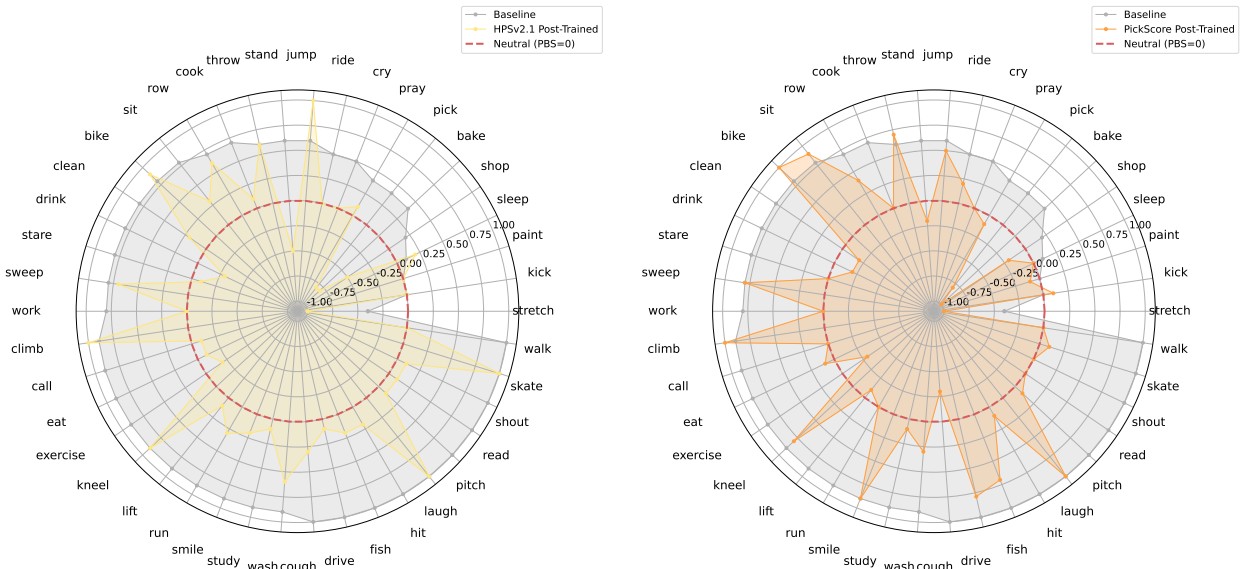

(a) VCM-VC2 (gray) vs. VCM-VC2+**HPSv2.1** (yellow)  (b) VCM-VC2 (gray) vs. VCM-VC2+**PickScore** (orange)

Figure 32: Ethnicity-aware gender bias for the **East Asian** subgroup across 42 verbs. We compare the unaligned baseline video diffusion model VCM-VC2 with its preference-aligned variants trained using HPSv2.1 and PickScore rewards. Scores are plotted relative to a zero-centered neutral zone; values *outside* this region indicate systematic gender bias, with *positive* values denoting man-preference (+) and *negative* values denoting woman-preference (−).

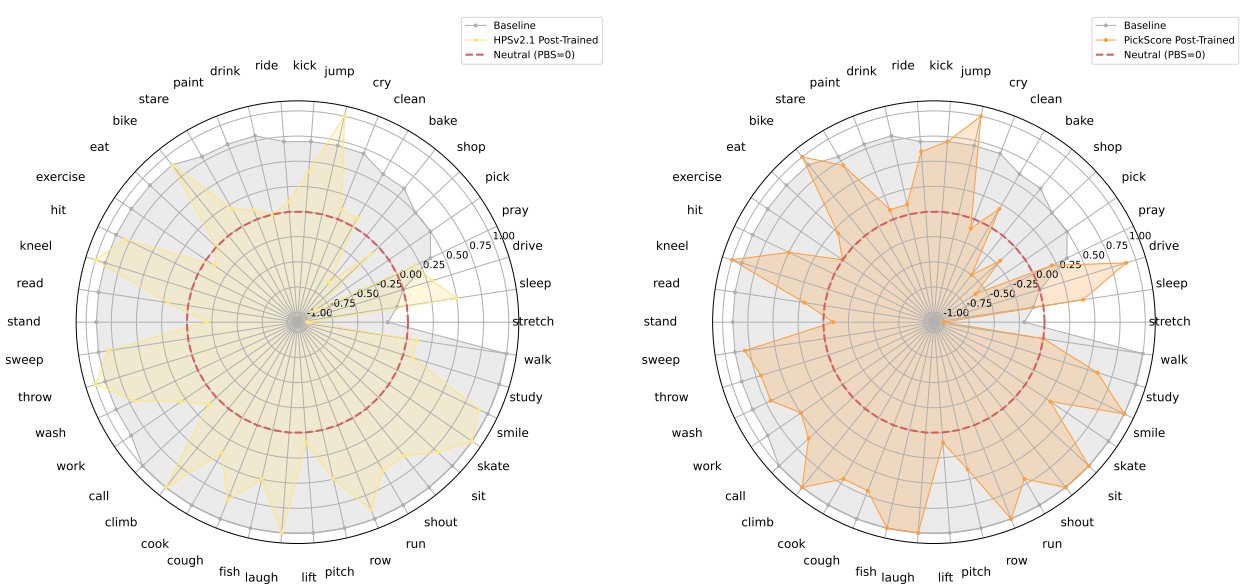

(a) VCM-VC2 (gray) vs. VCM-VC2+**HPSv2.1** (yellow)    (b) VCM-VC2 (gray) vs. VCM-VC2+**PickScore** (orange)

Figure 33: Ethnicity-aware gender bias for the **Southeast Asian** subgroup across 42 verbs. We compare the unaligned baseline video diffusion model VCM-VC2 with its preference-aligned variants trained using HPSv2.1 and PickScore rewards. Scores are plotted relative to a zero-centered neutral zone; values *outside* this region indicate systematic gender bias, with *positive* values denoting man-preference (+) and *negative* values denoting woman-preference (−).

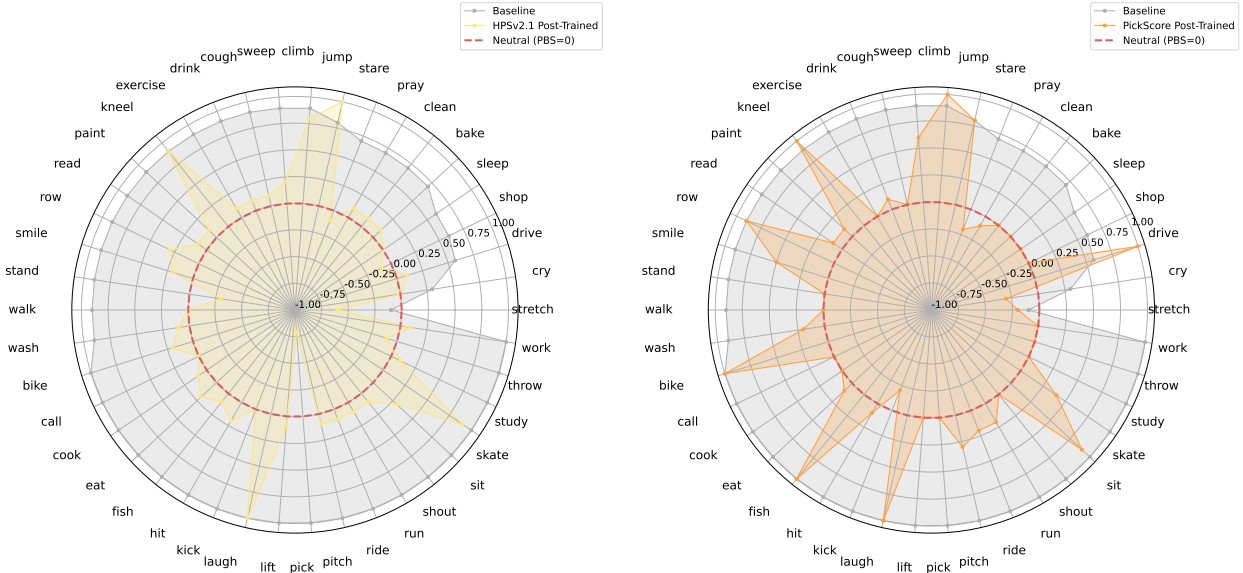

(a) VCM-VC2 (gray) vs. VCM-VC2+**HPSv2.1** (yellow)    (b) VCM-VC2 (gray) vs. VCM-VC2+**PickScore** (orange)

Figure 34: Ethnicity-aware gender bias for the **Indian** subgroup across 42 verbs. We compare the unaligned baseline video diffusion model VCM-VC2 with its preference-aligned variants trained using HPSv2.1 and PickScore rewards. Scores are plotted relative to a zero-centered neutral zone; values *outside* this region indicate systematic gender bias, with *positive* values denoting man-preference (+) and *negative* values denoting woman-preference (−).

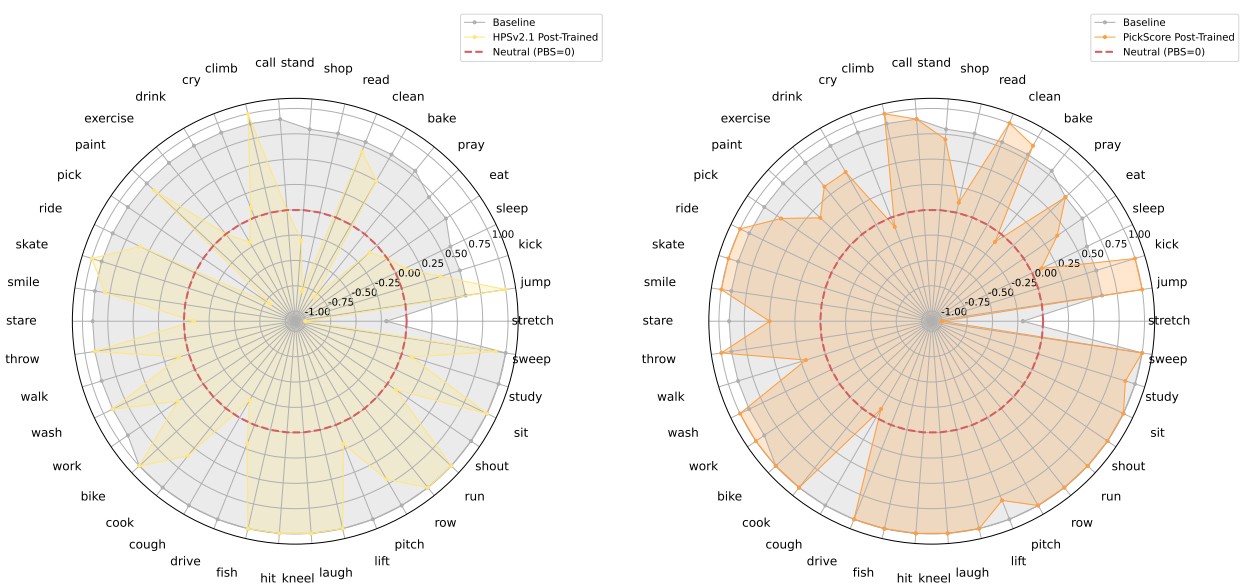

(a) VCM-VC2 (gray) vs. VCM-VC2+**HPSv2.1** (yellow)  (b) VCM-VC2 (gray) vs. VCM-VC2+**PickScore** (orange)

Figure 35: Ethnicity-aware gender bias for the **Latino** subgroup across 42 verbs. We compare the unaligned baseline video diffusion model VCM-VC2 with its preference-aligned variants trained using HPSv2.1 and PickScore rewards. Scores are plotted relative to a zero-centered neutral zone; values *outside* this region indicate systematic gender bias, with *positive* values denoting man-preference (+) and *negative* values denoting woman-preference (−).

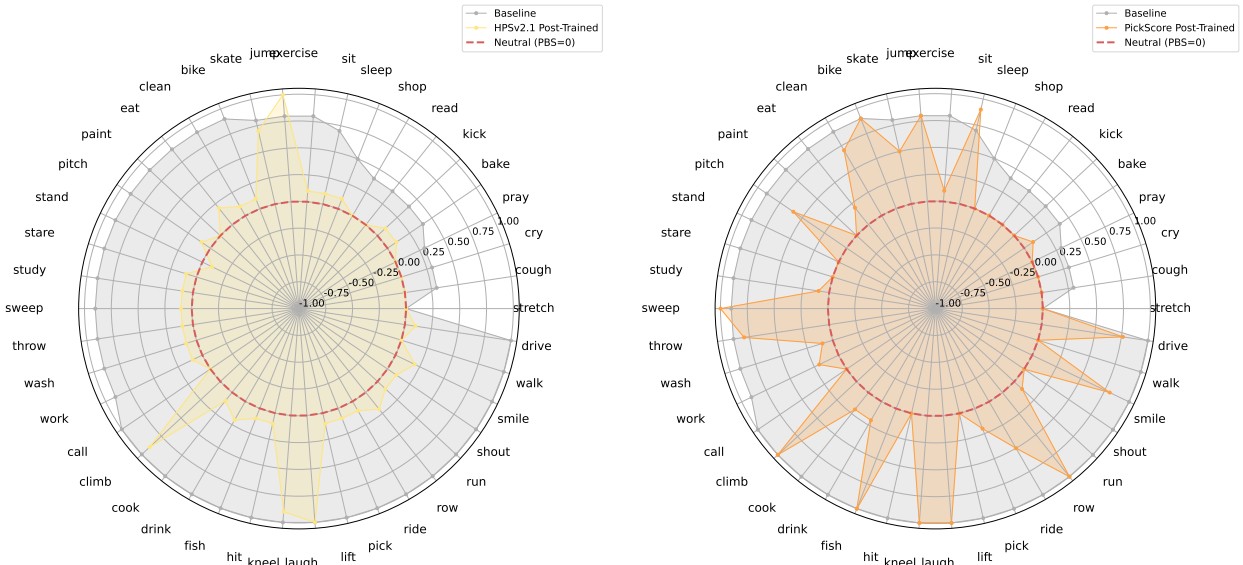

(a) VCM-VC2 (gray) vs. VCM-VC2+**HPSv2.1** (yellow)  (b) VCM-VC2 (gray) vs. VCM-VC2+**PickScore** (orange)

Figure 36: Ethnicity-aware gender bias for the **Middle Eastern** subgroup across 42 verbs. We compare the unaligned baseline video diffusion model VCM-VC2 with its preference-aligned variants trained using HPSv2.1 and PickScore rewards. Scores are plotted relative to a zero-centered neutral zone; values *outside* this region indicate systematic gender bias, with *positive* values denoting man-preference (+) and *negative* values denoting woman-preference (−).

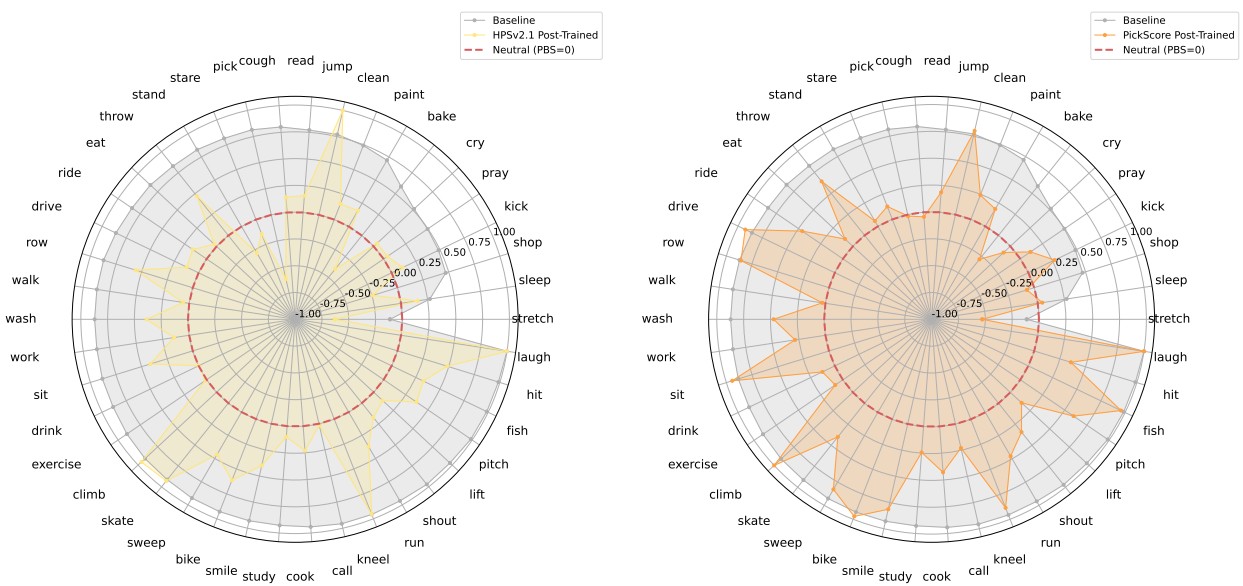

(a) VCM-VC2 (gray) vs. VCM-VC2+**HPSv2.1** (yellow)  (b) VCM-VC2 (gray) vs. VCM-VC2+**PickScore** (orange)

Figure 37: Ethnicity-aware gender bias (**averaged**) for all subgroups across 42 verbs. We compare the unaligned baseline video diffusion model VCM-VC2 with its preference-aligned variants trained using HPSv2.1 and PickScore rewards. Scores are plotted relative to a zero-centered neutral zone; values *outside* this region indicate systematic gender bias, with *positive* values denoting man-preference (+) and *negative* values denoting woman-preference (−).

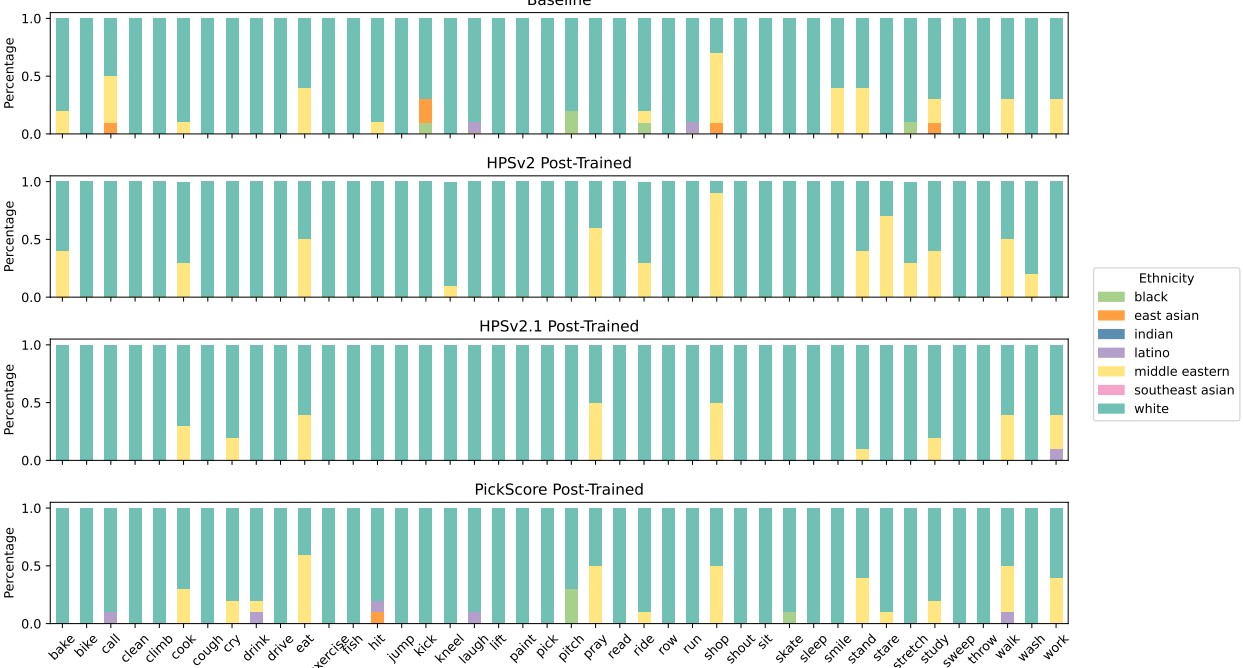

Figure 38: Ethnicity bias distribution across models. Shown are the unaligned baseline video diffusion model VCM-VC2 and its preference-aligned variants trained with HPSv2.0, HPSv2.1, and PickScore reward signals.

# E  Controllable Preference Modeling for Video Diffusion Models

Building on prior findings, we observe that reward models trained on imbalanced image preference datasets inherit and amplify social biases. These biases are then reflected in video diffusion models fine-tuned with such reward signals, often leading to unbalanced outputs. In this section, we explore whether manipulating the distribution of social attributes in image datasets allows for controllable bias in reward models, enabling video models to produce more equitable outputs (Sheng et al., 2020).

## E.1  Image Reward Dataset Construction

Building on the generated images from §5.1, we construct two case-specific reward datasets: one with a man-preferred bias and the other with a woman-preferred bias. The man-preferred dataset is designed to steer both the reward model and the downstream diffusion model toward favoring man representations. Conversely, the woman-preferred dataset encourages a shift toward woman representations. Notably, when applied to a base video diffusion model that exhibits a man-preference bias, the woman-preferred dataset can serve as an effective counterbalance, enabling the training of models with more equitable gender representation.

More specifically, we construct preference pairs using images from the `Gender+Ethnicity` dataset by selecting two images that depict the same action and belong to the same ethnicity group, one featuring a man and the other a woman (for example, images M-1 and W-1 in Figure 5). These image pairs are used to train reward models with prompts of the form: "A/An `[ethnicity]` person is `[action]`-ing `[context]`." For the man-preference dataset, we assign a reward score of 1 to the image with a man character and 0 to the image with a woman character. In contrast, f or the woman-preference dataset, we assign a reward score of 0 to the image with a man character and 1 to the image with a woman character. This process results in 2.94 million preference pairs in each dataset, calculated as 42 verbs multiplied by seven ethnicity groups, with 100 male and 100 female images per group. To improve the representation of no-face content, we additionally incorporate 537,660 face-free image pairs from HPDv2, which enhances balance in our proposed reward datasets.

## E.2  Image Reward Model Development & Alignment Tuning

Leveraging the man-preferred and woman-preferred image datasets introduced in §7.1, we fine-tune two reward models on top of a pre-trained CLIP vision encoder: the Man-Preferred Reward Model ($RM_M$) and the Woman-Preferred Reward Model ($RM_W$). Each model is trained to reflect gender-specific preferences based on its respective dataset. As shown in Table 7, $RM_M$ consistently assigns greater $PBS_G$ scores across all demographic groups, indicating a strong alignment with man-preferred representations. In contrast, $RM_W$ exhibits an opposite trend, systematically favoring woman-preferred content. The clear divergence between these models highlights the effectiveness of reward tuning in capturing and reinforcing gendered preferences.

Building on our earlier reward model training, we applied $RM_M$ and $RM_W$ to guide alignment tuning of a base video diffusion model using the same preference-driven training strategy. These reward signals enabled the generation of two distinct variants: one aligned with man-preferred content and the other with woman-preferred content. As shown in Table 8, alignment with $RM_M$ led to consistently greater $PBS_G$ scores across all demographic groups, reinforcing man-preference bias. Conversely, alignment with $RM_W$ resulted in substantially smaller scores, indicating a strong shift toward woman-preference bias. These results confirm that our controllable preference modeling approach can effectively modulate gender bias in video generation, offering a flexible mechanism to either amplify or reduce specific social tendencies in model outputs. Figures 39 to 43 presents the $PBS_G$ scores across 42 verbs for each ethnicity group.

## E.3  Verbs Correlation Analysis

We analyze the changes in the reward model preference for 42 events and the bias of the video generation model before and after post-training, using the training results from §7. In Figure 44, the horizontal axis represents the reward model preference ($PBS_G$), and the vertical axis represents the change in the video generation model's bias before and after post-training ($\Delta PBS_G$). In Figure 45 and Figure 46, the horizontal axis represents the event, and the vertical axis represents the change in the video generation model's bias before and after post-training ($\Delta PBS_G$) divided by the reward model preference ($PBS_G$). This ratio indicates

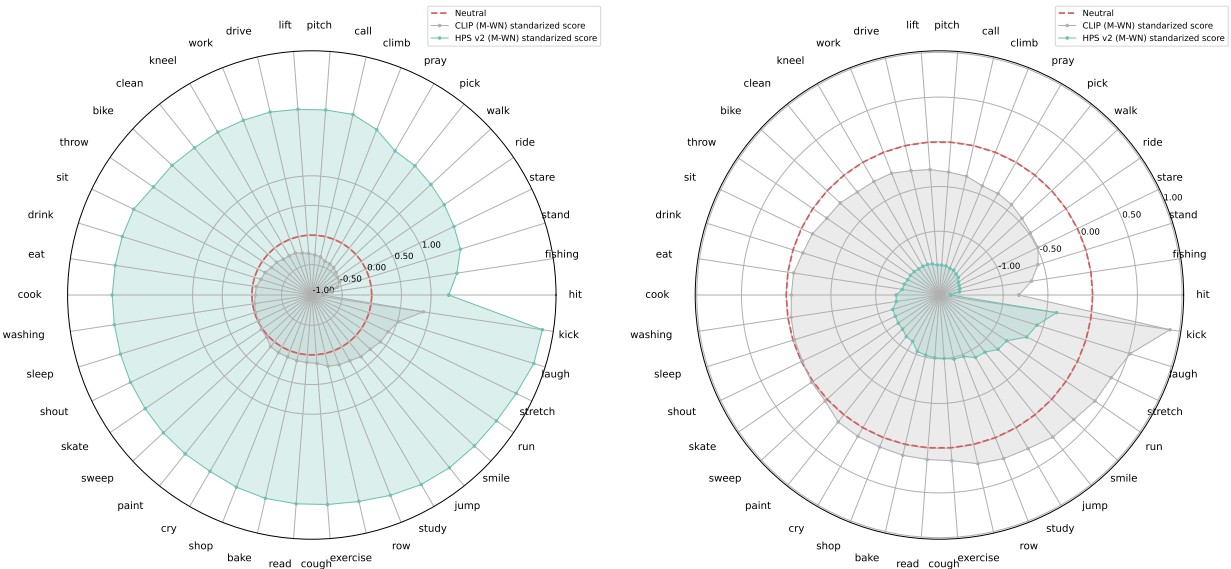

(a) We compare the baseline image reward model CLIP (gray) with its **man-preferred**, preference-aligned variants ($RM_M$, teal).

(b) We compare the baseline image reward model CLIP (gray) with its **woman-preferred**, preference-aligned variants ($RM_W$, teal).

Figure 39: **Averaged** ethnicity-aware gender bias across all subgroups and 42 verbs. Scores are plotted relative to a zero-centered neutral zone; values *outside* this region indicate systematic gender bias, with *positive* values denoting man-preference (+) and *negative* values denoting woman-preference (−).

the sensitivity of a particular event to the bias during post-training. We have arranged the events in the figure from left to right in ascending order of the vertical axis values; events further to the right are more sensitive.

## F  Reward Model Training and Inference Details

For both the training and inference of the reward model, we largely follow the settings outlined in (Wu et al., 2023). We also utilize the HPSv2 codebase available at `https://github.com/tgxs002/HPSv2` for these processes. For **training**, we employ a batch size of 16 and the AdamW optimizer. The man-preferred and woman-preferred datasets that we construct are adapted to the data loading format specified in the HPSv2 codebase (`https://github.com/tgxs002/HPSv2`). Ultimately, we train the reward models for man-preferred and woman-preferred data for 1 epochs (equivalent to 23000 steps), with no data repetition within each step. The model training is initialized from a CLIP checkpoint. For **inference**, we use the CLIP score as the inference score for the reward models.

## G  Video Model Post-Training and Inference Details

For **post-training** during alignment tuning, we used the T2V-Turbo-V1 codebase (Li et al., 2024), available at `https://github.com/Ji4chenLi/t2v-turbo`. A reward model loss scale of 1 was applied. The video model is jointly trained with both the reward model loss and the diffusion loss over 200 steps, using data sampled from the WebVid-10M dataset. For **inference**, we also utilize the same T2V-Turbo-V1 codebase. Each inference setting is run 10 times with different random seed to ensure consistency and robustness of the results.

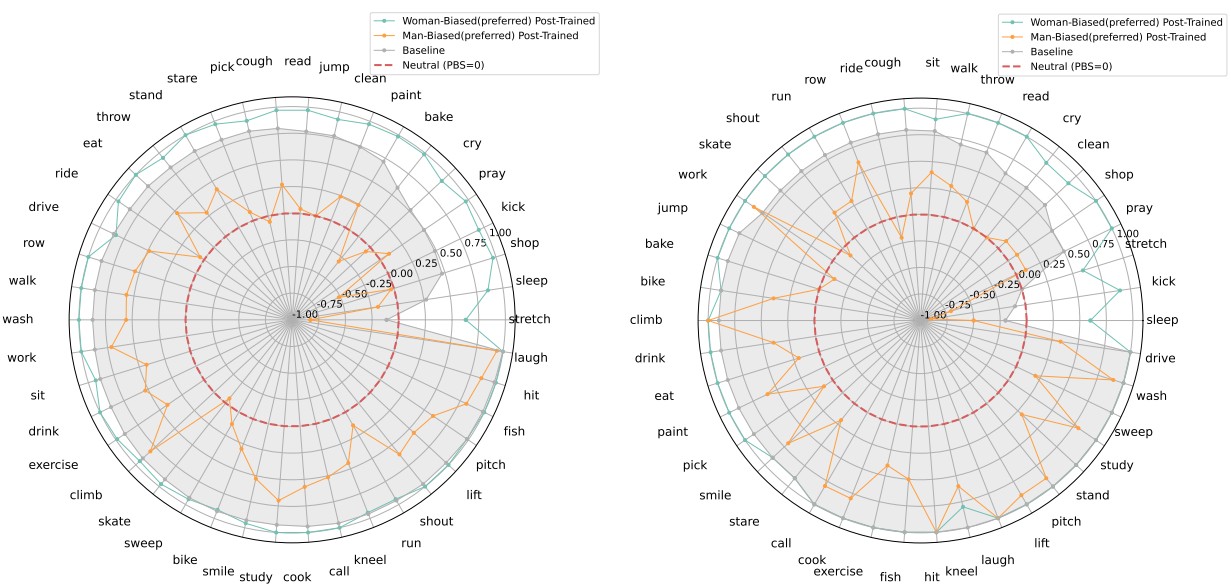

(a) Ethnicity-aware gender bias (**averaged**) of man-preferred (orange) and woman-preferred (teal) post-trained video generation model by reward model $\text{RM}_M$ and $\text{RM}_W$.

(b) Ethnicity-aware gender bias (**White**) of man-preferred (orange) and woman-preferred (teal) post-trained video generation model by reward model $\text{RM}_M$ and $\text{RM}_W$.

Figure 40: Ethnicity-aware gender bias of man-preferred and woman-preferred post-trained video generation model by reward model $\text{RM}_M$ and $\text{RM}_W$.

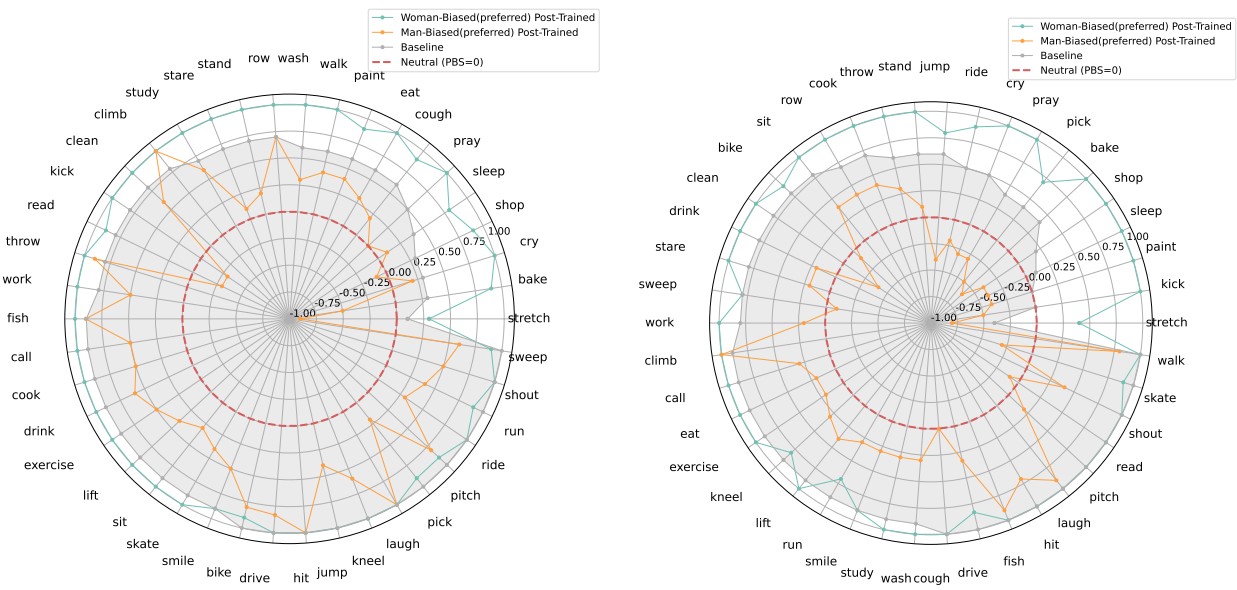

(a) Ethnicity-aware gender bias (**Black**) of man-preferred (orange) and woman-preferred (teal) post-trained video generation model by reward model $\text{RM}_M$ and $\text{RM}_W$.

(b) Ethnicity-aware gender bias (**East Asian**) of man-preferred (orange) and woman-preferred (teal) post-trained video generation model by reward model $\text{RM}_M$ and $\text{RM}_W$.

Figure 41: Ethnicity-aware gender bias of man-preferred and woman-preferred post-trained video generation model by reward model $\text{RM}_M$ and $\text{RM}_W$.

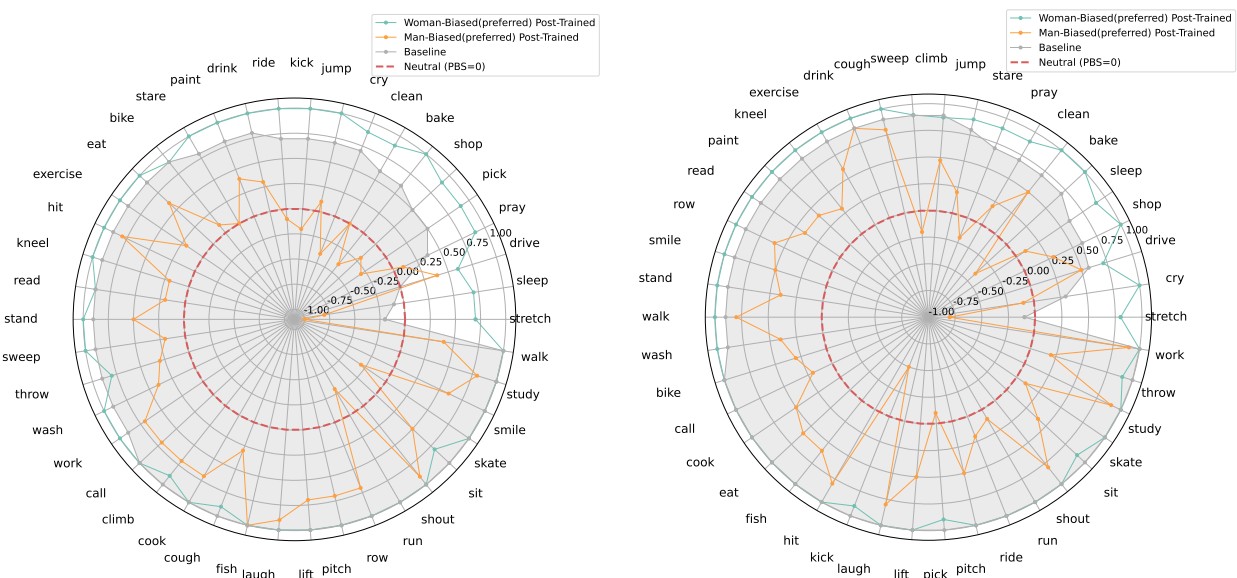

(a) Ethnicity-aware gender bias (**SE Asian**) of man-preferred (orange) and woman-preferred (teal) post-trained video generation model by reward model $\text{RM}_M$ and $\text{RM}_W$.

(b) Ethnicity-aware gender bias (**Indian**) of man-preferred (orange) and woman-preferred (teal) post-trained video generation model by reward model $\text{RM}_M$ and $\text{RM}_W$.

Figure 42: Ethnicity-aware gender bias of man-preferred and woman-preferred post-trained video generation model by reward model $\text{RM}_M$ and $\text{RM}_W$.

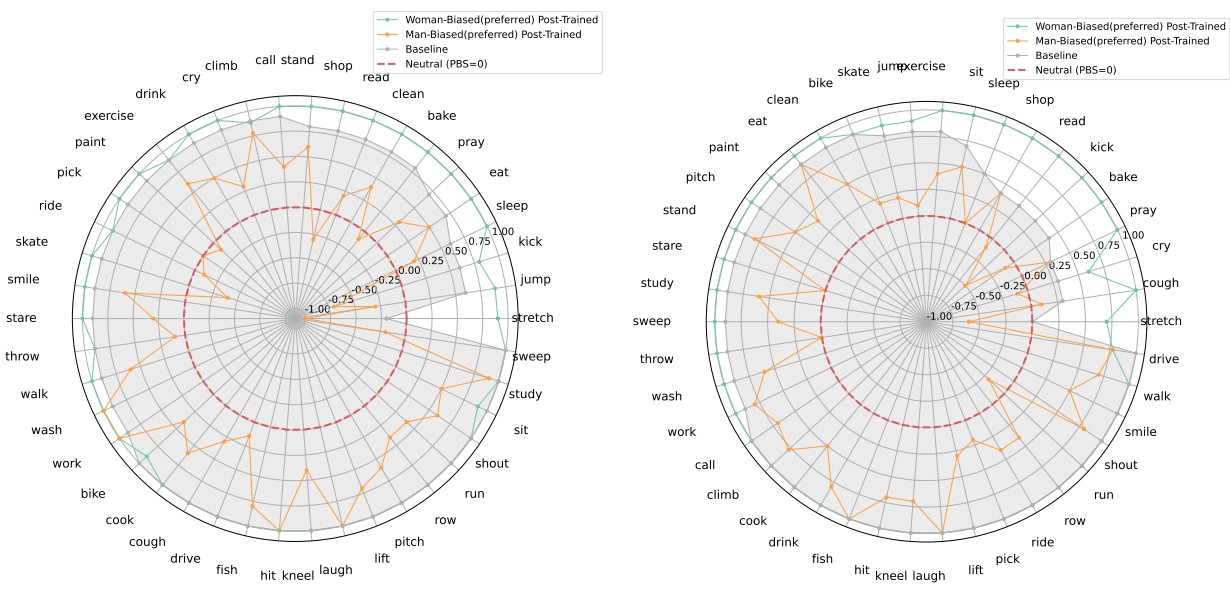

(a) Ethnicity-aware gender bias (**Latino**) of man-preferred (orange) and woman-preferred (teal) post-trained video generation model by reward model $\text{RM}_M$ and $\text{RM}_W$.

(b) Ethnicity-aware gender bias (**Mid Eastern**) of man-preferred (orange) and woman-preferred (teal) post-trained video generation model by reward model $\text{RM}_M$ and $\text{RM}_W$.

Figure 43: Ethnicity-aware gender bias of man-preferred and woman-preferred post-trained video generation model by reward model $\text{RM}_M$ and $\text{RM}_W$.

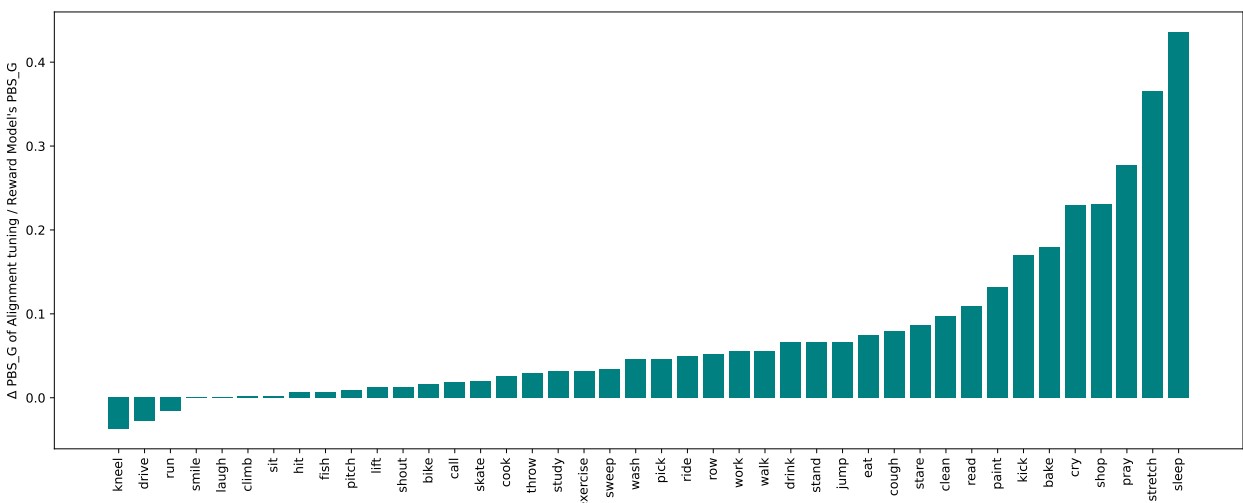

Figure 44: $\Delta$ PBS$_G$ of video generation model before and after alignment tuning by RM$_M$ and RM$_W$. Results are broken down into verbs. Figure 45 and Figure 46 are based on this figure.

Figure 45: Sensitive verbs in man-preferred post-training.

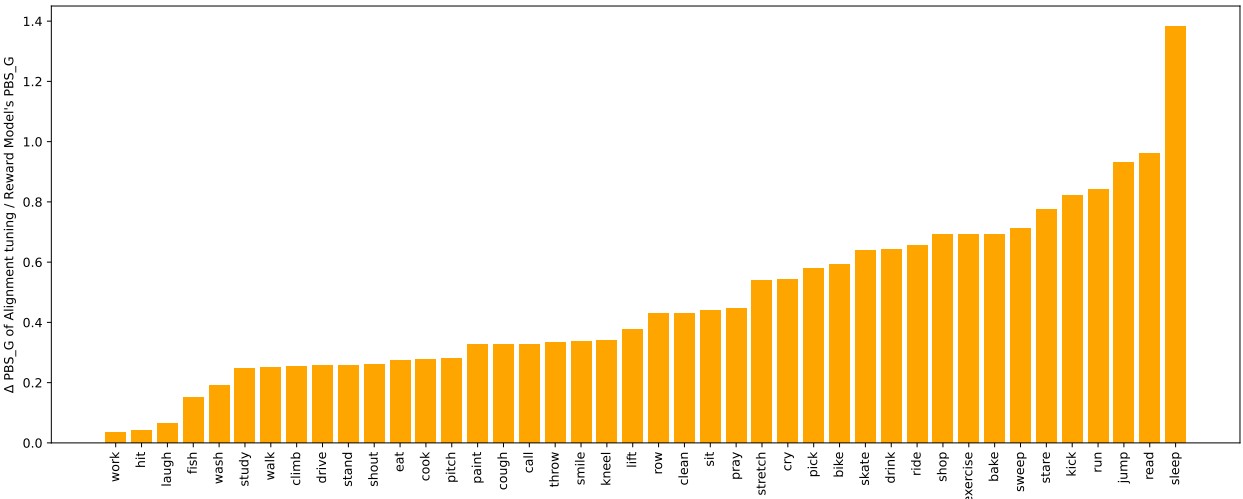

Figure 46: Sensitive verbs in woman-preferred post-training.

