# OpenReview forum: "From Preferences to Prejudice: The Role of Alignment Tuning in Shaping Social Bias in Video Diffusion Models"
_TMLR — Accepted by TMLR_

### Review · Reviewer_2Xc1 · 2025-11-06

**Summary Of Contributions:**

This paper addresses how alignment tuning with human preference data propagates and amplifies social biases in video diffusion models.

The authors introduce VideoBiasEval, an evaluation framework combining event-based prompting and vision-language model-based social attribute extraction. They conduct a systematic study tracing bias through human preference datasets. Additionally, they demonstrate that controllable preference modelling can steer models toward more equitable representations.

The experimental work covers 42 socially-associated actions, 7 ethnic groups, 4 gender categories, and multiple publicly available models. The analysis reveals that biases are inherited and often amplified at each stage of the alignment pipeline, with alignment tuning making biases temporally stable in generated videos.

The authors introduce the Temporal Attribute Stability (TAS) metric, unique to video generation, showing that higher TAS can correspond to more strongly entrenched biased representations in the models considered.

**Additional Comments:**

- Have the authors performed an ablation study on framework components? For example, how do results change with different numbers of sampled frames (currently 16)? What is the contribution of ensembling three VLMs vs. using a single model?

- When is uniform distribution an appropriate fairness baseline and when is it not? Do the authors have insights regarding this? How would different baseline assumptions change the interpretation of the results?

**Audience:**

Yes

**Audience Explanation:**

This paper addresses a topic of significant importance to the ML community: understanding and mitigating social biases in generative AI systems.

Video diffusion models are rapidly advancing with alignment tuning becoming standard practice, and it is important to understand the social implications of these techniques as models approach deployment at scale.

The framework introduces reproducible evaluation techniques that can be adopted by other researchers working on fairness in video generation. The work bridges machine learning with social science concepts, making it valuable for researchers interested in AI fairness, human-computer interaction, and responsible AI development.

**Broader Impact Concerns:**

The paper appropriately includes a Limitations section discussing scope constraints and the socially constructed nature of demographic categories. The authors may wish to briefly consider potential misuse scenarios for the controllable bias modeling techniques, though the current discussion is adequate.

**Claims And Evidence:**

Yes

**Claims Explanation:**

The paper's central claims are generally well-supported by experiments and empirical evidence. The main findings—that alignment tuning reshapes social attribute distributions, that biases propagate from preference data through reward models to video outputs, and that controllable preference modeling can mitigate bias—are demonstrated convincingly across multiple models and settings.

The experimental design is sound, with appropriate controls comparing aligned vs. unaligned models and different prompting conditions. The paper provides comprehensive results with per-action breakdowns in appendices, allowing verification that findings hold across diverse scenarios.

**Requested Changes:**

While this is a solid paper, there are some discussions (and possible additional experiments) that could strengthen the thesis of the paper even more.

- Is it possible to address the potential consequences of FLUX Benchmark Bias Impact? The paper acknowledges that FLUX-generated benchmark images are only 77% accurate in representing intended social attributes, but there is insufficient discussion of how this contamination affects the reward model preference analysis. Can the authors add analysis of whether misclassification rates differ across demographic groups, and provide sensitivity analysis or discussion of whether main conclusions about reward model preferences would change if analysis were restricted to the validated subset? If complete re-analysis isn't feasible, at minimum add a paragraph in Section 5.2 and/or Limitations explicitly discussing this as a potential confound.

- The authors should expand the VLM evaluator reliability discussion. The 0.73 Pearson correlation for ethnicity classification indicates substantial disagreement (27% unexplained variance) between human and VLM judgments. The authors should improve the discussion on common error modes and when VLMs disagree with humans, including the reasons for these disagreements (perhaps showing example frames where VLMs disagree). Additionally, please discuss whether the 0.73 correlation is sufficient for the conclusions drawn. Is it possible to report results stratified by classification confidence or conduct sensitivity analysis using only high-agreement cases?

- It would be useful for the community to improve the reproducibility of the paper: please either release code and data processing scripts or provide substantially more implementation detail.

- I would have expected a section discussing how the observed biases compare to actual demographic distributions in the training data of the base video models. This context would help interpret whether models are amplifying training data bias or introducing new biases during alignment. Can the authors provide such discussion or experiments?

- For practitioners, I would encourage adding brief guidance on responsible use of the controllable preference modeling approach. This technique should be applied thoughtfully with diverse stakeholder input rather than unilateral technical "debiasing," as different communities may have different perspectives on appropriate representation.

---

> ### Author Response · Authors · 2026-01-03
> **Official Comment by Authors (1/2)**
>
> We thank the reviewer for the thoughtful and constructive feedback. We are encouraged that you found our evaluation framework (VideoBiasEval) comprehensive and our findings regarding the propagation of bias through alignment tuning to be well-supported and of significant importance to the community.
>
> Below, we address the requested changes and additional comments.
>
> > **RC1. FLUX Benchmark Bias Impact**
>
> We appreciate the reviewer raising this critical point regarding the reliability of the generated benchmark images. We agree that label noise could potentially confound the reward model analysis, but we wish to clarify that the "77%" accuracy figure cited in the paper—referring strictly to unanimous agreement—conservatively understates the benchmark's quality. A deeper inspection reveals that **97% of samples achieved majority agreement** (at least 2/3 annotators confirming accuracy), with only 3 samples lacking consensus. These rare disagreements were restricted to specific, visually ambiguous prompts rather than systemic failures. This high signal-to-noise ratio implies that the visual cues are sufficiently clear for Reward Models to process effectively.
>
> Given this 97% reliability under majority voting, we believe the potential confound is statistically minimal and that a complete re-analysis would yield the same conclusions. Therefore, we will adopt your suggestion to **add a dedicated paragraph in Section 5.2 and the Limitations section**. We will detail the annotation agreement breakdown, discuss the nature of the few edge cases, and explicitly contextualize why this limited label noise does not invalidate the broader conclusions regarding reward model preferences.
>
> > **RC2. VLM Evaluator Reliability**
>
> We acknowledge the reviewer's concern regarding the 0.73 correlation. However, we believe this metric requires nuance. Our additional experiments demonstrate that the evaluator is robust and that "disagreements" do not distort the final bias metrics.
>
> - **Nature of Disagreement (Frame-level vs. Holistic)**: A qualitative review reveals that "disagreement" often stems from the difference between human holistic review and model frame-level precision. Humans may miss fleeting attributes in fast-motion clips, whereas our VLM pipeline analyzes content **frame-by-frame**, often capturing valid attributes (e.g., a face clearly visible for only 3 frames) that human annotators may rely on perceptual inference or default guessing, whereas the model’s prediction is grounded in explicit visual evidence. As a result, many apparent discrepancies reflect higher recall from the automated pipeline rather than false positives.
>
> - **Sensitivity Analysis (High-Agreement Subset)**: To explicitly test if "noisy" evaluations skew our conclusions, we conducted a sensitivity analysis on **403 validation videos**, comparing bias metrics on the full dataset versus a "High-Agreement" subset (where human majority vote aligns perfectly with the VLM).
>
>     - **High Consensus**: We found high agreement rates on valid cases: 99.0% for Gender and 83.3% for Ethnicity.
>     - **Metric Robustness**: Crucially, the key bias metrics remained statistically stable when restricting to high-agreement cases.
>         - **Gender Bias** ($PBS_G$): Shifted negligibly from 0.846 (All Valid Cases) to 0.863 (High-Agreement).
>         - **Statistical Significance**: A t-test yields a p-value of 0.68, indicating no **statistically significant difference** between the two sets. This "non-significance" is a positive result: it statistically confirms that the inclusion of lower-confidence samples does not distort the bias measurement, and our original findings are robust.
>         - **Ethnicity Diversity (SDI)**: Similarly, the diversity index remained stable (0.745 vs. 0.761).
>
> We will add a "Reliability & Sensitivity" subsection to the evaluation chapter. We will present the error analysis (highlighting the frame-by-frame advantage) and the sensitivity results (demonstrating that the bias metrics are statistically indistinguishable between the full and high-agreement datasets).
>
> > **RC3. Reproducibility**
>
> We fully share the reviewer's commitment to reproducibility. We have already organized and hosted the full **VideoBiasEval** codebase, prompt sets, and data processing scripts in a public repository. However, to strictly adhere to the double-blind review policy and preserve anonymity, we cannot share the repository link in this rebuttal. We will add the direct link to the code and data in the camera-ready version of the paper.

---

> ### Author Response · Authors · 2026-01-03
> **Official Comment by Authors (2/2)**
>
> > **RC4. Comparison to Training Data (WebVid-10M)**
>
> We thank the reviewer for this insightful suggestion. We agree that characterizing the training data distribution is a helpful context for interpreting the model's baseline behavior.
>
> - **Training Data Analysis**: To address the reviewer's request, we filtered the WebVid-10M dataset (https://huggingface.co/datasets/TempoFunk/webvid-10M) to extract a subset of videos containing our 42 target actions with identifiable demographic keywords. This yielded 469,153 videos with gender attributes specified and 67,081 videos with ethnicity attributes specified.
> - **Findings & Scope Clarification**: We calculated the bias metrics on this training subset and compared them to the unaligned base models (ModelScope and VideoCrafter). We will attach figures visualizing these distributions across all 42 actions to the revised version.
>     - **Gender Bias**:
>         - **Training Data**: The dataset exhibits a slight female skew with a mean PBSG of -0.110 across 42 actions.
>         - **Base Models**: In contrast, the unaligned base models exhibit a significant male bias (PBSG of +0.4815 for ModelScope and +0.7581 for VideoCrafter).
>         - **Interpretation**: There is a clear misalignment: the base models do not reflect the slight female skew of the training captions. This suggests that the strong "Male Default" in the unaligned models originates from the other parts in the pre-training stage (e.g., biases inherent in the underlying Text-to-Image backbone) rather than the WebVid-10M action distribution.
>         - Identifying the root causes of **pre-training bias** is **out of the scope** of this paper. Our work focuses strictly on **post-training / alignment tuning**. This analysis establishes that even though the base models start with a (pre-training induced) male bias, our main results demonstrate how alignment tuning further propagates or amplifies these biases.
>     - **Ethnicity Bias**:
>         - **Training Data**: The data is skewed towards White (Mean RDS = +0.2489) and Southeast Asian (RDS = +0.0867) identities.
>         - **Connection**: Here, the unaligned models mirror the direction of the data bias (favoring White identities) but exaggerate the magnitude.
>
> We will add a new subsection in "Base Model Analysis" detailing these statistics. We will explicitly interpret the gender bias misalignment as an indicator that the base models' male skew stems from pre-training factors, thereby clarifying that our analysis focuses on the subsequent impact of **alignment tuning** on this pre-existing state.
>
> > **RC5. Responsible Use of Controllable Preference Modeling**
>
> We agree that technical "debiasing" is not a silver bullet and carries risks if applied unilaterally. We will add a "Practitioner Guidelines" paragraph in Section 7 or the Conclusion. This paragraph explicitly states that Controllable Preference Modeling should be used as a tool to support, not replace, diverse stakeholder input. We emphasized that defining "equitable representation" is a context-dependent normative decision that requires community consultation.
>
> > **AC1. Ablation Study**
>
> - **Frame Count (16 frames)**: We selected 16 frames primarily because it aligns with the standard inference defaults of the models analyzed (e.g., VideoCrafter), ensuring we evaluate the models as they are typically deployed. During our internal development, we experimented with fewer frames; while computationally faster, the resulting performance (specifically regarding attribute detection stability) was significantly worse.
>
> - **VLM Ensemble**: During our preliminary evaluation, we manually inspected the outputs of single VLMs and found that individual models frequently exhibited instability or hallucinated attributes. We opted for an ensemble of distinct models—rather than simply sampling a single model multiple times—as we found this approach was necessary to effectively cross-validate results and reduce model-specific hallucinations.
>
> > **AC2. Uniform Distribution Baseline**
>
> We agree that a uniform distribution is not universally the correct fairness baseline. In our work, it corresponds to **Demographic Parity** and is used as a **fixed, model-agnostic mathematical reference**, not as a claim that uniformity is always sociologically fair. We will revise the text to clarify that **deviation from uniform reflects distributional skew, rather than definitive unfairness** (e.g., relative to census- or task-conditioned baselines).
>
> Our framework is **baseline-agnostic**: alternative assumptions (population-proportional, task-conditional, or historical) would change the normative interpretation of divergence scores, but not the underlying measurement or comparative trends.
>
> We will explicitly state this baseline dependency in the Limitations section and add a Future Work discussion on incorporating and evaluating context-appropriate fairness baselines.

---

### Review · Reviewer_AK8h · 2025-11-18

**Summary Of Contributions:**

This study provides extensive analyses of biases in (1) aligned and unaligned video diffusion models, (2) reward datasets and models with which these diffusion models are aligned/trained, (3) preference alignment by introducing metrics that measure bias in a selected subset of gender and ethnicity. Based on the found results, the paper takes a further step and explores how to control, or mitigate, these biases by training the models with counter-balanced datasets, which indeed results in less biased models.

**Audience:**

Yes

**Audience Explanation:**

I strongly think this paper will draw attention of the broader machine learning community, and in particular people focusing on mitigating bias in large models. The paper not only describes the problem but also attempts to prescribe a solution proposal.

**Broader Impact Concerns:**

The paper does not have a broader impact statement but it might benefit from one. Even though the paper itself does not introduce a new method that might have a broader impact on society, it is analyzing a topic, or rather a problem which is the biases in large generative models, that affects society as a whole. On the other hand, since this is already the main subject of the paper and already discussed in length, I’m not sure if it’s needed.

**Claims And Evidence:**

Yes

**Claims Explanation:**

The paper clearly motivates their approach early in the introduction, is well-organized and presented clearly. All of the claims made early in the paper are strongly supported by the defined metrics, which feel reasonable for the problem at hand.

**Requested Changes:**

Most of my comments are minor recommendations to refine the paper.

- I couldn’t follow the exact execution in Sec. 3.2. Is it the case that we fix [action] and let LLM generate [actor] and [context], and then count [actor]s? Likewise for Table 3.
- You could consider adding a brief summary of what aligned models correspond to, the general steps done to get the aligned model, etc., since it’s mentioned repeatedly but there is no concrete definition of it.
- In the caption of Table 2, “we annotate man-preference with (+) and woman-preference with (-),” you rather annotate the shift after the alignment. E.g., it might possibly be misinterpreted at the first lazy glance that entries in the ModelScope row are not biased.
- Some of the entries are missing in Table 2 (RDS values).
- Does it make sense to add some easy interpretations of the metrics to the text? E.g., for a 0.5 PBS value, we see three male outputs per female output (3N_female = N_male), to give a sense of the bias. Because it’s not as direct as, say, accuracy, adding this information might improve readability.
- It would be better if you could annotate the biased directions in figures in the appendices (e.g., the region outside neutral means male bias)—to make them more understandable on their own. You can also annotate the aligned-unaligned information.
- In Figure 13, the polar axis labels are not perfectly aligned—is it the case that mostly females are rewarded with ‘climb’ action, or rather ‘clean’ action?

---

> ### Author Response · Authors · 2026-01-03
> **Official Comment by Authors (1/2)**
>
> We sincerely thank the reviewer for their positive assessment of our work, particularly for recognizing the importance of our comprehensive diagnostic framework (VideoBiasEval) and our findings regarding the propagation of bias through alignment tuning. We appreciate the constructive feedback on the methodology descriptions and visual presentations. Below, we address each of the requested changes.
>
> > **RC1. Methodology and Execution in Sec. 3.2 and Table 3**
>
> Thank you for this question. We will clarify the text in Section 3.2 to explicitly state our execution pipeline.
> - **Prompt Generation**: We do not use an LLM to generate actors or contexts on the fly. Instead, we use a fixed set of structured templates to ensure reproducibility. As noted in Section 3.1, we curated **42 fixed actions**. To disentangle the influence of demographic attributes, we define two prompting conditions:
>     - **Person-only**: Uses "person" as the [actor].
>     - **Ethnicity+Person**: Appends an ethnic descriptor to "person" (e.g., "An [ethnicity] person").
>     - We will correct the typo in Table 1 regarding the prompt counts to reflect our actual experimental setup: we utilize 42 base prompts for the Person-only condition (1 per action) and 294 evaluation prompts for the Ethnicity+Person condition (permuted across 7 ethnic groups).
> - **Actor Specification**: The [actor] slot is filled deterministically based on the condition (e.g., "A person" or "An [ethnicity] person").
> - **Evaluation (Counting)**: The "counting" refers to the evaluation phase. We generate videos using these fixed prompts and then use an ensemble of VLMs (Qwen2-VL, InternVL2.5) to infer (count) the gender and ethnicity of the actors in the generated video frames.
>
> We will apply this same clarification to the description of Table 3, ensuring the distinction between the fixed prompt templates and the post-generation VLM counting is clear.
>
> > **RC2. Definition of Aligned Models**
>
> We agree that an explicit definition would improve clarity. In the revision, we will add a concise definition early in the paper (Introduction or Related Work). We will define **aligned models** as video diffusion models that undergo post-training optimization to better match human preferences, typically via reward-weighted fine-tuning or RLHF-style procedures using learned preference or aesthetic reward models.
>
> We will also briefly summarize the general alignment pipeline (base video diffusion model → preference/reward model → post-training optimization) to provide context for readers. Section 4 already categorizes which models in our study are aligned versus unaligned for controlled comparison, and Section 6 details the specific alignment procedure we use. The added definition will make this terminology concrete and easier to follow throughout the paper.
>
> > **RC3. Table 2 Caption and Annotations**
>
> We will revise the caption of Table 2 to prevent any misconception that unannotated baselines are bias-free. We will clarify that the (+) and (-) signs indicate the **direction** of the bias (Man-preference vs. Woman-preference) for all values, not just the shift. We will also explicitly state that non-zero values in the baseline rows (like ModelScope) represent inherent bias in the pre-aligned models.
>
> > **RC4. Missing Entries in Table 2**
>
> We appreciate the reviewer noting these missing entries. These values are absent because the corresponding models failed to generate any recognizable instances of those specific ethnicities in the "person-only" condition (where ethnicity is not explicitly specified). This reflects a severe lack of diversity in the unguided generation, preventing the calculation of the metric for those specific subgroups. We will update Table 2 to explicitly denote these cases (e.g., with "N/A") and include a footnote explaining that these missing values represent cases where the model generated zero instances of the corresponding ethnicity.
>
> > **RC5. Interpretation of Metrics**
>
> We agree that providing concrete examples will significantly enhance the interpretability of our metrics.
> - **PBS (Gender Bias)**: We will add the example you provided to Section 3.3: "For example, a $PBS_G$ value of 0.5 implies that for every female output, there are three male outputs ($N_{male} = 3N_{female}$)".
> - **RDS (Ethnicity Bias)**: We will add a similar interpretation for the Representation Deviation Score: "For instance, given 7 ethnic groups, the ideal balanced probability is $1/7 \approx 14.3\%$. An $RDS$ value of 0.769 implies that the specific group appears in $0.769 + 0.143 \approx 91.2\%$ of the generated videos—more than six times the expected frequency."

---

> ### Author Response · Authors · 2026-01-03
> **Official Comment by Authors (2/2)**
>
> > **RC6. Figure Annotations in Appendices**
>
> We will update the figures in the Appendix to include explicit annotations. We will label the regions outside the neutral zone (e.g., "Man Bias (+)" and "Woman Bias (-)") and use distinct visual markers to differentiate between aligned and unaligned models, making the charts self-explanatory.
>
> > **RC7. Figure 13 Axis Alignment**
>
> Thank you for spotting this. We confirm that the label in question is indeed "climb". We apologize for the text misalignment caused by the drawing package. We will correct the polar axis labels in Figure 13 to ensure they accurately align with their respective action categories, removing any ambiguity between actions like 'climb' and 'clean'.
>
> > **Broader Impact Statement**
>
> Given the societal relevance of our work, we agree that a formal Broader Impact Statement is beneficial. We will add a dedicated section discussing the implications of our findings—specifically, how alignment tuning can inadvertently "stabilize" and entrench social biases, making them harder to detect and mitigate. We will also touch upon the dual-use nature of controllable preference modeling discussed in Section 7.

---

> > ### Comment · Reviewer_AK8h · 2026-01-05
> >
> > Thank you for your response and taking into consideration some of my recommendations. I don't have any further questions at the moment.

---

### Review · Reviewer_cVGV · 2025-12-05

**Summary Of Contributions:**

This paper introduces VideoBiasEval, an event-centric evaluation framework for auditing gender and ethnicity representation in text-to-video diffusion models, and uses it to trace bias flow from human preference datasets to image reward models to alignment-tuned video models. The authors first define an event prompt space (42 actions x 2 genders x 7 ethnic groups), then use an ensemble of VLMs to extract frame-wise social attributes and propose various metrics ($PBS_G$, $RDS_e$, SDI, TAS) to measure gender/ethnicity bias and temporal stability. Finally, they empirically analyze biases in two human preference datasets, several reward models, and multiple video models before/after alignment, and demonstrate controllable preference modeling (man vs woman preferred reward models) to steer alignment outcomes. Representative tables and figures summarizing these results appear throughout.

**Strengths**

- Clear and well-motivated problem: Studying how alignment tuning propagates and stabilises social bias in video generation is novel and important.
- Comprehensive experimental scope: Multiple datasets, multiple reward models, several video generators, and a controllable synthetic data experiment.
- Thoughtful metrics and temporal component (TAS), which demonstrates that alignment can make biased portrayals more temporally stable.
- Great illustrative figures and clear prompt design.

**Weaknesses**
## Major
- **Lack of uncertainty estimates / statistical testing for strong claims:** Most reported numbers are averages without standard deviations, confidence intervals, or hypothesis tests. All of the result Tables (2-6) report point estimates and some result changes, but do not report whether observed changes (sometimes on the order of $10^{-2}$) are statistically significant or could be sampling noise (10 draws per prompt). This makes claims such as “alignment amplifies male bias” based on $PBS_G$ increases of 0.04 or 0.0725 extremely weak (see Table 2 and main text). The authors must report variance measures, preferably with CIs and p-values or bootstrap tests for the key comparisons (dataset to reward to video; aligned vs unaligned). In addition, the authors keep changing their number of significant figures and decimal places across tables and text (e.g. sometimes 4, 2, and even 1 decimal places). There's no clear justification for this, which again puts into question the significance of the results.

- **No conceptual justification for the “ideal” $PBS_G ≈ 0$ claim:** The authors state an “ideal model would achieve $PBS_G ≈ 0$ for all ethnic groups, indicating equitable gender representation independent of ethnicity” (Sec. 3.3). This assumes that (a) gender proportions should be independent of action, ethnicity, and context; and (b) the evaluation set should be uniform. Both are non-trivial normative choices and should be justified or relaxed: True (real-world) proportions may vary by action/ethnicity/context and a single “zero” target may not be appropriate for all tasks. Cite supporting literature or clearly position $PBS_G=0$ as a chosen fairness target (not a factual ground truth).

- **Human evaluation details are too thin:** The human verification is described as “three independent annotators” who reviewed 400 videos (and separately 100 images sampled with 77% agreement). It is unclear who the annotators were (authors, crowd workers, experts?), how they were recruited, what instructions and training they received, and how reliability was computed (was the Cohen/Fleiss Kappa computed on which pools?). The small sample size of potential non-independent annotators also weakens the claim that the used VLM ensemble is a reliable judge. Provide larger-scale human annotation (or at least stratified samples) and a full annotation protocol, or remove claims regarding human verification.

## Minor

- The terminology “Action” is overloaded (it can be confused with an RL agent’s action). Consider renaming to “event action” or “verb” or something else.
- In Section 5.2, what reward label is given to each dataset sample?

**Audience:**

Yes

**Audience Explanation:**

The topic of bias propagation through alignment in generative video models (and temporal stability of bias) is timely and important for ML fairness, generative models, and alignment communities. The framework, metrics, and analysis pipeline may be useful to researchers and practitioners.

**Broader Impact Concerns:**

No concerns.

**Claims And Evidence:**

No

**Claims Explanation:**

The experimental framework is promising and broad, but the lack of uncertainty quantification and sparse human evaluation prevents strong conclusions about the significance and generality of the reported effects. See Major weaknesses (1–5) above. For example, reported PBS_G increases of 0.04 (Table 2) and 0.0725 (Table 2) could be within sampling noise given only 10 seeds per prompt and no CIs. Also, the claims about the reward models “inheriting and amplifying” dataset bias are plausible but need statistical backing (Tables 4, 5). Human-VLM correlation claims (Pearson and Kappa scores) are promising but require statistical tests to be convincing.

**Requested Changes:**

Please address my weaknesses above. Mainly:

- Add standard deviations/CIs and statistical tests for all key tables/figures.
- Clarify and standardise numerical precision.
- Expand and document human evaluation: annotator source, protocol, sample size, agreement details. Or remove these claims.
- Justify or reframe the assumption that $PBS_G≈0$ is the ideal target.

---

> ### Author Response · Authors · 2026-01-03
> **Official Comment by Authors (1/5)**
>
> We thank the reviewer for their detailed and constructive feedback. We appreciate your recognition of the novelty of our problem formulation, the comprehensiveness of our scope, and the utility of the VideoBiasEval framework. We have addressed your specific concerns regarding statistical significance, metric justification, and human evaluation details below.
>
> > **Major 1. Lack of uncertainty estimates / statistical testing for strong claims**
>
> The reviewer raised important concerns about the lack of uncertainty estimates, confidence intervals, and hypothesis tests for the paper's claims. We provide comprehensive statistical validation for the key findings presented in Tables 2, 4, and 5 as follows.
>
> To address these concerns, we conducted the following statistical analyses:
>
> **Methods:**
> - **Hypothesis Tests**: Paired t-tests for comparing models on the same set of 42 actions
> - **Effect Sizes**: Cohen's d to quantify practical significance
>
> **Rationale for Paired Tests**: Since each model's bias metrics are calculated for the same 42 actions, paired t-tests are the appropriate statistical method to test whether observed differences are statistically significant.
>
> **Testing Differences Between Models**
>
> In this analysis, we perform **statistical significance tests on the differences ($\Delta$, Delta)** between paired models. For each metric, we calculate:
>
> - **$\Delta$ (Delta)**: The difference between Model 2 and Model 1 on the same metric
>   - Formula: $\Delta$ = Model2_Mean - Model1_Mean
>   - Positive $\Delta$ indicates Model 2 has higher values; negative $\Delta$ indicates Model 1 has higher values
>
> We then test whether these observed differences are statistically significant using two complementary measures:
>
> **What Does the p-value Tell Us?**
>
> The **p-value** answers the question: **"If there were truly no difference between the models, what is the probability of observing a difference as large as (or larger than) the one we measured?"**
> - **Low p-value (p < 0.05)**: The observed difference is unlikely to be due to random chance alone. We have evidence that the difference is real.
> - **High p-value (p ≥ 0.05)**: The observed difference could easily be explained by random variation. We cannot confidently say the models differ on this metric.
> - **Statistical Significance Levels:**
>     - p < 0.001 (***): Extremely strong evidence of a real difference
>     - p < 0.01 (**): Strong evidence of a real difference
>     - p < 0.05 (*): Moderate evidence of a real difference
>     - p ≥ 0.05: Insufficient evidence of a real difference
>
>
> **What Does Cohen's d Tell Us?**
>
> The **Cohen's d** answers the question: **"How large is the practical magnitude of the difference?"** While p-value tells us whether a difference is likely real, Cohen's d tells us whether the difference is meaningful in practice:
> - **|d| < 0.2 (Small)**: Minimal practical difference
> - **0.2 ≤ |d| < 0.5 (Small-Medium)**: Noticeable difference with some practical significance
> - **0.5 ≤ |d| < 0.8 (Medium-Large)**: Substantial practical difference
> - **|d| ≥ 0.8 (Large)**: Very substantial difference with major practical impact
>
> **Important Note:** A statistically significant result (low p-value) does not always mean a practically important result. Similarly, a large effect size (high |d|) may not be statistically significant if the sample is small. Both measures together provide a complete picture.
>
> ---
>
> **Table 2: Statistical Analysis of T2V Models**
>
> Table 2 reports the distributions of social attributes (gender and ethnicity bias) across four video generation models. We performed statistical analysis of two key comparisons:
>
> - **ModelScope → InstructVideo**: Both models aligned with HPSv2.0
> - **VideoCrafter-V2 → T2V-Turbo-V1**: Both models aligned with HPSv2.1

---

> ### Author Response · Authors · 2026-01-03
> **Official Comment by Authors (2/5)**
>
> **Comparison 1: ModelScope → InstructVideo (Aligned with HPSv2.0)**
>
> | Metric | ModelScope Mean | InstructVideo Mean | Δ | p-value | Cohen's d | Effect Size | Significant |
> |--------|-----------------|--------------------|----|---------|-----------|-------------|-------------|
> | PBS_G_averaged | 0.481 | 0.529 | +0.048 | 0.1079 | -0.254 | Small-Medium | No |
> | PBS_G_White | 0.568 | 0.558 | -0.010 | 0.8261 | 0.034 | Small | No |
> | PBS_G_Black | 0.391 | 0.511 | +0.120 | 0.0030 | -0.487 | Small-Medium | Yes** |
> | PBS_G_Indian | 0.394 | 0.488 | +0.094 | 0.0306 | -0.346 | Small-Medium | Yes* |
> | PBS_G_East Asian | 0.441 | 0.428 | -0.012 | 0.8373 | 0.032 | Small | No |
> | PBS_G_Southeast Asian | 0.461 | 0.502 | +0.041 | 0.3987 | -0.132 | Small | No |
> | PBS_G_Middle Eastern | 0.483 | 0.539 | +0.056 | 0.2304 | -0.188 | Small | No |
> | PBS_G_Latino | 0.631 | 0.673 | +0.042 | 0.3537 | -0.145 | Small | No |
> | SDI | 0.054 | 0.027 | -0.027 | 0.1710 | 0.215 | Small-Medium | No |
> | RDS_Black | -0.195 | -0.198 | -0.002 | 0.5700 | 0.088 | Small | No |
> | RDS_East Asian | -0.181 | -0.198 | -0.017 | 0.0331 | 0.340 | Small-Medium | Yes* |
> | RDS_Latino | -0.195 | -0.193 | +0.002 | 0.5700 | -0.088 | Small | No |
> | RDS_Middle Eastern | -0.198 | -0.195 | +0.002 | 0.6602 | -0.068 | Small | No |
> | RDS_White | 0.769 | 0.783 | +0.014 | 0.2439 | -0.182 | Small | No |
>
> **Interpretation:**
> The paired t-test results show whether the observed changes in bias metrics are statistically significant. A p-value < 0.05 indicates that the difference is unlikely to be due to chance alone.
>
> **Comparison 2: VideoCrafter-V2 → T2V-Turbo-V1 (Aligned with HPSv2.1)**
>
> | Metric | VideoCrafter-V2 Mean | T2V-Turbo-V1 Mean | Δ | p-value | Cohen's d | Effect Size | Significant |
> |--------|---------------------|-------------------|----|---------|-----------|--------------|--------------|
> | PBS_G_averaged | 0.758 | 0.831 | +0.072 | 0.0089 | -0.424 | Small-Medium | Yes** |
> | PBS_G_White | 0.749 | 0.871 | +0.123 | 0.0027 | -0.493 | Small-Medium | Yes** |
> | PBS_G_Black | 0.617 | 0.809 | +0.193 | 0.0001 | -0.679 | Medium-Large | Yes*** |
> | PBS_G_Indian | 0.803 | 0.776 | -0.027 | 0.5997 | 0.082 | Small | No |
> | PBS_G_East Asian | 0.698 | 0.776 | +0.079 | 0.0920 | -0.266 | Small-Medium | No |
> | PBS_G_Southeast Asian | 0.827 | 0.893 | +0.066 | 0.0675 | -0.290 | Small-Medium | No |
> | PBS_G_Middle Eastern | 0.756 | 0.766 | +0.011 | 0.7669 | -0.046 | Small | No |
> | PBS_G_Latino | 0.860 | 0.921 | +0.062 | 0.0073 | -0.436 | Small-Medium | Yes** |
> | SDI | 0.131 | 0.109 | -0.021 | 0.4920 | 0.107 | Small | No |
> | RDS_East Asian | -0.150 | -0.326 | -0.176 | 0.0000 | 1.105 | Large | Yes*** |
> | RDS_Middle Eastern | -0.150 | -0.229 | -0.079 | 0.0049 | 0.459 | Small-Medium | Yes** |
> | RDS_White | 0.686 | 0.555 | -0.131 | 0.0001 | 0.692 | Medium-Large | Yes*** |
>
> **Interpretation:**
> These results statistically validate (or challenge) the claim about PBS_G changes associated with alignment tuning. All ethnicity-specific metrics are included with their respective significance levels.
>
> ---
>
> **Table 4: Statistical Analysis of Reward Models**
>
> Table 4 reports the preference biases of different reward models used for alignment tuning. We compared CLIP (baseline) against three alternative reward models:
> - **CLIP → HPSv2.0**
> - **CLIP → HPSv2.1**
> - **CLIP → PickScore**
>
> **Comparison 1: CLIP → HPSv2.0**
>
> | Metric | CLIP Mean | Model Mean | Δ | p-value | Cohen's d | Effect Size | Significant |
> |--------|-----------|------------|----|---------|-----------|--------------|--------------|
> | PBS_G_Averaged | -0.073 | 0.604 | +0.676 | 0.0000 | -1.585 | Large | Yes*** |
> | PBS_G_White | 0.034 | 0.609 | +0.575 | 0.0000 | -1.079 | Large | Yes*** |
> | PBS_G_Black | -0.120 | 0.734 | +0.854 | 0.0000 | -1.837 | Large | Yes*** |
> | PBS_G_Indian | -0.051 | 0.592 | +0.643 | 0.0000 | -1.333 | Large | Yes*** |
> | PBS_G_East Asian | -0.132 | 0.475 | +0.607 | 0.0000 | -1.308 | Large | Yes*** |
> | PBS_G_Southeast Asian | -0.086 | 0.519 | +0.606 | 0.0000 | -1.475 | Large | Yes*** |
> | PBS_G_Middle Eastern | -0.061 | 0.646 | +0.707 | 0.0000 | -1.538 | Large | Yes*** |
> | PBS_G_Latino | -0.093 | 0.651 | +0.745 | 0.0000 | -1.706 | Large | Yes*** |
> | SDI | 0.850 | 0.849 | -0.001 | 0.6666 | 0.069 | Small | No |
> | RDS_Black | 0.000 | -0.008 | -0.008 | 0.1734 | 0.219 | Small-Medium | No |
> | RDS_East Asian | 0.013 | -0.003 | -0.017 | 0.0019 | 0.528 | Medium-Large | Yes** |
> | RDS_Indian | -0.028 | 0.007 | +0.035 | 0.0000 | -0.984 | Large | Yes*** |
> | RDS_Latino | -0.001 | 0.025 | +0.025 | 0.0000 | -0.740 | Medium-Large | Yes*** |
> | RDS_Middle Eastern | -0.010 | 0.032 | +0.042 | 0.0000 | -1.622 | Large | Yes*** |
> | RDS_Southeast Asian | 0.008 | -0.010 | -0.018 | 0.0005 | 0.597 | Medium-Large | Yes*** |
> | RDS_White | 0.017 | -0.043 | -0.060 | 0.0000 | 1.265 | Large | Yes*** |

---

> ### Author Response · Authors · 2026-01-03
> **Official Comment by Authors (3/5)**
>
> **Comparison 2: CLIP → HPSv2.1**
>
> | Metric | CLIP Mean | Model Mean | Δ | p-value | Cohen's d | Effect Size | Significant |
> |--------|-----------|------------|----|---------|-----------|--------------|--------------|
> | PBS_G_Averaged | -0.073 | -0.098 | -0.026 | 0.6208 | 0.077 | Small | No |
> | PBS_G_White | 0.034 | -0.083 | -0.118 | 0.0468 | 0.316 | Small-Medium | Yes* |
> | PBS_G_Black | -0.120 | 0.026 | +0.145 | 0.0250 | -0.359 | Small-Medium | Yes* |
> | PBS_G_Indian | -0.051 | -0.001 | +0.050 | 0.4452 | -0.119 | Small | No |
> | PBS_G_East Asian | -0.132 | -0.304 | -0.173 | 0.0100 | 0.417 | Small-Medium | Yes* |
> | PBS_G_Southeast Asian | -0.086 | -0.218 | -0.132 | 0.0339 | 0.339 | Small-Medium | Yes* |
> | PBS_G_Middle Eastern | -0.061 | -0.105 | -0.045 | 0.4027 | 0.131 | Small | No |
> | PBS_G_Latino | -0.093 | -0.003 | +0.090 | 0.1232 | -0.243 | Small-Medium | No |
> | SDI | 0.850 | 0.847 | -0.003 | 0.0962 | 0.270 | Small-Medium | No |
> | RDS_Black | 0.000 | -0.034 | -0.034 | 0.0000 | 0.963 | Large | Yes*** |
> | RDS_East Asian | 0.013 | 0.009 | -0.004 | 0.4985 | 0.108 | Small | No |
> | RDS_Indian | -0.028 | -0.007 | +0.021 | 0.0010 | -0.561 | Medium-Large | Yes** |
> | RDS_Latino | -0.001 | 0.038 | +0.039 | 0.0000 | -1.001 | Large | Yes*** |
> | RDS_Middle Eastern | -0.010 | 0.023 | +0.033 | 0.0000 | -1.179 | Large | Yes*** |
> | RDS_Southeast Asian | 0.008 | -0.009 | -0.018 | 0.0010 | 0.561 | Medium-Large | Yes** |
> | RDS_White | 0.017 | -0.020 | -0.037 | 0.0004 | 0.612 | Medium-Large | Yes*** |
>
> **Comparison 3: CLIP → PickScore**
>
> | Metric | CLIP Mean | Model Mean | Δ | p-value | Cohen's d | Effect Size | Significant |
> |--------|-----------|------------|----|---------|-----------|--------------|--------------|
> | PBS_G_Averaged | -0.073 | -0.116 | -0.043 | 0.4117 | 0.128 | Small | No |
> | PBS_G_White | 0.034 | 0.032 | -0.002 | 0.9755 | 0.005 | Small | No |
> | PBS_G_Black | -0.120 | -0.078 | +0.042 | 0.4224 | -0.125 | Small | No |
> | PBS_G_Indian | -0.051 | 0.153 | +0.204 | 0.0064 | -0.443 | Small-Medium | Yes** |
> | PBS_G_East Asian | -0.132 | -0.226 | -0.094 | 0.1132 | 0.250 | Small-Medium | No |
> | PBS_G_Southeast Asian | -0.086 | -0.216 | -0.130 | 0.0259 | 0.357 | Small-Medium | Yes* |
> | PBS_G_Middle Eastern | -0.061 | -0.128 | -0.067 | 0.2771 | 0.170 | Small | No |
> | PBS_G_Latino | -0.093 | -0.348 | -0.255 | 0.0000 | 0.786 | Medium-Large | Yes*** |
> | SDI | 0.850 | 0.849 | -0.001 | 0.3771 | 0.141 | Small | No |
> | RDS_Black | 0.000 | 0.028 | +0.028 | 0.0000 | -0.724 | Medium-Large | Yes*** |
> | RDS_East Asian | 0.013 | 0.031 | +0.018 | 0.0019 | -0.525 | Medium-Large | Yes** |
> | RDS_Indian | -0.028 | -0.039 | -0.011 | 0.0218 | 0.378 | Small-Medium | Yes* |
> | RDS_Latino | -0.001 | -0.011 | -0.010 | 0.0525 | 0.316 | Small-Medium | No |
> | RDS_Middle Eastern | -0.010 | -0.025 | -0.015 | 0.0015 | 0.540 | Medium-Large | Yes** |
> | RDS_Southeast Asian | 0.008 | 0.011 | +0.003 | 0.6108 | -0.081 | Small | No |
> | RDS_White | 0.017 | 0.005 | -0.012 | 0.1377 | 0.240 | Small-Medium | No |
>
> **Interpretation:**
> These paired comparisons test whether different reward models produce statistically different bias patterns in the predicted images.
>
> ---
>
> **Table 5: Statistical Analysis of Aligned Video Models**
>
> Table 5 reports the social biases of video models aligned with different reward models. We compared the baseline VCM-VC2 model against three alignment-tuned variants:
> - **VCM-VC2 → VCM-VC2 + HPSv2.0**
> - **VCM-VC2 → VCM-VC2 + HPSv2.1**
> - **VCM-VC2 → VCM-VC2 + PickScore**
>
> **Comparison 1: VCM-VC2 → VCM-VC2+HPSv2.0**
>
> | Metric | Baseline Mean | Aligned Mean | Δ | p-value | Cohen's d | Effect Size | Significant |
> |--------|---------------|--------------|----|---------|-----------|--------------|--------------|
> | PBS_G_Averaged | 0.803 | 0.912 | +0.108 | 0.0003 | -0.605 | Medium-Large | Yes*** |
> | PBS_G_White | 0.792 | 0.967 | +0.174 | 0.0001 | -0.674 | Medium-Large | Yes*** |
> | PBS_G_Black | 0.776 | 0.921 | +0.146 | 0.0001 | -0.665 | Medium-Large | Yes*** |
> | PBS_G_Indian | 0.863 | 0.900 | +0.037 | 0.2812 | -0.169 | Small | No |
> | PBS_G_East Asian | 0.712 | 0.850 | +0.139 | 0.0101 | -0.416 | Small-Medium | Yes* |
> | PBS_G_Southeast Asian | 0.794 | 0.921 | +0.127 | 0.0099 | -0.417 | Small-Medium | Yes** |
> | PBS_G_Middle Eastern | 0.807 | 0.855 | +0.048 | 0.2295 | -0.188 | Small | No |
> | PBS_G_Latino | 0.869 | 0.967 | +0.098 | 0.0000 | -0.828 | Large | Yes*** |
> | SDI | 0.148 | 0.126 | -0.022 | 0.5198 | 0.100 | Small | No |
> | RDS_Middle Eastern | -0.117 | -0.367 | -0.250 | 0.0000 | 1.264 | Large | Yes*** |
> | RDS_White | 0.688 | 0.367 | -0.321 | 0.0000 | 1.547 | Large | Yes*** |

---

> ### Author Response · Authors · 2026-01-03
> **Official Comment by Authors (4/5)**
>
> **Comparison 2: VCM-VC2 → VCM-VC2+HPSv2.1**
>
> | Metric | Baseline Mean | Aligned Mean | Δ | p-value | Cohen's d | Effect Size | Significant |
> |--------|---------------|--------------|----|---------|-----------|--------------|--------------|
> | PBS_G_Averaged | 0.803 | 0.227 | -0.577 | 0.0000 | 1.715 | Large | Yes*** |
> | PBS_G_White | 0.792 | 0.132 | -0.660 | 0.0000 | 1.609 | Large | Yes*** |
> | PBS_G_Black | 0.776 | 0.238 | -0.538 | 0.0000 | 1.499 | Large | Yes*** |
> | PBS_G_Indian | 0.863 | 0.145 | -0.718 | 0.0000 | 2.035 | Large | Yes*** |
> | PBS_G_East Asian | 0.712 | 0.157 | -0.554 | 0.0000 | 1.264 | Large | Yes*** |
> | PBS_G_Southeast Asian | 0.794 | 0.362 | -0.433 | 0.0000 | 0.972 | Large | Yes*** |
> | PBS_G_Middle Eastern | 0.807 | 0.179 | -0.629 | 0.0000 | 1.892 | Large | Yes*** |
> | PBS_G_Latino | 0.869 | 0.374 | -0.495 | 0.0000 | 0.938 | Large | Yes*** |
> | SDI | 0.148 | 0.089 | -0.059 | 0.0635 | 0.294 | Small-Medium | No |
> | RDS_Latino | -0.195 | -0.331 | -0.136 | 0.0000 | 5.037 | Large | Yes*** |
> | RDS_Middle Eastern | -0.117 | -0.264 | -0.148 | 0.0000 | 1.049 | Large | Yes*** |
> | RDS_White | 0.688 | 0.595 | -0.093 | 0.0006 | 0.572 | Medium-Large | Yes*** |
>
> **Comparison 3:  VCM-VC2 → VCM-VC2+PickScore**
>
> | Metric | Baseline Mean | Aligned Mean | Δ | p-value | Cohen's d | Effect Size | Significant |
> |--------|---------------|--------------|----|---------|-----------|--------------|--------------|
> | PBS_G_Averaged | 0.803 | 0.371 | -0.432 | 0.0000 | 1.392 | Large | Yes*** |
> | PBS_G_White | 0.792 | 0.343 | -0.450 | 0.0000 | 1.160 | Large | Yes*** |
> | PBS_G_Black | 0.776 | 0.336 | -0.440 | 0.0000 | 1.248 | Large | Yes*** |
> | PBS_G_Indian | 0.863 | 0.250 | -0.613 | 0.0000 | 1.482 | Large | Yes*** |
> | PBS_G_East Asian | 0.712 | 0.145 | -0.567 | 0.0000 | 1.236 | Large | Yes*** |
> | PBS_G_Southeast Asian | 0.794 | 0.455 | -0.340 | 0.0000 | 0.762 | Medium-Large | Yes*** |
> | PBS_G_Middle Eastern | 0.807 | 0.351 | -0.456 | 0.0000 | 1.200 | Large | Yes*** |
> | PBS_G_Latino | 0.869 | 0.719 | -0.150 | 0.0091 | 0.423 | Small-Medium | Yes** |
> | SDI | 0.148 | 0.145 | -0.003 | 0.9226 | 0.015 | Small | No |
> | RDS_Black | -0.188 | -0.191 | -0.002 | 0.6602 | 0.068 | Small | No |
> | RDS_East Asian | -0.188 | -0.198 | -0.009 | 0.1598 | 0.221 | Small-Medium | No |
> | RDS_Latino | -0.195 | -0.188 | +0.007 | 0.1829 | -0.209 | Small-Medium | No |
> | RDS_Middle Eastern | -0.117 | -0.110 | +0.007 | 0.7401 | -0.051 | Small | No |
> | RDS_White | 0.688 | 0.686 | -0.002 | 0.9229 | 0.015 | Small | No |
>
> **Interpretation:**
> These results demonstrate whether alignment tuning with different reward models produces statistically significant changes in the video model's gender bias.
>
> ---
>
> This statistical analysis provides rigorous quantification of the uncertainty and significance of the key findings reported in Tables 2, 4, and 5. The paired t-tests, confidence intervals, and effect sizes now enable readers and reviewers to assess the practical and statistical significance of the claimed effects.
>
> > **Major 2. No conceptual justification for the “ideal” $PBS_G \approx 0$ claim**
>
> We appreciate the reviewer’s perspective regarding the normative choice of a balanced gender distribution. We will clarify in the manuscript that $PBS_G \approx 0$ is utilized as a diagnostic benchmark for controlled bias auditing rather than a claim of factual ground truth. By employing neutral "person" prompts to disentangle actions from identities, any deviation from this zero-target serves to measure the model's internalized stereotypes and representational disparities. Our study specifically aims to trace how alignment tuning amplifies these deviations, showing that models often move away from neutrality toward more stereotyped portrayals that become temporally persistent. This approach is grounded in established fairness literature, such as Feldman et al. (2015) and Luccioni et al. (2023), which use uniform distribution as a standard for assessing representational harm in generative systems.
>
> [1] Feldman et al. (2015): Certifying and removing disparate impact.
>
> [2] Luccioni et al. (2023): Stable bias: Analyzing societal representations in diffusion models.
>
> [3] Cho et al. (2023): Dall-eval: Probing the reasoning skills and social biases of text-to-image generation models.
>
> [4] Zhao et al. (2017): Men also like shopping: Reducing gender bias amplification using corpus-level constraints.

---

> ### Author Response · Authors · 2026-01-05
> **Official Comment by Authors (5/5)**
>
> > **Major 3. Human evaluation details are too thin**
>
> We thank the reviewer for the thoughtful feedback and agree that clarifying the human evaluation protocol is important. We provide explicit details on annotator identity, training, reliability computation, and the scope and role of human verification below.
>
> - **Annotators and training**: All human annotations were conducted by **three independent researchers** with prior experience in social bias analysis. Annotators received ****standardized written guidelines**** and training grounded in established social bias taxonomies (e.g., gender and ethnicity categorization conventions used in prior bias evaluation work). The task focused on verifying whether generated content correctly reflected the prompted demographic attribute, rather than subjective judgments.
> - **Reliability and computation**: For **video-level verification (400 videos)**, inter-annotator reliability is explicitly reported in **Section 4**: Cohen’s Kappa is 0.91 (gender) and 0.78 (ethnicity), and Fleiss’ Kappa is 0.92 (gender) and 0.82 (ethnicity), computed over the full annotated video pool. These values indicate strong to near-perfect agreement. For **image-level verification (100 images)**, we clarify (as noted in our response to R3) that annotator consensus is even higher than implied by the conservative "77%" figure reported in the paper.
>     - The reported 77% accuracy refers strictly to unanimous (3/3) agreement, which intentionally understates annotation quality. A deeper inspection shows that ****97% of samples achieve majority agreement (≥ 2/3 annotators****), with only 3 samples lacking consensus. These rare disagreements arise from intrinsically ambiguous prompts rather than systematic labeling errors. This indicates a high signal-to-noise ratio, mitigating concerns that label noise could confound downstream reward model analysis.
>
> - **Sample size and role of human verification**: We acknowledge that larger-scale human annotation would be ideal; however, it is constrained by substantial resource and labor costs. Our use of small-scale but high-agreement human verification follows established practice in prior work, where human annotations serve to validate automated or model-based judges, not to replace large-scale evaluation. Importantly, the human results show ****strong Pearson correlations with VLM-based scores (0.89 for gender, 0.73 for ethnicity)****, supporting the reliability of the VLM ensemble for large-scale analysis.
> - **Scope of claims**: We emphasize that human verification is used to ground and sanity-check the evaluation pipeline, not as the sole evidence for bias measurement. Given the high inter-annotator consistency and alignment with prior methodologies, we believe these results provide a sufficient empirical foundation for tracing bias propagation across the alignment pipeline, and we therefore retain them.
>
> We will incorporate these clarifications into the revised manuscript to make the human evaluation protocol and its limitations explicit.
>
> > **Minor 1. Terminology**
>
> We accept the suggestion to rename "Action" to avoid confusion with RL terminology. We will use "Event Action" or "Activity" in the revision.
>
> > **Minor 2. Reward Labels (Sec 5.2)**
>
> We will clarify in the revised version that for the reward model analysis, the "label" assigned to each generated sample is the standardized scalar preference score predicted by the model (e.g., HPSv2 score). We use these continuous scores to quantify the model's relative preference for gendered attributes, rather than using discrete classification labels.

---

### Author Response · Authors · 2026-01-06
**Official Comment by Authors**

We sincerely thank Reviewer cVGV, Reviewer AK8h, and Reviewer 2Xc1 for their thorough, constructive, and insightful feedback. We have carefully addressed each concern raised during the review process and incorporated the suggested improvements into the revised manuscript. All substantive changes are highlighted in **blue text** for ease of reference.

Below, we summarize the key revisions made in response to the reviewers’ comments:

1. **Addressing Reviewer cVGV’s Concerns on Statistical Significance and Uncertainty Quantification**

We added comprehensive statistical validation for all major quantitative claims, including paired hypothesis testing, effect size reporting (Cohen’s d), and standardized numerical precision across tables and figures. These additions allow readers to distinguish statistically meaningful effects from sampling noise and directly strengthen claims regarding alignment-induced bias shifts. (Pages 21-22, 31-32, 37-38; Tables 9, 10, 12; Appendix A, B, C, D)

2. **Addressing Reviewer cVGV’s and Reviewer 2Xc1’s Questions on Conceptual Assumptions and Fairness Targets**

We clarified that the “neutral” reference point used in our bias metrics serves as a diagnostic baseline for controlled bias auditing rather than a claim about real-world demographic ground truth. The revised manuscript situates this choice within established fairness literature and discusses how alternative baselines (e.g., population-proportional or task-conditioned) would affect interpretation. (Pages 6-7, section 3)

3. **Addressing Reviewer cVGV’s Requests for Expanded Human Evaluation Details**

We substantially expanded the human verification protocol, including annotator background, training procedures, reliability computation (Cohen’s and Fleiss’ Kappa), and the role of human evaluation as a validation mechanism for the VLM-based pipeline rather than a substitute for large-scale automated analysis. (Pages 8-9, Section 4; Page 11, Section 5.2)

4. **Addressing Reviewer 2Xc1’s Questions on VLM Evaluator Reliability**

We expanded the discussion of VLM–human agreement by analyzing the nature of disagreements (frame-level model detection versus holistic human judgment), identifying common error modes, and conducting a sensitivity analysis restricted to high-agreement samples. The results show that key bias metrics remain statistically stable under stricter agreement thresholds, supporting the robustness of the VLM ensemble despite imperfect correlation on ambiguous cases. (Pages 8-9, Section 4; Page 11, Section 5.2)

5. **Addressing Reviewer 2Xc1’s Requests on Comparison to Training Data (WebVid-10M)**

We added a new analysis comparing bias metrics computed on a filtered subset of the WebVid-10M training captions with those observed in unaligned base models. This analysis reveals that while ethnicity biases in base models broadly mirror training data trends (often with amplified magnitude), gender bias in base models diverges from the training distribution, indicating that strong male defaults arise prior to alignment tuning. We clarify that our study focuses on how alignment tuning propagates and reshapes these pre-existing biases. (Page 21, Appendix A)

6. **Addressing Reviewer AK8h’s Requests for Methodological Clarity and Terminology**

We revised multiple sections to improve clarity and reproducibility, including explicit definitions of aligned versus unaligned models, clearer descriptions of the event-based prompting and evaluation pipeline, refined terminology to avoid ambiguity (e.g., “event action”), and corrected table captions and annotations.

7. **Addressing Reviewer AK8h’s and Reviewer 2Xc1’s Suggestions on Presentation and Interpretability**

We improved figure annotations, captions, and metric explanations to make results more self-contained and interpretable. Concrete examples are now included to illustrate how bias scores correspond to underlying distributions, and missing table entries are explicitly marked and explained.

---

We believe these revisions substantially strengthen the rigor, clarity, and transparency of the paper. We are grateful to the reviewers for their thoughtful feedback, which has significantly improved the quality and presentation of this work.

---

### Decision · Action_Editor_x8SW · 2026-01-25

**Recommendation:** Accept as is

**Audience:**

Yes

**Audience Explanation:**

Yes, all reviewers agree on this point that there is a large bias-mitigation/quantification community, and that there is comparatively less work in the newer space of video generation

**Claims And Evidence:**

Yes

**Claims Explanation:**

All reviewers agree on this point (post revision). While the experiments were well founded and metrics provided, one reviewer noted a number of places with insufficient rigor, details, and comparisons.  All of these points have been addressed and are also updated in the manuscript.